# BENIGN OR NOT-BENIGN OVERFITTING IN TOKEN SELECTION OF ATTENTION MECHANISM

## ABSTRACT

Modern over-parameterized neural networks can be trained to fit the training data perfectly while still maintaining a high generalization performance. This "benign overfitting" phenomenon has been studied in a surge of recent theoretical work; however, most of these studies have been limited to linear models or two-layer neural networks. In this work, we analyze benign overfitting in the token selection mechanism of the attention architecture, which characterizes the success of transformer models. We first show the existence of a benign overfitting solution and explain its mechanism in the attention architecture. Next, we discuss whether the model converges to such a solution, raising the difficulties specific to the attention architecture. We then present benign overfitting cases and not-benign overfitting cases by conditioning different scenarios based on the behavior of attention probabilities during training. To the best of our knowledge, this is the first study to characterize benign overfitting for the attention mechanism.

## 1 INTRODUCTION

Modern over-parameterized neural networks achieve a high generalization accuracy while perfectly fitting to the training data (Zhang et al., 2021). This "benign overfitting" phenomenon has attracted attention over the past few years because it contrasts with the conventional wisdom that achieving better generalization requires balancing training error and model complexity through appropriate regularization techniques, which prevents overfitting to training noise. The theoretical understanding of benign overfitting is crucial for obtaining insights into over-parameterized networks.

There are lines of studies analyzing the benign overfitting phenomenon in various settings, linear regression (Bartlett et al., 2020; Tsigler & Bartlett, 2023), linear classification (Chatterji & Long, 2021; Cao et al., 2021), and two-layer neural networks (Frei et al., 2022; Cao et al., 2022). However, the analysis of benign overfitting is limited to these types of architecture. As far as we know, there have been no theoretical analyses on attention architecture, which is a core component of the transformer model (Vaswani et al., 2017). Given the current success of transformer models across a wide range of fields (Brown et al., 2020; Baevski et al., 2020; Dosovitskiy et al., 2021; Wei et al., 2022b;a; Touvron et al., 2023; Chowdhery et al., 2023), analyzing the attention mechanism that distinguishes them from other architectures has become increasingly significant.

The concept of benign overfitting in transformers is not yet well-defined. We do not even know what aspects should be analyzed in the first place. As a first step towards addressing this problem, we focus on analyzing token selection, a key property of the attention mechanism that characterizes transformers. We consider a one-layer attention network $f(\mathbf{X}) = \boldsymbol{\nu}^\top \mathbf{X}^\top \mathbb{S}(\mathbf{X}\mathbf{W}^\top \mathbf{p}) \in \mathbb{R}$, where $\mathbb{S}(\cdot)$ is the softmax function, $\mathbf{X} = (\mathbf{x}_1, \ldots, \mathbf{x}_T)^\top \in \mathbb{R}^{T \times d}$ is the sequence of input tokens, $\mathbf{p} \in \mathbb{R}^d$ is a tunable token, $\mathbf{W} \in \mathbb{R}^{d \times d}$ is the key-query matrix, and $\boldsymbol{\nu} \in \mathbb{R}^d$ is a well-pretrained linear head, on a separated binary classification task. Here, $\mathbf{p}$ corresponds to the [CLS] token or prompt tuning in the application of transformers, and we followed the common setup studied in other topics, such as implicit bias (Tarzanagh et al., 2023a;b; Oymak et al., 2023). Our main contributions are summarized as follows:

- **Existence of Benign Overfitting Solution (Thm 4.1):** We first prove the existence of a benign overfitting solution and explain its mechanism in the token selection of attention architecture. Specifically, in addition to learning the signal with class information, benign overfitting is achieved

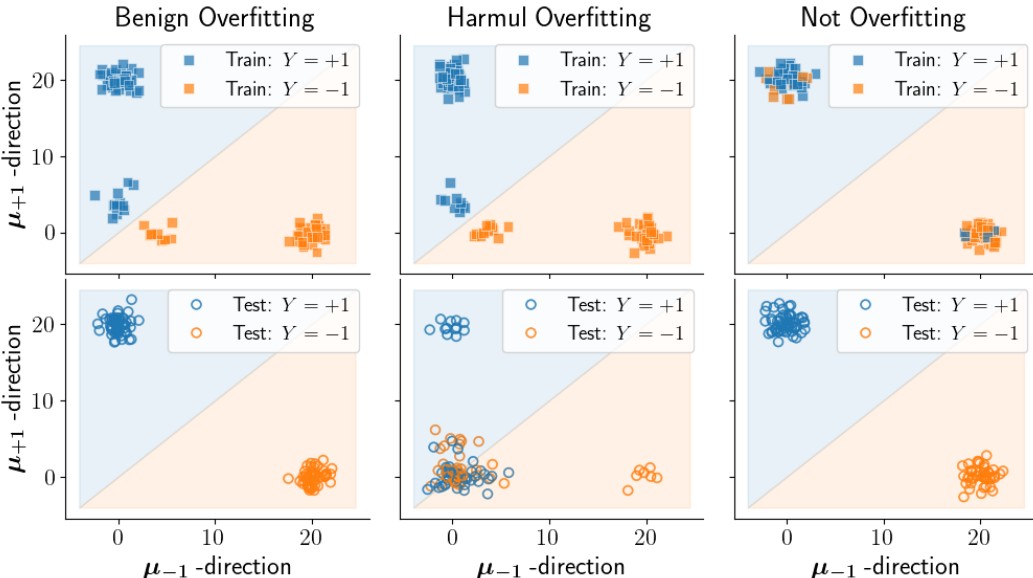

Figure 1: Projection of one selected token per sequence. Each point indicates $\mathbf{x}_t^{(i)}$ selected by attention from each input $\mathbf{X}^{(i)} = (\mathbf{x}_1^{(i)}, \ldots, \mathbf{x}_T^{(i)})^\top$ in the direction of class signals $\boldsymbol{\mu}_{+1}$ and $\boldsymbol{\mu}_{-1}$ for the three scenarios of benign overfitting, harmful overfitting, and not overfitting. **Top:** Training data with label noise. **Bottom:** Test data. The decision boundary is common because the output head $\boldsymbol{\nu}$ is fixed, but **the model can select an appropriate token that belongs to the desired output region.**

by memorizing tokens that can fit the label noise. In this scenario, tokens that are useful for class prediction are selected for the unseen data, but for noisy data in the training set, tokens that can adapt to the noise are picked, suppressing the class-relevant tokens. This aligns with the conventional understanding of benign overfitting: the principal components explain the overall prediction, and noise memorization accounts for fitting noise. Please refer to Figure 1 for an explanation of benign overfitting in token selection, as well as other scenarios.

- **Convergence with Gradient Descent (Thm 4.2):** We discuss whether the model weights can converge to such a benign overfitting solution with gradient descent. Two factors specific to attention models complicate the straightforward application of existing analysis: (i) the existence of local minima, which trap the optimization process, and (ii) the diminishing updates of desirable parameters as learning progresses and token selection advances. Particularly due to property (ii), the signal updates in each step are not determined based solely on the label noise ratio but rather depend on the other tokens and training data in a complex manner. In this work, we show that the one-layer attention can be trained to fit the training data with gradient descent. Then, by conditioning different scenarios of the training trajectory, we present cases where benign overfitting occurs and cases where it leads to harmful overfitting.

From a practical perspective, benign overfitting solely through the token selection mechanism has implications, particularly in parameter-efficient fine-tuning. For example, prompt-tuning (Li & Liang, 2021; Lester et al., 2021) trains only the tunable input tokens, and LoRA (Hu et al., 2022) focuses on training only the attention weights. The presence of benign overfitting suggests that adapting the model to low-quality downstream tasks with label noise would not be problematic in terms of generalization. In this paper, we consider binary classification for simplicity of discussion, but as shown in Section F, it can be extended to the multi-class setting without fundamentally modifying the argument.

## 2  RELATED WORK

**Benign Overfitting.**   The success of modern over-parameterized models has led to numerous studies attempting to understand why and when benign overfitting occurs. The analysis is interesting

because the standard generalization bound based on uniform convergence does not explain this phenomenon well (Nagarajan & Kolter, 2019). For comprehensive surveys of the literature on this topic, please see work such as Bartlett et al. (2021); Belkin (2021). Please also refer to Table 2 in the appendix for the comparison of existing work, including ours.

Benign overfitting in regression has been studied in the linear model (Bartlett et al., 2020; Hastie et al., 2022) and kernel regression (Liang & Rakhlin, 2020; Tsigler & Bartlett, 2023). It is complicated to analyze the classification task because the explicit formula for the min-norm separator is not obtained. One approach is to track the training dynamics with gradient descent in the linear classifier and two-layer neural network (Chatterji & Long, 2021; Frei et al., 2022; Xu & Gu, 2023; Cao et al., 2022; Kou et al., 2023; George et al., 2023; Meng et al., 2024; Xu et al., 2024). Another line of work is built on the results of implicit bias to the max-margin solution (Cao et al., 2021; Wang et al., 2021; Wang & Thrampoulidis, 2022; Frei et al., 2023a). These studies base their discussion of convergence on existing research on implicit bias (Soudry et al., 2018; Ji & Telgarsky, 2019; Lyu & Li, 2020; Frei et al., 2023b). Specifically, they analyze the properties of the solution through the KKT conditions of the max-margin problem in order to examine whether the solution at convergence shows benign overfitting or not. For instance, in the setting of support vector machine, Hsu et al. (2021); Muthukumar et al. (2021) have shown that all training points become support vectors in a high-dimensional setup. The analysis becomes more challenging in the case of general deep neural networks, and Zhu et al. (2023) conducts an analysis under the neural tangent kernel (NTK) regime.

Emerging research directions have explored the benignity of overfitting, given that real-world models do not always exhibit benign overfitting. Mallinar et al. (2022) introduced the term "tempered overfitting" to analyze the overfitting between benign and catastrophic. Wen et al. (2023) examined tempered overfitting for the mild-overparameterized model, and Kornowski et al. (2024) investigated it for a two-layer ReLU neural network with low input dimensionality.

**Token-selection in Attention.** The attention mechanism is a core architecture in the transformer model, and theoretical analyses from the perspectives of generalization ability (Jelassi et al., 2022; Li et al., 2023a;b) and expressive ability (Yun et al., 2020a;b; Dong et al., 2021) have been progressing in recent years. The most related line of work to ours deals with the implicit bias of gradient descent in training a one-layer attention model (Tarzanagh et al., 2023a;b; Li et al., 2024; Vasudeva et al., 2024). While these studies share common aspects with our work, their primary focus is optimizing the given training set without considering the underlying distribution (see Section 4.3 for details). Furthermore, our work is also influenced by the problem setting of (Oymak et al., 2023). However, they discuss the importance of the softmax function in the attention mechanism, whereas our work is largely different in that we focus on the fitting to label noise and benign overfitting.

## 3 PROBLEM SETTING

In this section, we introduce the notation and the problem settings in the rest of the paper.

### 3.1 NOTATIONS

We use lower-case and upper-case bold letters (e.g., $\mathbf{a}$ and $\mathbf{A}$) to represent vectors and matrices, and their entries are denoted as $\mathbf{a}_i$, $\mathbf{A}_{i,j}$. Let $[n]$ be a shorthand for the set $\{1, \dots, n\}$. We denote a multivariate Gaussian distribution with mean vector $\boldsymbol{\eta}$ and covariance matrix $\boldsymbol{\Sigma}$ by $N(\boldsymbol{\eta}, \boldsymbol{\Sigma})$. Denote by $\mathbb{S} : \mathbb{R}^T \to \mathbb{R}^T, \mathbb{S}(\mathbf{v})_t = \exp(\mathbf{v}_t)/\sum_{t' \in [T]} \exp(\mathbf{v}_{t'})$ the softmax function. The standard Big-O notations $\mathcal{O}(\cdot), \Theta(\cdot)$, and $\Omega(\cdot)$ are used to hide absolute constants, and we denote equality and inequality ignoring constant factors by $\sim$ and $\gtrsim, \lesssim$, respectively. Additionally, we use the symbol $\pm$ when its meaning is clear from the context; this denotes the possible range of values.

### 3.2 ATTENTION MODEL

Given a sequential input $\mathbf{X} = (\mathbf{x}_1, \dots \mathbf{x}_T)^\top \in \mathbb{R}^{T \times d}$, a single-head self-attention layer $f_{sa} : \mathbb{R}^{T \times d} \to \mathbb{R}^{T \times m}$ is

$$f_{sa}(\mathbf{X}) = \mathbb{S}(\mathbf{X}\mathbf{W}_Q \mathbf{W}_K^\top \mathbf{X}^\top)\mathbf{X}\mathbf{W}_V,$$

with trainable weights $\mathbf{W}_Q, \mathbf{W}_K \in \mathbb{R}^{d \times d}$, and $\mathbf{W}_V \in \mathbb{R}^{d \times m}$. Here, the softmax function $\mathbb{S}(\cdot)$ is applied row-wise with the abuse of notation.

In practice, an additional tunable token $\mathbf{p} \in \mathbb{R}^d$ is concatenated to the input, and this position is used for the model prediction. This setup is widely used in, for example, the classification token [CLS] in BERT (Devlin et al., 2018) and ViT (Dosovitskiy et al., 2021), and prompt-tuning technique (Li & Liang, 2021; Lester et al., 2021). Let the concatenated input be $\mathbf{X_p} := [\mathbf{p}, \mathbf{X}^\top]^\top \in \mathbb{R}^{(T+1) \times d}$; then the cross-attention feature between $\mathbf{X_p}$ and $\mathbf{X}$ is given by

$$\begin{bmatrix} f(\mathbf{X})^\top \\ f_{sa}(\mathbf{X}) \end{bmatrix} = \mathbb{S}(\mathbf{X_p}\mathbf{W}\mathbf{X}^\top)\mathbf{X}\mathbf{W}_V = \begin{bmatrix} \mathbb{S}(\mathbf{p}^\top\mathbf{W}\mathbf{X}^\top) \\ \mathbb{S}(\mathbf{X}\mathbf{W}\mathbf{X}^\top) \end{bmatrix} \mathbf{X}\mathbf{W}_V, \tag{1}$$

where we use $\mathbf{W}$ to denote a key-query weight matrix $\mathbf{W}_Q\mathbf{W}_K^\top$, and the output corresponding to the position of $\mathbf{p}$ is denoted by $f(\mathbf{X}) = \mathbf{W}_V^\top\mathbf{X}^\top\mathbb{S}(\mathbf{X}\mathbf{W}^\top\mathbf{p}) \in \mathbb{R}^m$. In this work, we use the model output for binary classification, leading to the output dimension being $m = 1$, and we denote the value prediction head by $\boldsymbol{\nu} = \mathbf{W}_V \in \mathbb{R}^d$. Therefore, the model under our analysis is of the form

$$f(\mathbf{X}) = \boldsymbol{\nu}^\top\mathbf{X}^\top\mathbb{S}(\mathbf{X}\mathbf{W}^\top\mathbf{p}), \tag{2}$$

The output can be regarded as an affine combination of the token scores $\{\boldsymbol{\nu}^\top\mathbf{x}_t\}_{t \in [T]}$, using the learned softmax probabilities. We denote this token score by $\gamma_t := \boldsymbol{\nu}^\top\mathbf{x}_t \in \mathbb{R}$. Furthermore, let $\mathbf{s} \in \mathbb{R}^T$ be a shorthand for the softmax vector $\mathbb{S}(\mathbf{X}\mathbf{W}^\top\mathbf{p})$.

### 3.3 DATA MODEL

In the analysis of benign overfitting, we typically need to consider the specific shape of the underlying data distribution to evaluate the generalization error without using a uniform convergence argument. Data models based on signal and noise widely appear in the existing benign overfitting studies (Chatterji & Long, 2021; Frei et al., 2022; Cao et al., 2022), and ours is a natural extension to the sequential inputs of the attention model. Such data models based on signal and noise are not limited to the analysis of benign overfitting but are also commonly observed in other analyses of attention architecture (Jelassi et al., 2022; Li et al., 2023a; Oymak et al., 2023).

We consider the following data distribution $P$ defined over $(\mathbf{X}, Y) \in \mathbb{R}^{T \times d} \times \{\pm 1\}$. In this paper, we consider binary classification for simplicity of discussion, but the same argument applies to the multi-class case. Please refer to Section F in the appendix for more details.

**Definition 3.1.** Let $\boldsymbol{\mu}_{+1}, \boldsymbol{\mu}_{-1} \in \mathbb{R}^d$ be fixed signal vectors representing the class of each data point. The input $\mathbf{X} = [\mathbf{x}_1, \ldots, \mathbf{x}_T]^\top \in \mathbb{R}^{T \times d}$ has $T$ tokens that are split into three groups: *relevant token* $\mathcal{R} \in [T]$ containing the strong signal for true class, *weakly relevant token* $\mathcal{W} \in [T] \setminus \mathcal{R}$ containing weak class signals, and *irrelevant token* $\mathcal{I} = [T] \setminus (\mathcal{R} \cup \mathcal{W})$ containing only noise. Let clean distribution $P^*$ be the distribution over $\mathbb{R}^{T \times d} \times \{\pm 1\}$ such that $(\mathbf{X}, Y^*)$ is sampled as follows:

1. The clean label $Y^*$ is sampled from a uniform distribution on $\{\pm 1\}$.

2. The noise vectors $(\boldsymbol{\epsilon}_t)_{t \in [T]}$ are sampled independently from $N(\mathbf{0}, \boldsymbol{\Sigma})$ for some diagonal matrix $\boldsymbol{\Sigma} \in \mathbb{R}^{d \times d}$. We assume the diagonal elements are positive and of constant order.

3. The relevant tokens $\mathbf{x}_t, t \in \mathcal{R}$ are generated with $\mathbf{x}_t = \boldsymbol{\mu}_{Y^*} + \boldsymbol{\epsilon}_t$.

4. The weakly relevant tokens $\mathbf{x}_u, u \in \mathcal{W}$ are generated with $\mathbf{x}_u = \rho\boldsymbol{\mu}_{w_u} + \boldsymbol{\epsilon}_u$, where $\rho = \Omega(1) > 0$ is a small scale parameter, and $(w_u)_{u \in \mathcal{W}}$ are sampled from a uniform distribution on $\{\pm 1\}$. For simplicity, we assume at least one token $\mathbf{x}_u$ aligns with each class, which holds with high probability (see Lemma B.9 for details).

5. The irrelevant tokens $\mathbf{x}_v, v \in \mathcal{I}$ are generated with $\mathbf{x}_v = \boldsymbol{\epsilon}_v$.

Here, we assume the ratios of relevant and weakly relevant tokens are constant and denote them by $\zeta_\mathcal{R} = |\mathcal{R}|/T \in [1/T, 1 - 1/T]$, and $\zeta_\mathcal{W} = |\mathcal{W}|/T \in [1/T, 1 - 1/T]$, respectively. The data distribution $P$ is defined as the label-corrupted version of $P^*$ with the level of label noise $\eta > 0$. The data point $(\mathbf{X}, Y)$ from $P$ is generated by first sampling $(\mathbf{X}, Y^*)$ from clean distribution $P^*$ and then setting $Y = -Y^*$ with probability $\eta$ and $Y = Y^*$ with probability $1 - \eta$.

Training data $S = (\mathbf{X}^{(i)}, Y^{(i)})_{i=1}^n$ are sampled i.i.d. from $P$. We denote the clean data $\{i \in [n] : Y^{(i)} = Y^{*(i)}\}$ and noisy data $\{i \in [n] : Y^{(i)} \neq Y^{*(i)}\}$ by $\mathcal{C}$ and $\mathcal{N}$, respectively. Furthermore, the set of data $\{i \in \mathcal{C} \mid Y^{(i)} = 1\}$ are denoted as $\mathcal{C}_{+1}$, and $\{i \in \mathcal{C} \mid Y^{(i)} = -1\}$ are denoted as $\mathcal{C}_{-1}$. The same notation is applied to $\mathcal{N}$. The superscript $(i)$ denotes that the variable corresponds to the training data $i \in [n]$. For instance, we write $\mathbf{X}^{(i)} = (\mathbf{x}_1^{(i)}, \dots, \mathbf{x}_T^{(i)})^\top$ and use $\gamma_t^{(i)}$ to represent the token scores, which were defined in the previous section. Since the input tokens to the attention model are position invariant and the token ratios $\zeta_\mathcal{R}, \zeta_\mathcal{W}$ are constant, we use the same notation $\mathcal{R}, \mathcal{W}, \mathcal{I}$ across all samples. Although the size of $\mathcal{W}$ is fixed, each token is randomly correlated with some class; therefore, we used the notation $\mathcal{W}_{+1}^{(i)}$ and $\mathcal{W}_{-1}^{(i)}$ for each sample to clarify the class of the weak relevance. For simplicity, we make the next assumption on the signal.

**Assumption A.** *The class signal vectors hold the following:*

$$\|\boldsymbol{\mu}_{+1}\|_2 = \|\boldsymbol{\mu}_{-1}\|_2, \ \|\boldsymbol{\mu}_{+1}\|_\mathbf{\Sigma} = \|\boldsymbol{\mu}_{-1}\|_\mathbf{\Sigma}, \ \langle \boldsymbol{\mu}_{+1}, \boldsymbol{\mu}_{-1} \rangle = 0, \ \langle \boldsymbol{\mu}_{+1}, \boldsymbol{\mu}_{-1} \rangle_\mathbf{\Sigma} = 0,$$

*where $\|\boldsymbol{\mu}\|_\mathbf{\Sigma} = \sqrt{\boldsymbol{\mu}^\top \mathbf{\Sigma} \boldsymbol{\mu}}$ and $\langle \boldsymbol{\mu}_{+1}, \boldsymbol{\mu}_{-1} \rangle_\mathbf{\Sigma} = \boldsymbol{\mu}_{+1}^\top \mathbf{\Sigma} \boldsymbol{\mu}_{-1}$.*

In addition, for simplicity of notation, we denote the norm of the signals by $\|\boldsymbol{\mu}\|_2 := \|\boldsymbol{\mu}_{+1}\|_2 = \|\boldsymbol{\mu}_{-1}\|_2$ and $\|\boldsymbol{\mu}\|_\mathbf{\Sigma} := \|\boldsymbol{\mu}_{+1}\|_\mathbf{\Sigma} = \|\boldsymbol{\mu}_{-1}\|_\mathbf{\Sigma}$.

**Remark 1** (Weakly Relevant Token). Weakly relevant tokens represent tokens with weak signal strength and confusing class information, reflecting a more realistic scenario than a clean separation into relevant and irrelevant tokens. For instance, if the label noise derives from annotation errors, it is plausible that weakly relevant tokens aligning with the label noise exist.

### 3.4 GRADIENT-DESCENT TRAINING

The learnable parameters $(\mathbf{p}, \mathbf{W}, \boldsymbol{\nu})$ are trained to minimize the empirical risk objective:

$$\widehat{\mathcal{L}}(\mathbf{p}, \mathbf{W}, \boldsymbol{\nu}) = \frac{1}{n} \sum_{i \in [n]} \ell\left(Y^{(i)} \cdot f(\mathbf{X}^{(i)})\right), \ \ell(z) = \log(1 + \exp(-z)), \tag{3}$$

where $\ell : \mathbb{R} \to \mathbb{R}$ is a binary cross-entropy loss. In this paper, since our interest lies primarily in the token-selection mechanism, i.e., the inside softmax, we will discuss benign overfitting of $f$ under a fixed well-pretrained linear head $\boldsymbol{\nu} \propto \boldsymbol{\mu}_{+1} - \boldsymbol{\mu}_{-1}$. This setup is supported by the following facts.

**Lemma 3.1** (Informal). *Suppose that $\mathbf{p} = \mathbf{0}$. Then, the gradient descent direction of the expected risk at $\boldsymbol{\nu} = \mathbf{0}$ aligns with $\boldsymbol{\mu}_{+1} - \boldsymbol{\mu}_{-1}$.*

In the general multi-class case, the gradient direction of the expected risk aligns with the Equiangular Tight Frame (ETF) with class vectors (Papyan et al., 2020), which will be discussed in Section G in the appendix. The final layer fixed to ETF geometry during the training is often observed in the context of imbalanced data and transfer learning (Yang et al., 2022; Ali et al., 2024).

The remaining trainable parameters are $(\mathbf{p}, \mathbf{W})$; however, they essentially play the same role inside the softmax function, as shown in Lemma A.6. Thus, we only consider optimizing $\mathbf{p}$ inside softmax and fixing $\mathbf{W}$ throughout training. Let $\mathbf{W}$ be initialized as follows and fixed in the rest of the paper.

**Assumption B.** *The weight $\mathbf{W}$ is initialized with the orthogonal matrix: $\mathbf{W}\mathbf{W}^\top = \mathbf{W}^\top \mathbf{W} = \mathbf{I}$.*

Let $\mathbf{p}$ be initialized as $\mathbf{p}(0) = \mathbf{0}$; therefore, for any input, the model calculates the uniform attention $1/T$ at time step $\tau = 0$. The parameters are optimized by gradient descent with a step size $\alpha > 0$:

$$\mathbf{p}(\tau + 1) = \mathbf{p}(\tau) - \alpha \nabla_\mathbf{p} \widehat{\mathcal{L}}(\mathbf{p}(\tau)). \tag{4}$$

Let $\mathbf{p}(\tau)$ and $f_\tau$ be the $\mathbf{p}$ and $f$ after the $\tau$ gradient descent step, respectively. We also denote the Lipschitz constant of $\nabla_\mathbf{p} \widehat{\mathcal{L}}(\mathbf{p})$ by $\|\nabla_\mathbf{p} \widehat{\mathcal{L}}(\mathbf{p})\|_{\mathrm{Lip}}$, which represents the smoothness of the empirical loss function.

### 3.5 ASSUMPTION ON PARAMETERS

In this section, we first discuss the necessity of the assumptions on parameters and compare them with existing studies, followed by a list of the assumptions used in our work. In analyzing benign

overfitting, the balance between memorization and generalization is significant. Therefore, we impose assumptions on the balance between the dimensionality $\mathrm{Tr}(\mathbf{\Sigma})$ and the signal strength $\|\boldsymbol{\mu}\|_2$.

The over-parameterization assumptions (A1) and (A'1) are necessary for fitting label noise, and similar terms $n\|\boldsymbol{\mu}\|_2^2$ can be found in (Chatterji & Long, 2021; Frei et al., 2022; Kou et al., 2023). Assumptions (A2) and (A'2) are required for generalization, while $\mathrm{Tr}(\mathbf{\Sigma})^{1/4}$ in the right-hand side also appears in (Xu & Gu, 2023), and $n^{1/4}\mathrm{Tr}(\mathbf{\Sigma})^{1/4}$ is found in (Xu et al., 2024). For these assumptions on $\mathrm{Tr}(\mathbf{\Sigma})$ and $\|\boldsymbol{\mu}\|_2$, since this is the first analysis of benign overfitting in the attention architecture, the dependence on sequence length $T$ is newly introduced, and the difficulties unique to this setting, which will be discussed later in Section 4.3, involve stronger assumptions. Assumptions (A3) and (A'3) are also necessary for fitting label noise, and (A'3) is stronger for simplifying the conditions in Theorem 4.2. This assumption is non-vacuous because the value of $\rho$ can take in a wide range when $\mathrm{Tr}(\mathbf{\Sigma})$ and $\|\boldsymbol{\mu}\|_2$ are significantly larger than $T$ and $n$. For example, $n = \Omega(\mathrm{polylog}(\mathrm{Tr}(\mathbf{\Sigma})))$ is assumed in (Cao et al., 2022), which is based on a similar proof technique to ours. Assumptions (A5) and (A6) are placed to evaluate the class balance in the training data and the amount of noisy data, but in this paper, they are absorbed into the conditions of Eqs.9-13 and are not directly used.

Now, we state our assumptions in the following. Given a small failure probability $\delta > 0$ and a large enough universal constant $C$, we make the assumptions for each parameter as follows:

(A1) The covariance matrix $\mathbf{\Sigma}$ satisfies $\mathrm{Tr}(\mathbf{\Sigma}) \geq Cn\|\boldsymbol{\mu}\|_2^2$.

(A2) The signal strength satisfies $\|\boldsymbol{\mu}\|_2 \geq C\,\mathrm{Tr}(\mathbf{\Sigma})^{1/4}\sqrt{\log(Tn/\delta)}$.

(A3) The weak signal strength $\rho$ satisfies $C\sqrt{\log(Tn/\delta)}/\|\boldsymbol{\mu}\|_2 \leq \rho \leq 1/C$.

(A4) The step size $\alpha$ satisfies $\alpha \leq \min\{1/\|\nabla_{\mathbf{p}}\widehat{\mathcal{L}}(\mathbf{p})\|_{\mathrm{Lip}}, n/\mathrm{Tr}(\mathbf{\Sigma})\}/C$.

(A5) The number of training data $n$ satisfies $n \geq C\log(1/\delta)$.

(A6) The noise rate $\eta$ satisfies $\eta \leq 1/C$.

Furthermore, stronger versions of assumption (A1), (A2), (A3) are introduced to discuss the convergence of the gradient descent in Theorem 4.2.

(A'1) The covariance matrix $\mathbf{\Sigma}$ satisfies $\mathrm{Tr}(\mathbf{\Sigma}) \geq CT^2n^2\log(Tn/\delta)\|\boldsymbol{\mu}\|_2^2$.

(A'2) The signal strength satisfies $\|\boldsymbol{\mu}\|_2 \geq C(Tn)^{1/4}\mathrm{Tr}(\mathbf{\Sigma})^{1/4}\sqrt{\log(Tn/\delta)}$.

(A'3) The weak signal $\rho$ satisfies $CT\sqrt{\log(Tn/\delta)}/\|\boldsymbol{\mu}\|_2 \leq \rho \leq \min\left\{1/T, 1/\sqrt{Tn}\right\}/C$.

In addition, the equations derived from these assumptions, which will be used in our proofs, are presented in Lemma A.8 in the appendix.

# 4 MAIN RESULTS

In this section, we first demonstrate the existence of a benign overfitting solution in Section 4.1 and then discuss the convergence of gradient descent towards this benign overfitting solution in Section 4.2. Finally, in Section 4.3, we explain the unique difficulties posed by the attention architectures in analyzing benign overfitting and provide justification for the conditions used in the theorem and approach adopted in this work.

## 4.1 EXISTENCE OF BENIGN OVERFITTING

We first show the existence of the solution that exhibits benign overfitting. This result is useful for explaining the key idea that both signal learning and noise memorization are necessary for benign overfitting in token selection. Here, note that the model simply learning the signal vectors does not overfit to noisy data.

**Theorem 4.1** (Existence of benign overfitting solution). *Suppose that the assumptions (A1)-(A6) hold. Let the linear head $\boldsymbol{\nu} \propto \boldsymbol{\mu}_{+1} - \boldsymbol{\mu}_{-1}$ be fixed during the training, and $\beta_j = \Theta(1/n) > 0, j \in \mathcal{N}$*

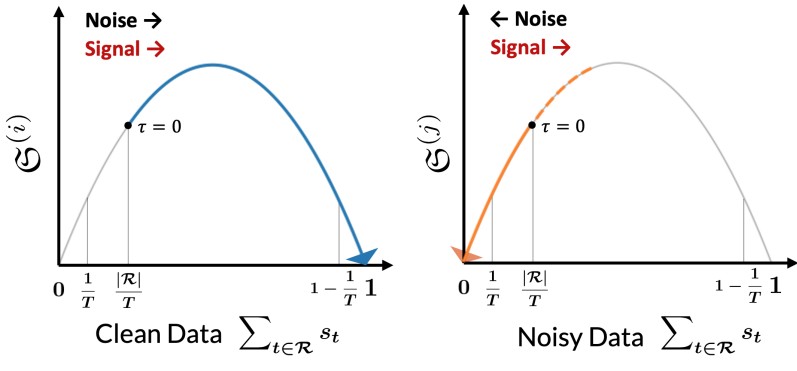

Figure 2: Illustration for the training dynamics of the probability assigned to relevant tokens in clean data $i \in \mathcal{C}$ and noisy data $j \in \mathcal{N}$. The y-axis shows $\mathfrak{S} := \left( \sum_{t \in \mathcal{R}} s^{(i)}(\tau)_t \right) \left( 1 - \sum_{t \in \mathcal{R}} s^{(i)}(\tau)_t \right)$ in Definition 4.1, which is essential for learning the signal.

*be some constants. For any weakly relevant token $u_j \in \mathcal{W}_{Y^{(j)}}^{(j)}$, which aligns with label noise and sufficiently large constant $R > 0$,*

$$\mathbf{p} = R \cdot \mathbf{W} \left( \boldsymbol{\mu}_{+1} + \boldsymbol{\mu}_{-1} + \sum_{j \in \mathcal{N}} \beta_j \boldsymbol{\epsilon}_{u_j}^{(j)} \right)$$

*is a benign overfitting solution with probability at least $1 - \delta$. In other words, we have*

1. *(Overfitting) The model $f$ overfits to the training data $\mathcal{S}$ with label noise:*
$$\forall i \in [n], \ f(\mathbf{X}^{(i)}) = Y^{(i)}, \tag{5}$$

2. *(Generalization) The generalization loss of $f$ is bounded as:*
$$\Pr_{(\mathbf{X}, Y^*) \sim P^*} [\mathrm{sign}(f(\mathbf{X})) \neq Y^*] < \delta. \tag{6}$$

This theorem reflects the intuition that 1) selecting class-relevant tokens is desirable for generalization and 2) selecting alternative tokens that align with the label noise is necessary for fitting noisy data. The proof provided in Section C demonstrates such token selection is performed.

## 4.2 CONVERGENCE TO BENIGN OVERFITTING

While we have seen the existence of a benign overfitting solution, we will further analyze in this section whether gradient descent converges to such a solution and what conditions are necessary for benign overfitting. Theorem 4.1 shows that both the signal required for generalization and the noise memorization for fitting label noise are significant, and their balance is essential for benign overfitting. In practice, the model is trained by gradient descent based on the empirical loss function; therefore, we must handle the influence of label noise in the signal learning, and noise memorization occurs for all tokens besides the desirable token $u_j \in \mathcal{W}_{Y^{(j)}}^{(j)}, j \in \mathcal{N}$, making it challenging to draw simple conclusions for optimization.

Before presenting the statement, we introduce the following value, which is crucial for understanding the gradient descent behavior in the attention architecture.

**Definition 4.1** (Uncertainty in probability of picking relevant token). For any $i \in [n]$ and time step $\tau \geq 0$, we define $\mathfrak{S}^{(i)}(\tau) \in [0, 1/4]$ as:

$$\mathfrak{S}^{(i)}(\tau) := \left( \sum_{t \in \mathcal{R}} s^{(i)}(\tau)_t \right) \left( 1 - \sum_{t \in \mathcal{R}} s^{(i)}(\tau)_t \right), \tag{7}$$

where recall that $\mathbf{s}^{(i)}(\tau) \in \mathbb{R}^T$ be a shorthand for the softmax probability $\mathbb{S}(\mathbf{X}^{(i)} \mathbf{W}^\top \mathbf{p}(\tau))$.

This value can be interpreted as the variance of the Bernoulli distribution, where the parameter is the attention probability assigned to the relevant token of training sample $i \in [n]$ at time step $\tau$. This behavior is crucial in the analysis of signal learning. Specifically, in learning signal vectors, the balance between the contributions of $\mathfrak{S}(\tau)$ from clean data and from noisy data is significant and determines the progress of signal learning at time step $\tau$. Figure 2 illustrates the training dynamics of this quantity for clean and noisy data. As shown later, the softmax probability converges to either 0 or 1 for any token under the parameter assumptions. Therefore, $\mathfrak{S}(\tau)$ will eventually converge to 0 as illustrated in Figure 2.

By dividing cases for the behavior of $\mathfrak{S}$, we obtain the following results for the convergence.

**Theorem 4.2** (Convergence of benign overfitting solution). *Suppose that the assumptions (A'1)-(A'3), (A4)-(A6) hold, and the norm of the fixed linear head $\boldsymbol{\nu} \propto \boldsymbol{\mu}_{+1} - \boldsymbol{\mu}_{-1}$ scales as $\Theta(1/\|\boldsymbol{\mu}\|_2)$. With probability of at least $1 - \delta$, there exists a sufficiently large time step $T_0$, and for all time step after it: $\tau \geq T_0$, the model overfits to the training data $\mathcal{S}$ with label noise:*

$$\forall i \in [n], \; f_\tau(\mathbf{X}^{(i)}) = Y^{(i)}, \tag{8}$$

*At this time, we can further discuss the generalization error of the model if the following conditions on the training trajectory are satisfied:*

*1. For any class, the time accumulation of $\mathfrak{S}$ summed up in the clean data dominates the accumulation of each $s_t^{(i)}(1 - s_t^{(i)})$:*

$$\min_{c \in \{\pm 1\}} \left\{ \sum_{i \in \mathcal{C}_c} \sum_{0 \leq \tau' \leq \tau} \mathfrak{S}^{(i)}(\tau') \right\} > C_1 \max_{i \in [n], t \in [T]} \left\{ \sum_{0 \leq \tau' \leq \tau} s^{(i)}(\tau')_t \left(1 - s^{(i)}(\tau')_t\right) \right\}. \tag{9}$$

*2. The time accumulation of $\mathfrak{S}$ summed up in the clean data is balanced among labels:*

$$\frac{1}{C_2} \sum_{i \in \mathcal{C}_{-1}} \sum_{0 \leq \tau' \leq \tau} \mathfrak{S}^{(i)}(\tau') < \sum_{i \in \mathcal{C}_{+1}} \sum_{0 \leq \tau' \leq \tau} \mathfrak{S}^{(i)}(\tau') < C_2 \sum_{i \in \mathcal{C}_{-1}} \sum_{0 \leq \tau' \leq \tau} \mathfrak{S}^{(i)}(\tau'), \tag{10}$$

*Here $C_1, C_2 > 0$ are some absolute constants. Then, we have*

*1. (Benign) For any class $c \in \{\pm 1\}$, if the training trajectory satisfies*

$$\sum_{i \in \mathcal{C}_c} \sum_{0 \leq \tau' \leq \tau} \mathfrak{S}^{(i)}(\tau') > C_3 \sum_{j \in \mathcal{N}_{-c}} \sum_{0 \leq \tau' \leq \tau} \mathfrak{S}^{(j)}(\tau'), \tag{11}$$

*for some constant $C_3 > 1$, then with probability at least $1 - \delta$, the weight $\mathbf{p}(\tau)$ is a benign overfitting solution:*

$$\Pr_{(\mathbf{X}, Y^*) \sim P^*} \left[\text{sign}\left(f_\tau(\mathbf{X})\right) \neq Y^*\right] < \epsilon. \tag{12}$$

*2. (Harmful) For any class $c \in \{\pm 1\}$, if the training trajectory satisfies*

$$\sum_{i \in \mathcal{C}_c} \sum_{0 \leq \tau' \leq \tau} \mathfrak{S}^{(i)}(\tau') < C_4 \sum_{j \in \mathcal{N}_{-c}} \sum_{0 \leq \tau' \leq \tau} \mathfrak{S}^{(j)}(\tau'), \tag{13}$$

*for some constant $0 < C_4 < 1$, then the weight $\mathbf{p}(\tau)$ is a not-benign overfitting solution:*

$$\Pr_{(\mathbf{X}, Y^*) \sim P^*} \left[\text{sign}\left(f_\tau(\mathbf{X})\right) \neq Y^*\right] \geq \Theta(1). \tag{14}$$

This theorem shows that under the assumptions in Section 3.5, the model overfits the training data with gradient descent, and it further reduces the discussion of generalization ability to the accumulation of $\mathfrak{S}$ during the training. Eqs. 11 and 13 are regarded as conditions on the balance between the contributions of clean and noisy data. The reason for requiring such conditions on $\mathfrak{S}$ will be discussed later in the second paragraph of Section 4.3. The condition Eq.9 requires that the total sum of the cumulative $\mathfrak{S}$ over the clean data exceeds the accumulation of $\mathfrak{S}$ for each data point. The condition Eq.10 concerns the balance of the cumulative $\mathfrak{S}$ among classes, and in the existing analysis not involving softmax probability, it is straightforward to verify that the number of samples for each class in the training data is equal ignoring constants with high probability. We will provide the experiments that discuss the validity of Eqs.9 and 10 in Section I. The proof sketch for the main theorems is provided in Section A.2, while the proof of Theorem 4.2 is detailed in Section E.

### 4.3 DIFFICULTY SPECIFIC TO ATTENTION ANALYSIS

In this section, we compare our work with the existing analysis of benign overfitting (Chatterji & Long, 2021; Frei et al., 2022; Xu & Gu, 2023) tracking gradient descent dynamics, which is the approach we took in this work, and highlight the unique difficulties inherent in the attention architecture. The first two paragraphs focus on these difficulties, and in the final part, we explain the limitations of the other approach.

**Presence of local minima.** The first difficulty comes from the presence of local minima that do not fit all the training data. In the previous settings, the gradient of the loss function (or the margin increase) at each time step can be bounded below by some form of training error; therefore, proving convergence of the gradient implies overfitting to the training set. In this paper setting, the gradient of the empirical loss function is given as follows:

$$\nabla_{\mathbf{p}} \widehat{\mathcal{L}}(\mathbf{p}) = \frac{1}{n} \sum_{i=1}^{n} \ell' \left( Y^{(i)} \cdot f(\mathbf{X}^{(i)}) \right) \cdot Y^{(i)} \cdot \left( \sum_{t \in [T]} s_t^{(i)} \left( \gamma_t^{(i)} - \sum_{u \in [T]} s_u^{(i)} \gamma_u^{(i)} \right) \mathbf{W} \mathbf{x}_t^{(i)} \right), \quad (15)$$

where $\ell'$ is the derivative of the binary cross-entropy loss, given in Eq.3. Since the learning token selection does not change the scale of output, the $\ell'$ term does not converge to zero, and the gradient becomes zero only when the attention probabilities converge to 1 or 0, as will be shown in Lemma D.3. Therefore, undesirable tokens can be picked even when the gradient becomes zero, and we have to track the dynamics of token probability during training. In the appendix, we not only analyzed the convergence of the training loss gradient but also tracked the probabilities assigned to each token, identifying which token converges to 1 and is thus selected by the model.

**Diminishing parameter updates due to softmax.** The second difficulty is that the gradient descent updates are involved with the softmax probabilities. The relationship between the contributions from clean and noisy data, as well as the updates for signals and noises, cannot be evaluated independently of the current time step, as the gradient descent updates depend on the value of $s_t(\tau)(1 - s_t(\tau))$. As shown in Figure 2, the closer the probability of selecting the desired token approaches 1, the smaller the updates will be for selecting that token. This *diminishing growth of model parameters* complicates the convergence analysis of benign overfitting in the attention, compared to existing work on linear classifiers and two-layer networks. For example, while relevant tokens in the clean data are quickly picked, driving $\mathfrak{S}(\tau)$ toward 0, if $\mathfrak{S}(\tau)$ is still large in noisy data (i.e., not yet near both ends in Figure 2), the contribution of noisy data to the learning can become more significant than that of clean data even when the number of noisy data is much smaller. To address this, we introduced a metric in Definition 4.1, which allows us to present scenarios that lead to either benign or harmful overfitting, as demonstrated in Eqs. 11 and 13.

**Obstacles in approach from max-margin problem** One potential approach is based on the convergence of max-margin solutions for token-separation (Tarzanagh et al., 2023b; Vasudeva et al., 2024), similar to the existing work using KKT conditions. However, this approach is challenging for two reasons: 1) there is no guarantee of global convergence of the max-margin problem, and 2) there is a mismatch between the max-margin solution and the local solution of empirical loss due to the difference in token scores. Please refer to Section H.3 in the appendix for further details.

## 5 EXPERIMENTS

In this section, we conduct synthetic experiments based on the data model in Definition 3.1 to support our analysis. The code used for the experiments is available on GitHub [1]. For more detailed experiments varying $d$ and $\|\boldsymbol{\mu}\|_2$, as well as experiments on real-world image and text datasets containing label noise, please refer to Section I in the appendix.

We train the same model $f$ as in Eq.2 with gradient descent, using the same data model as Definition 3.1 with the covariance matrix $\boldsymbol{\Sigma} = \mathbf{I}$. Specifically, we consider the setting with $n = 20$, $T = 8$, $\eta = 0.2$, $\rho = 0.2$ and $\alpha =$1e$-4$, changing the value of the dimension $d = \mathrm{Tr}(\boldsymbol{\Sigma})$ and the signal size $\|\boldsymbol{\mu}\|_2$. For simplicity, we set $|\mathcal{R}| = 1$ and $|\mathcal{W}_{+1}| = |\mathcal{W}_{-1}| = 1$.

---

[1] https://github.com/repository-name

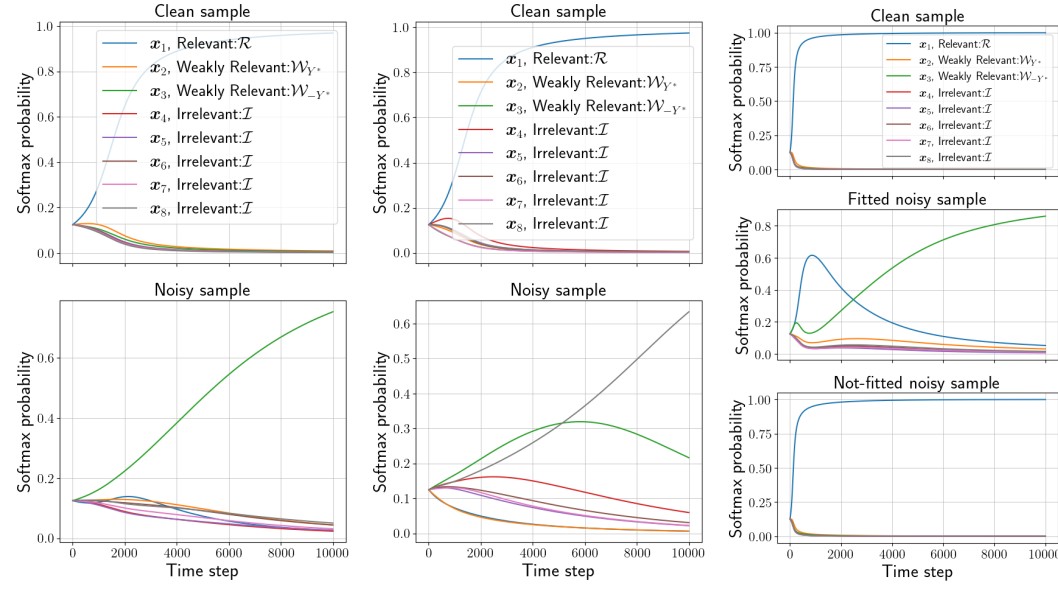

(a) Balanced setting (**satisfying** assumption): $d = 2000$, $\|\boldsymbol{\mu}\|_2 = 20$. Final training accuracy is $\mathbf{1.0}$ and test accuracy is $\mathbf{1.0}$.

(b) Large noise setting (not satisfying assumption): $d = 4500$, $\|\boldsymbol{\mu}\|_2 = 5$. Final training accuracy is $\mathbf{1.0}$ and test accuracy is $\mathbf{0.91}$.

(c) Large signal setting (not satisfying assumption): $d = 1000$, $\|\boldsymbol{\mu}\|_2 = 80$. Final training accuracy is $\mathbf{0.9}$ and test accuracy is $\mathbf{1.0}$.

Figure 3: Dynamics of softmax probability. The top represents a clean sample, while the bottom represents a noisy sample for each column. From left to right, the figures correspond to the cases of benign overfitting, harmful overfitting, and not overfitting.

Figure 3 shows the dynamics of softmax probabilities for clean and noisy training samples from the initial value $\mathbf{p}(0) = \mathbf{0}$. The left figure shows the case where the balance between signal and noise satisfies the theorem assumptions (see assumptions (A1) and (A2)), and benign overfitting is achieved. In this figure, selecting the weakly relevant token $u \in \mathcal{W}_{-Y^*(j)}$ for the noisy data $j \in \mathcal{N}$ aligns with our analysis. In the middle, where the $d$ is larger compared to $\|\boldsymbol{\mu}\|_2$, memorization becomes dominant. The bottom figure shows that weakly relevant tokens are not always selected for noisy data; instead, irrelevant tokens that fit the label noise are picked. This model can fit the training data with noise components, which hinders signal learning and reduces generalization ability. In the right figure, where signal norm $\|\boldsymbol{\mu}\|_2$ is large, fitting the noisy data becomes challenging. The middle of Figure 3c shows that the model successfully selects weakly relevant tokens aligning with the label noise, but in the bottom figure, some noisy data are influenced toward selecting relevant tokens that should not be chosen for fitting noisy data. We can see that the balance between the dimension and the signal norm, as well as a low label noise, is significant for achieving low train and test loss.

## 6 CONCLUSION

In this paper, we analyzed benign overfitting in the token selection of the attention architecture and further supported the analysis with synthetic experiments. As a natural next step, a more detailed technical analysis of the attention probability behavior during training, which appeared as conditions in our results, can be considered. Additionally, extending the attention analysis to next-token prediction or a full self-attention model is also an interesting research direction.

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

# A PRELIMINARIES

## A.1 NOTATION

We first list the notations used in this work in Table 1.

Furthermore, the basic computations are presented here for convenience. The gradient $\nabla_{\mathbf{P}} \widehat{\mathcal{L}}(\mathbf{p})$ used in the training can be explicitly computed as follows. Since the derivative of the softmax function is given by

$$\nabla_{\mathbf{v}} \mathbb{S}(\mathbf{v}) = \mathrm{diag}(\mathbb{S}(\mathbf{v})) - \mathbb{S}(\mathbf{v}) \mathbb{S}(\mathbf{v})^{\top},$$

Table 1: Notations used in this work.

| | |
|---|---|
| $\mathbf{X}$ | Sequence of input tokens, $\mathbf{X} = (\mathbf{x}_1, \ldots, \mathbf{x}_T)^\top$ |
| $Y$ | Noise corrupted label |
| $Y^*$ | True label |
| $T$ | Length of input tokens |
| $d$ | Dimension of token embedding |
| $n$ | Number of training data |
| $\mathbf{s}^{(i)}(\tau)$ | Probability vector for $\mathbf{X}^{(i)}$ at $\tau$-th step, $\mathbb{S}(\mathbf{X}^{(i)}\mathbf{W}^\top\mathbf{p}(\tau))$ |
| $\gamma_t^{(i)}$ | Token score, $\mathbf{x}_t^{(i)\top}\boldsymbol{\nu}$ |
| $\boldsymbol{\mu}_{+1}, \boldsymbol{\mu}_{-1}$ | Signal vectors for class 1 and $-1$, respectively |
| $\lambda_{+1}, \lambda_{-1}$ | Coefficients for $\boldsymbol{\mu}_{+1}$ and $\boldsymbol{\mu}_{-1}$, respectively, in the parameter $\mathbf{p}$ |
| $\boldsymbol{\Sigma}$ | Covariance matrix of noise vector |
| $\boldsymbol{\epsilon}_t^{(i)}$ | Noise component in $\mathbf{x}_t^{(i)}$ |
| $\rho_t^{(i)}$ | Coefficient for $\boldsymbol{\epsilon}_t^{(i)}$ in the parameter $\mathbf{p}$ |
| $\mathcal{R}$ | Set of relevant token: $\mathbf{x}_t = \boldsymbol{\mu}_{Y^*} + \boldsymbol{\epsilon}_t, t \in \mathcal{R}$ |
| $\mathcal{W}$ | Set of weakly relevant token: $\mathbf{x}_u = \rho\boldsymbol{\mu}_{w_u} + \boldsymbol{\epsilon}_u, u \in \mathcal{W}, w_u \sim$ Unif$\{\pm 1\}$ |
| $\mathcal{W}_{+1}, \mathcal{W}_{-1}$ | Set of weak relevant token with specific label, $\{u \in \mathcal{W} \mid w_u = +1\}$, $\{u \in \mathcal{W} \mid w_u = -1\}$. We also use the notation $\mathcal{W}_{+1}^{(i)}$ and $\mathcal{W}_{-1}^{(i)}$ for the input $\mathbf{X}^{(i)}$. |
| $\mathcal{I}$ | Set of irrelevant token: $\mathbf{x}_v = \boldsymbol{\epsilon}_v, v \in \mathcal{I}$ |
| $\zeta_{\mathcal{R}}, \zeta_{\mathcal{W}}$ | Ratio of relevant and weakly relevant tokens: $|\mathcal{R}|/T, |\mathcal{W}|/T$ |
| $\mathfrak{S}^{(i)}(\tau)$ | Value defined as $\left(\sum_{t \in \mathcal{R}} s^{(i)}(\tau)_t\right)\left(1 - \sum_{t \in \mathcal{R}} s^{(i)}(\tau)_t\right)$ |
| $\mathcal{C}$ | Set of clean data: $\{i \in [n] : Y^{(i)} = Y^{*(i)}\}$ |
| $\mathcal{C}_{+1}, \mathcal{C}_{-1}$ | Set of clean data with label 1 and $-1$, respectively |
| $\mathcal{N}$ | Set of noisy data: $\{i \in [n] : Y^{(i)} \neq Y^{*(i)}\}$ |
| $\mathcal{N}_{+1}, \mathcal{N}_{-1}$ | Set of noisy data with label 1 and $-1$, respectively |
| $\alpha$ | Learning rate |
| $\eta$ | Level of label noise |
| $t$ | Index used mainly for $[T]$ or $\mathcal{R}$ |
| $u$ | Index used mainly for $\mathcal{W}$ |
| $v$ | Index used mainly for $\mathcal{I}$ |
| $i$ | Index used mainly for $[n]$ or $\mathcal{C}$ |
| $j$ | Index used mainly for $\mathcal{N}$ |

Table 2: Comparison of model architecture, analysis approaches, data model, and mention of a multiclass case between our study and existing work for classification on linearly separable data distribution. Here, "GM" stands for Gaussian mixture. Some studies consider a broader setting of sub-Gaussian mixtures, but for simplicity, we refer to them uniformly as Gaussian mixtures here.

| Paper | Model Architecture | Approach | Data Model | Mention of Multiclass |
|---|---|---|---|---|
| Cao et al. (2021) | Linear | Max-margin | GM | No |
| Wang et al. (2021) | Linear | Max-margin | GM & Multinomial Logit | Yes |
| Chatterji & Long (2021) | Linear | Tracking dynamics | GM | No |
| Frei et al. (2022) | Two-layer NN | Tracking dynamics | GM | No |
| Cao et al. (2022) | Two-layer CNN | Tracking dynamics | Concatenated Feature | No |
| Xu & Gu (2023) | Two-layer NN | Tracking dynamics | GM | No |
| Kou et al. (2023) | Two-layer CNN | Tracking dynamics | Concatenated Feature | No |
| Frei et al. (2023a) | Two-layer NN | Max-margin | GM & Discriminative | No |
| Zhu et al. (2023) | DNN | NTK | GM | No |
| **Ours** | One-layer Attention | Tracking dynamics | GM on Sequence | Yes |

where $\mathrm{diag}(\mathbf{v}) \in \mathbb{R}^{T \times T}$ denotes the diagonal matrix whose $(i,i)$-th entry equals to $v_i$. Using the denominator layout, we have

$$\nabla_{\mathbf{p}} \widehat{\mathcal{L}}(\mathbf{p}) = \frac{1}{n} \sum_{i=1}^{n} \ell_i' \cdot Y^{(i)} \cdot \nabla_{\mathbf{p}} f(\mathbf{X}^{(i)}) \tag{16}$$

$$= \frac{1}{n} \sum_{i=1}^{n} \ell_i' \cdot Y^{(i)} \cdot \mathbf{W} \mathbf{X}^{(i)\top} \left( \mathrm{diag}(\mathbb{S}(\mathbf{X}^{(i)} \mathbf{W}^\top \mathbf{p})) - \mathbb{S}(\mathbf{X}^{(i)} \mathbf{W}^\top \mathbf{p}) \mathbb{S}(\mathbf{X}^{(i)} \mathbf{W}^\top \mathbf{p})^\top \right) \mathbf{X}^{(i)} \boldsymbol{\nu} \tag{17}$$

$$= \frac{1}{n} \sum_{i=1}^{n} \ell_i' \cdot Y^{(i)} \cdot \left( \sum_{t \in [T]} s_t^{(i)} \left( \gamma_t^{(i)} - \sum_{u \in [T]} s_u^{(i)} \gamma_u^{(i)} \right) \mathbf{W} \mathbf{x}_t^{(i)} \right), \tag{18}$$

where $\ell_i'$ is abbreviation for $\ell'(Y^{(i)} \cdot \boldsymbol{\nu}^\top \mathbf{X}^{(i)\top} \mathbb{S}(\mathbf{X}^{(i)} \mathbf{W}^\top \mathbf{p}))$.

## A.2 PROOF SKETCH

In this section, we briefly introduce the proof sketch of the main theorem along with the key lemmas to provide the road maps in the appendix. Table 2 provides a concise comparison of analysis approaches with the existing benign overfitting work for classification. We hope this will help further understanding through comparison with other papers.

The whole proof of Theorem 4.1 is provided in Section C, and the proof of Theorem 4.2 is completed in Section E. We first show that the following events occur simultaneously with high probability under the assumptions in Section 3.5.

**Lemma A.1.** *Suppose that the assumptions (A1)-(A6) hold. There exists some constant $c_1, c_2 > 0$ and $C' > 0$, which depends on $C$, such that for all $c' > 0$, the following hold simultaneously with probability at least $1 - \delta$ over the realization of training data $\mathcal{S}$:*

*(i) (Noise Norm) For all $i \in [n], t \in [T]$, we have*

$$\left(1 - 1/\mathrm{Tr}(\boldsymbol{\Sigma}) - 1/C'\right) \sqrt{\mathrm{Tr}(\boldsymbol{\Sigma})} \le \|\boldsymbol{\epsilon}_t^{(i)}\|_2 \le \left(1 + 1/C'\right) \sqrt{\mathrm{Tr}(\boldsymbol{\Sigma})}. \tag{19}$$

*(ii) (Noise Inner-product) For any $i, j \in [n], t, u \in [T]$ such that $(i,t) \ne (j,u)$, we have*

$$|\langle \boldsymbol{\epsilon}_t^{(i)}, \boldsymbol{\epsilon}_u^{(j)} \rangle| < c_1 \sqrt{\mathrm{Tr}(\boldsymbol{\Sigma})} \log(Tn/\delta), \tag{20}$$

*(iii) (Signal-Noise Inner-product) For all $i \in [n], t \in [T], c \in \{\pm 1\}$, we have*

$$|\langle \boldsymbol{\epsilon}_t^{(i)}, \boldsymbol{\mu}_c \rangle| < c_2 \|\boldsymbol{\mu}\|_2 \sqrt{\log(Tn/\delta)}. \tag{21}$$

*(iv) Regarding the true label and the label noise, we have*

$$((1-\eta)/2 - c')n \leq |\mathcal{C}_{+1}|, |\mathcal{C}_{-1}| \leq ((1-\eta)/2 + c')n, \tag{22}$$

$$(\eta/2 - c')n \leq |\mathcal{N}_{+1}|, |\mathcal{N}_{-1}| \leq (\eta/2 + c')n. \tag{23}$$

**Definition A.1.** We denote by "good run" the trial that the events from (i) to (iv) occur.

In the proof of the main theorems, we discuss the properties that occur with high probability by conditioning on these events. Lemma A.1 implies that a good run occurs with the probability at least $1 - \delta$ over the realization of training data $\mathcal{S}$. The proof of Lemma A.1 will be provided in Section B. The proof of Theorem 4.1 is based on these concentration inequalities and detailed evaluations of the softmax probabilities. For the complete proof, please refer to Section C.

The convergence analysis, due to the difficulties stated in Section 4.3, follows an approach of tracking the dynamics of the signal and noise components in the gradient method rather than setting desired converged weights and evaluating the alignment with them. Specifically, let $\mathbf{p}(\tau)$ be a gradient iteration at $\tau$-th time step; then, there exists unique coefficients such that

$$\mathbf{p}(\tau) = \lambda_{+1}(\tau)\mathbf{W}\boldsymbol{\mu}_{+1} + \lambda_{-1}(\tau)\mathbf{W}\boldsymbol{\mu}_{-1} + \sum_{i \in [n]} \sum_{t \in [T]} \rho_{i,t}(\tau)\mathbf{W}\boldsymbol{\epsilon}_t^{(i)}. \tag{24}$$

We will provide the complete form of this statement in Lemma D.1 and also refer to the recurrence equations that the coefficients $\lambda_{+1}, \lambda_{-1}, \{\rho_{i,t}\}_{i \in [n], t \in [T]}$ should satisfy. Our discussion for the convergence is based on the following lemma, which leverages the smoothness of the one-layer attention network:

**Lemma A.2.** *Suppose the step size $\alpha$ satisfies the assumption (A4). Then, there exists the token index $t_i^* \in [T]$ for each $i \in [n]$, and we have*

$$\lim_{\tau \to \infty} s_{t_i^*}^{(i)}(\tau) = 1, \quad and \quad \lim_{\tau \to \infty} s_t^{(i)}(\tau) = 0, \tag{25}$$

*for all $i \in [n], t \in [T] \setminus \{t_i^*\}$.*

We will further confirm that the desired token is picked at the converted weights. The following two lemmas describe the relationship among the attention probabilities that hold at any time step for both clean and noisy data.

**Lemma A.3.** *Suppose that the assumptions (A'1)-(A'3), (A4)-(A6) hold, and the norm of the linear head $\boldsymbol{\nu} \propto \boldsymbol{\mu}_{+1} - \boldsymbol{\mu}_{-1}$ scales as $\Theta(1/\|\boldsymbol{\mu}\|_2)$. For any clean data $i \in \mathcal{C}$, on a good run, for all time step $\tau \geq 0$ and all weakly relevant or irrelevant token $u \in \mathcal{W} \cup \mathcal{I}$, we have*

$$s^{(i)}(\tau)_u \leq \max_{t \in \mathcal{R}} \left\{ s^{(i)}(\tau)_t \right\}. \tag{26}$$

**Lemma A.4.** *Suppose that the assumptions (A'1)-(A'3), (A4)-(A6) hold, and the norm of the linear head $\boldsymbol{\nu} \propto \boldsymbol{\mu}_{+1} - \boldsymbol{\mu}_{-1}$ scales as $\Theta(1/\|\boldsymbol{\mu}\|_2)$. For any noisy data $j \in \mathcal{N}$, on a good run, for all time step $\tau \geq 0$ and all tokens except for the weakly relevant tokens that align with the label noise: $t \in [T] \setminus \mathcal{W}_{Y^{(j)}}^{(j)}$, we have*

$$s^{(j)}(\tau)_t \leq \max_{u \in \mathcal{W}_{Y^{(j)}}^{(j)}} \left\{ s^{(j)}(\tau)_u \right\}. \tag{27}$$

Using these lemmas, we can demonstrate overfitting in Theorem 4.2. Please refer to Section E.1 for the complete proof.

Next, we will discuss the generalization ability of this overfitted solution, which is closely related to the sign and order of the coefficients of the learned signal components, i.e., $\lambda_c, c \in \{\pm 1\}$ in Eq.24. The following is a critical lemma that shows how the conditions in the statement of Theorem 4.2 affect $\lambda, \rho$ in the decomposition given by Eq.24.

**Lemma A.5** (Simplified Version). *Suppose that the assumptions (A'1)-(A'3), (A4)-(A6) and the conditions Eq.9 and 10 hold. If the training trajectory satisfies Eq.11, then we have*

$$\lambda_{+1}(\tau) \sim \lambda_{-1}(\tau) \gtrsim \frac{1}{\sqrt{T}n} \sum_{i \in [n], t \in [T]} |\rho_{i,t}(\tau)|. \tag{28}$$

This lemma states that the signal coefficients are balanced among the classes in terms of order and that the learning process is not primarily a memorization of noise. If Eq.13 holds, we have a similar result where the signal coefficient becomes negative. For its proof and impact on the generalization ability, please refer to Section E as a complete proof of generalization part of Theorem 4.2.

### A.3 PRELIMINARY LEMMAS

Firstly, we introduce the lemmas necessary in preparation for the main proof.

The following lemma indicates that the dynamics of $\mathbf{W}$ can be described by the dynamics of $\mathbf{p}$. It justifies optimizing $\mathbf{p}$ alone while keeping $\mathbf{W}$ fixed throughout training. This is also intuitively understood from the fact that the gradient update of $\mathbf{W}$ always results in a rank-1 matrix. Since some modifications have been made to the original form, we will provide the statement with proof.

**Lemma A.6** (Rephrased from (Tarzanagh et al. (2023b), Lemma 1)). *Fix the linear head $\boldsymbol{\nu} \in \mathbb{R}^d$ throughout training. On the same training data $(\mathbf{X}^{(i)}, Y^{(i)})_{i=1}^n$, we define*

$$\widehat{\mathcal{L}}_{\mathbf{W}}(\mathbf{W}) \coloneqq \frac{1}{n} \sum_{i \in [n]} \ell \left( Y^{(i)} \cdot \boldsymbol{\nu}^\top \mathbf{X}^{(i)\top} \mathbb{S}(\mathbf{X}^{(i)} \mathbf{W}^\top \mathbf{p}_0) \right),$$

$$\widehat{\mathcal{L}}_{\mathbf{p}}(\mathbf{p}) \coloneqq \frac{1}{n} \sum_{i \in [n]} \ell \left( Y^{(i)} \cdot \boldsymbol{\nu}^\top \mathbf{X}^{(i)\top} \mathbb{S} \left( \mathbf{X}^{(i)} \mathbf{W}_0^\top \mathbf{p} \right) \right),$$

*where $\mathbf{p}_0 \in \mathbb{R}^d$ and $\mathbf{W}_0 \in \mathbb{R}^{d \times d}$ are fixed vector and matrix, respectively. Here, we assume that $\mathbf{W}_0$ is an orthogonal matrix, following assumption B. Consider the gradient descent iterations on $\mathbf{W}$ and $\mathbf{p}$ with initial values $\mathbf{W}(0)$ and $\mathbf{p}(0) = \mathbf{W}_0 \mathbf{W}(0)^\top \mathbf{p}_0$ and step sizes $\alpha$ and $\alpha \|\mathbf{p}\|_2^2$, respectively:*

$$\mathbf{W}(\tau + 1) = \mathbf{W}(\tau) - \alpha \nabla \widehat{\mathcal{L}}_{\mathbf{W}}(\mathbf{W}(\tau)),$$

$$\mathbf{p}(\tau + 1) = \mathbf{p}(\tau) - \alpha \|\mathbf{p}_0\|_2^2 \nabla \widehat{\mathcal{L}}_{\mathbf{p}}(\mathbf{p}(\tau)).$$

*Then, we have that $\mathbf{W}(\tau)^\top \mathbf{p}_0 = \mathbf{W}_0^\top \mathbf{p}(\tau)$ for all $\tau \geq 0$.*

*Proof.* We will proceed with induction. At time step 0, the claim holds from the definition of $\mathbf{W}_0$ and $\mathbf{p}_0$. Suppose that the claim at time step $\tau$ holds. Since we have,

$$\nabla \widehat{\mathcal{L}}_{\mathbf{W}}(\mathbf{W}(\tau)) = \frac{1}{n} \sum_{i=1}^n \ell_i' \cdot Y^{(i)} \cdot \mathbf{p}_0 \boldsymbol{\nu}^\top \mathbf{X}^{(i)\top} \nabla \mathbb{S} \left( \mathbf{X}^{(i)} \mathbf{W}(\tau)^\top \mathbf{p}_0 \right) \mathbf{X}^{(i)}, \tag{29}$$

$$\nabla \widehat{\mathcal{L}}_{\mathbf{p}}(\mathbf{p}(\tau)) = \frac{1}{n} \sum_{i=1}^n \ell_i' \cdot Y^{(i)} \cdot \mathbf{W}_0 \mathbf{X}^{(i)\top} \nabla \mathbb{S} \left( \mathbf{X}^{(i)} \mathbf{W}_0^\top \mathbf{p}(\tau) \right) \mathbf{X}^{(i)} \boldsymbol{\nu}, \tag{30}$$

we obtain $\nabla \widehat{\mathcal{L}}_{\mathbf{W}}(\mathbf{W}(\tau))^\top = \mathbf{W}_0^\top \nabla \widehat{\mathcal{L}}_{\mathbf{p}}(\mathbf{p}(\tau)) \mathbf{p}_0^\top$ using the induction hypothesis and the fact that $\mathbf{W}_0$ is an orthogonal matrix. Therefore, we have

$$\mathbf{W}(\tau + 1)^\top \mathbf{p}_0 = \mathbf{W}(\tau)^\top \mathbf{p}_0 - \alpha \nabla \widehat{\mathcal{L}}_{\mathbf{W}}(\mathbf{W}(\tau))^\top \mathbf{p}_0 \tag{31}$$

$$= \mathbf{W}_0^\top \mathbf{p}(\tau) - \alpha \|\mathbf{p}_0\|_2^2 \cdot \mathbf{W}_0^\top \nabla \widehat{\mathcal{L}}_{\mathbf{p}}(\mathbf{p}(\tau)) \tag{32}$$

$$= \mathbf{W}_0^\top \left( \mathbf{p}(\tau) - \alpha \|\mathbf{p}_0\|_2^2 \cdot \nabla \widehat{\mathcal{L}}_{\mathbf{p}}(\mathbf{p}(\tau)) \right) \tag{33}$$

$$= \mathbf{W}_0^\top \mathbf{p}(\tau + 1), \tag{34}$$

which concludes the proof. $\qquad\square$

Next, we will give the smoothness of the objective function with respect to $\mathbf{p}$, and as a property derived from this, we will show that the gradient will converge to zero with a sufficiently small step size. This is essentially a restatement of Lemma 6 in (Tarzanagh et al., 2023b), so please refer to that for the proof. We used assumption B and explicitly expressed $\ell$ as the binary cross-entropy loss.

**Lemma A.7** (Rephrased from (Tarzanagh et al. (2023b), Lemma 6)). *The function $\widehat{\mathcal{L}}(\mathbf{p})$ is $L$-smooth, where*

$$L := \frac{1}{n} \sum_{i \in [n]} \left( \|\boldsymbol{\nu}\|_2^2 \|\mathbf{X}^{(i)}\|_2^2 + 3\|\boldsymbol{\nu}\|_2 \|\mathbf{X}^{(i)}\|_2^3 \right). \tag{35}$$

*Furthermore, if a step size satisfies $\alpha < 1/L$, then, for any initialization $\mathbf{p}(0)$, we have*

$$\widehat{\mathcal{L}}\left(\mathbf{p}(\tau+1)\right) - \widehat{\mathcal{L}}\left(\mathbf{p}(\tau)\right) \le -\frac{\alpha}{2} \|\nabla_{\mathbf{p}} \widehat{\mathcal{L}}\left(\mathbf{p}(\tau)\right)\|_2^2, \tag{36}$$

*for all $t \ge 0$. This implies that*

$$\sum_{\tau=0}^{\infty} \|\nabla_{\mathbf{p}} \widehat{\mathcal{L}}\left(\mathbf{p}(\tau)\right)\|_2^2 < \infty, \quad \text{and} \quad \lim_{\tau \to \infty} \|\nabla_{\mathbf{p}} \widehat{\mathcal{L}}\left(\mathbf{p}(\tau)\right)\|_2^2 = 0. \tag{37}$$

Using Lemma A.7, by the definition of the smoothness, we have

$$\|\nabla_{\mathbf{p}} \widehat{\mathcal{L}}(\mathbf{p})\|_{\text{Lip}} \le L, \tag{38}$$

where $L$ is defined in Eq.35.

**Remark 2** (Smoothness under our data model). Lemma A.7 provides the smoothness of the empirical loss function using the operator norm of the training data inputs. Assumption (A4) requires that the step size is less than $1/\|\nabla_{\mathbf{p}} \widehat{\mathcal{L}}(\mathbf{p})\|_{\text{Lip}}$, and we comment on how small this value can be in the worst case under our data model. Here, suppose that the scale of the linear head is $\|\boldsymbol{\nu}\|_2 = \Theta\left(1/\|\boldsymbol{\mu}\|_2\right)$, and under Definition 3.1 for the data model, on a good run, we have

$$\max_{t \in [T]} \|\mathbf{x}_t^{(i)}\|_2 \le \|\boldsymbol{\mu}\|_2 + \max_{t \in [T]} \|\boldsymbol{\epsilon}_t^{(i)}\|_2 \le \|\boldsymbol{\mu}\|_2 + (1 + 1/C)\sqrt{\text{Tr}(\boldsymbol{\Sigma})}, \tag{39}$$

for any $i \in [n]$. Since $\|\mathbf{X}^{(i)}\|_2 \le \|\mathbf{X}^{(i)}\|_F \le \sqrt{T} \max_{t \in [T]} \|\mathbf{x}_t\|_2$, the smoothness $L$ in Lemma A.7 is upper-bounded as:

$$L \le \|\boldsymbol{\nu}\|_2^2 \max_{i \in [n]} \|\mathbf{X}^{(i)}\|_2^2 + 3\|\boldsymbol{\nu}\|_2 \max_{i \in [n]} \|\mathbf{X}^{(i)}\|_2^3 \tag{40}$$

$$\lesssim \frac{T}{\|\boldsymbol{\mu}\|_2^2} \left( \|\boldsymbol{\mu}\|_2 + (1 + 1/C)\sqrt{\text{Tr}(\boldsymbol{\Sigma})} \right)^2 + \frac{T^{3/2}}{\|\boldsymbol{\mu}\|_2} \left( \|\boldsymbol{\mu}\|_2 + (1 + 1/C)\sqrt{\text{Tr}(\boldsymbol{\Sigma})} \right)^3 \tag{41}$$

$$\lesssim T + T\frac{\text{Tr}(\boldsymbol{\Sigma})}{\|\boldsymbol{\mu}\|_2^2} + T^{3/2}\|\boldsymbol{\mu}\|_2^2 + T^{3/2}\frac{\text{Tr}(\boldsymbol{\Sigma})^{3/2}}{\|\boldsymbol{\mu}\|_2} \tag{42}$$

$$\lesssim T^{3/2}\|\boldsymbol{\mu}\|_2^5, \tag{43}$$

where the first inequality follows from Lemma A.7 and the next is derived by Eq.39. We use the inequality $(a+b)^2 \le 2(a^2+b^2)$ and $(a+b)^3 \le 4(a^3+b^3)$ in the third inequality, and the last line follows from the assumption (A2), which implies $\|\boldsymbol{\mu}\|_2 \gtrsim \text{Tr}(\boldsymbol{\Sigma})^{1/4}$. Consequently, we can bound the smoothness $L$ in Lemma A.7 by $T^{3/2}\|\boldsymbol{\mu}\|_2^5$.

Finally, we derive the equations that can be deduced from the assumptions in Section 3.5 and will be used in the remainder of the proof.

**Lemma A.8.** *Suppose the assumptions (A1) and (A2) hold. Then, we have*

$$\text{Tr}(\boldsymbol{\Sigma}) \ge Cn^2 \left(\log(Tn/\delta)\right)^2. \tag{44}$$

*Suppose the assumptions (A'1), (A'2), and (A'3) hold. Then we have,*

$$\text{Tr}(\boldsymbol{\Sigma}) \ge CT^4 n^2 \left(\log(Tn/\delta)\right)^2 / \rho^2, \tag{45}$$

$$\|\boldsymbol{\mu}\|_2 \ge CTn\sqrt{\log(Tn/\delta)}. \tag{46}$$

*Proof.* We will derive the equations using the assumptions. Combining assumption (A1) and (A2) gives us:

$$\text{Tr}(\boldsymbol{\Sigma}) \geq Cn\|\boldsymbol{\mu}\|_2^2 \geq Cn \cdot C^2 \text{Tr}(\boldsymbol{\Sigma})^{1/2} \log(Tn/\delta). \tag{47}$$

By rearranging this, we obtain Eq.44. From assumption (A'3), we have

$$CT^4 n^2 \left(\log(Tn/\delta)\right)^2 / \rho^2 \leq CT^4 n^2 \left(\log(Tn/\delta)\right)^2 \cdot \frac{\|\boldsymbol{\mu}\|_2^2}{C^2 T^2 \log(Tn/\delta)} \tag{48}$$

$$\leq \frac{T^2 n^2 \log(Tn/\delta)}{C} \cdot \frac{\text{Tr}(\boldsymbol{\Sigma})}{CT^2 n^2 \log(Tn/\delta)} \leq \text{Tr}(\boldsymbol{\Sigma}), \tag{49}$$

where the last line follows from assumption (A'1), and Eq.45 is derived. Finally, by combining assumptions (A'1) and (A'2), we have

$$\|\boldsymbol{\mu}\|_2^2 \geq C^2 (Tn)^{1/2} \log(Tn/\delta) \cdot C^{1/2} Tn \left(\log(Tn/\delta)\right)^{1/2} \|\boldsymbol{\mu}\|_2, \tag{50}$$

therefore, Eq.46 is shown by rearranging this equation. $\qquad\square$

# B PROOF OF LEMMA A.1

In this section, we will prove each high-probability event described in Lemma A.1. Specifically, we can show it by combining Lemma B.3, B.5, B.6, and B.8, using union bound argument. In the rest of this section, suppose that assumptions (A1)-(A6) hold.

First, we show the norm concentration of the Gaussian noise vectors. The next lemma gives the lower bound for the expectation of the $L_2$ norm.

**Lemma B.1.** *For a Gaussian vector $\mathbf{x} \sim N(0, \boldsymbol{\Sigma})$, we have*

$$\sqrt{\text{Tr}(\boldsymbol{\Sigma}) - 1} \leq \mathbb{E}\left[\|\mathbf{x}\|_2\right]. \tag{51}$$

*Proof of Lemma B.1.* We use the Gaussian Poincaré Inequality (Boucheron et al. (2003), Theorem 3.20):

$$\text{Var}\left(f(\mathbf{x})\right) \leq \mathbb{E}\left[\|\nabla f(\mathbf{x})\|_2^2\right], \tag{52}$$

where $f : \mathbb{R}^d \to \mathbb{R}$ is any continuously differentiable function. By taking $f$ as $f(\mathbf{x}) = \|\mathbf{x}\|_2$, since we have $\mathbb{E}\left[\|\nabla f(\mathbf{x})\|_2^2\right] = \mathbb{E}\left[\|\mathbf{x}/\|\mathbf{x}\|_2\|_2^2\right] = 1$,

$$1 \geq \text{Var}(f(\mathbf{x})) = \mathbb{E}\left[\|\mathbf{x}\|^2\right] - \left(\mathbb{E}\left[\|\mathbf{x}\|\right]\right)^2 = \text{Tr}(\boldsymbol{\Sigma}) - \left(\mathbb{E}\left[\|\mathbf{x}\|\right]\right)^2. \tag{53}$$

Rearranging the terms, we get

$$\sqrt{\text{Tr}(\boldsymbol{\Sigma}) - 1} \leq \mathbb{E}\left[\|\mathbf{x}\|_2\right], \tag{54}$$

which concludes the proof. $\qquad\square$

The following lemma is about the concentration of Lipschitz functions and is used to prove the norm concentration.

**Lemma B.2** (Rephrased from (Wainwright (2019), Theorem 3.16)). *For any $L$-Lipschitz function $f : \mathbb{R}^d \to \mathbb{R}$, we have*

$$\Pr_{\boldsymbol{\epsilon} \sim N(\mathbf{0}, \boldsymbol{\Sigma})} \left\{|f(\boldsymbol{\epsilon}) - \mathbb{E}\left[f(\boldsymbol{\epsilon})\right]| \geq w\right\} \leq 2\exp\left(-\frac{w^2}{4\sigma_{max}(\boldsymbol{\Sigma})L^2}\right). \tag{55}$$

Note that the coefficient 2 could be removed in the case of one-sided inequality.

**Remark 3.** Generally, this holds for strongly log-concave distributions, i.e., distributions with a density $p(x) = \exp(-\psi(x))$, where $\psi : \mathbb{R}^d \to \mathbb{R}$ is a strongly convex function. Here, we used the fact that the Gaussian distribution $N(\mathbf{0}, \boldsymbol{\Sigma})$ is a strongly log-concave distribution with parameter $\sigma_{min}(\boldsymbol{\Sigma}^{-1}) = 1/\sigma_{max}(\boldsymbol{\Sigma})$.

We are now ready to prove the norm concentration as follows.

**Lemma B.3** (Norm concentration). *There exists some constant $C' > 0$ depending on $C$ such that with probability at least $1 - \delta/4$,*

$$\left(1 - 1/\operatorname{Tr}(\boldsymbol{\Sigma}) - 1/C'\right) \sqrt{\operatorname{Tr}(\boldsymbol{\Sigma})} \le \|\boldsymbol{\epsilon}_t^{(i)}\|_2 \le (1 + 1/C') \sqrt{\operatorname{Tr}(\boldsymbol{\Sigma})}, \tag{56}$$

*for all $i \in [n], t \in [T]$*

*Proof of Lemma B.3.* From the definition of noise distribution, $\boldsymbol{\epsilon}_t^{(i)} \sim N(\mathbf{0}, \boldsymbol{\Sigma})$ for all $i \in [n], t \in [T]$. To begin with, we show the norm concentration of the Gaussian vector. For $\mathbf{x}, \mathbf{y} \sim N(\mathbf{0}, \boldsymbol{\Sigma})$, since we have

$$\|\mathbf{x}\|_2 - \|\mathbf{y}\|_2 = \frac{\|\mathbf{x}\|_2 - \|\mathbf{y}\|_2}{\|\mathbf{x} - \mathbf{y}\|_2} \|\mathbf{x} - \mathbf{y}\|_2 \le \|\mathbf{x} - \mathbf{y}\|_2, \tag{57}$$

$f(\mathbf{x}) = \|\mathbf{x}\|_2$ is 1-Lipschitz function. Using Lemma B.2, we get

$$\Pr\left\{\left|\|\boldsymbol{\epsilon}_t^{(i)}\|_2 - \mathbb{E}\left[\|\boldsymbol{\epsilon}\|_2\right]\right| > w\right\} \le 2 \exp\left(-\frac{w^2}{4\sigma_{max}(\boldsymbol{\Sigma})}\right), \tag{58}$$

for some $i \in [n], t \in [T]$. Taking union-bound gives

$$\forall i \in [n], \forall t \in [T], \ \Pr\left\{\left|\|\boldsymbol{\epsilon}_t^{(i)}\|_2 - \mathbb{E}\left[\|\boldsymbol{\epsilon}\|_2\right]\right| > w\right\} \le 2Tn \exp\left(-\frac{w^2}{4\sigma_{max}(\boldsymbol{\Sigma})}\right). \tag{59}$$

Lemma B.1 and Jensen inequality lead the following bound on the expectation of the Gaussian norm:

$$\sqrt{\operatorname{Tr}(\boldsymbol{\Sigma}) - 1} \le \mathbb{E}\left[\|\boldsymbol{\epsilon}\|_2\right] \le \sqrt{\operatorname{Tr}(\boldsymbol{\Sigma})}. \tag{60}$$

Using this and Eq. 59, we have with the probability at least $1 - \delta/4$,

$$\sqrt{\operatorname{Tr}(\boldsymbol{\Sigma}) - 1} - 2\sqrt{\sigma_{max}(\boldsymbol{\Sigma}) \log\left(\frac{8Tn}{\delta}\right)} \le \|\boldsymbol{\epsilon}_t^{(i)}\|_2 \le \sqrt{\operatorname{Tr}(\boldsymbol{\Sigma})} + 2\sqrt{\sigma_{max}(\boldsymbol{\Sigma}) \log\left(\frac{8Tn}{\delta}\right)}, \tag{61}$$

for all $i \in [n], t \in [T]$. By using Eq. 44 from assumption (A1), which implies $\operatorname{Tr}(\boldsymbol{\Sigma}) \ge C \log(Tn/\delta)$, we have

$$2\sqrt{\sigma_{max}(\boldsymbol{\Sigma}) \log\left(\frac{8Tn}{\delta}\right)} \le 2\sqrt{\frac{2\sigma_{max}(\boldsymbol{\Sigma}) \operatorname{Tr}(\boldsymbol{\Sigma})}{C}}, \tag{62}$$

where we used $8 < Tn/\delta$. Let $C' = \sqrt{C/(2\sigma_{max}(\boldsymbol{\Sigma}))}/2$, then Eqs. 61 and 62 give

$$\left(\sqrt{1 - 1/\operatorname{Tr}(\boldsymbol{\Sigma})} - 1/C'\right) \sqrt{\operatorname{Tr}(\boldsymbol{\Sigma})} \le \|\boldsymbol{\epsilon}_t^{(i)}\|_2 \le (1 + 1/C') \sqrt{\operatorname{Tr}(\boldsymbol{\Sigma})}, \tag{63}$$

for all $i \in [n], t \in [T]$. Since $1 - 1/\operatorname{Tr}(\boldsymbol{\Sigma}) < \sqrt{1 - 1/\operatorname{Tr}(\boldsymbol{\Sigma})}$, we obtain the desired inequality. $\square$

Next, we move on to the concentration inequality for the Gaussian random variables.

**Lemma B.4** (Gaussian tail bound, (Vershynin (2018), Prop 2.1.2)). *For a Gaussian variable $x \sim N(0, \sigma^2)$, the tail bound is given by*

$$\left(\frac{\sigma}{w} - \frac{\sigma^3}{w^3}\right) \cdot \frac{1}{\sqrt{2\pi}} \exp\left(-\frac{w^2}{2\sigma^2}\right) \le \Pr\{x \ge w\} \le \frac{1}{w} \cdot \frac{\sigma}{\sqrt{2\pi}} \exp\left(-\frac{w^2}{2\sigma^2}\right). \tag{64}$$

Using this, we can show the following result for the inner products of the noise vectors.

**Lemma B.5** (Inner-product of Noises). *There exists some constant $c_1 > 0$ such that with the probability at least $1 - \delta/4$,*

$$|\langle \boldsymbol{\epsilon}_t^{(i)}, \boldsymbol{\epsilon}_u^{(j)} \rangle| < c_1 \sqrt{\operatorname{Tr}(\boldsymbol{\Sigma})} \log(Tn/\delta), \tag{65}$$

*for all $i, j \in [n], t, u \in [T], (i, t) \ne (j, u)$.*

*Proof of Lemma B.5.* Before delving into the main part of the proof, we first show

$$\Pr\left\{\sqrt{\boldsymbol{\epsilon}^{\top}\boldsymbol{\Sigma}\boldsymbol{\epsilon}} \geq w\right\} \leq 2\exp\left(-\frac{w^2}{2\operatorname{Tr}(\boldsymbol{\Sigma}^2)}\right), \tag{66}$$

for the Gaussian vector $\boldsymbol{\epsilon} \sim N(\mathbf{0}, \boldsymbol{\Sigma})$. This is a result of the concentration inequality for the norm of Gaussian distribution, but unlike Lemma B.3, it handles the norm itself rather than the deviation around the mean of the norm, which is more useful to prove this lemma.

Let $\boldsymbol{\zeta} \sim N(\mathbf{0}, \mathbf{I})$, then $\boldsymbol{\Sigma}^{1/2}\boldsymbol{\zeta}$ follows the same distribution as $\boldsymbol{\epsilon}$. Since $\|\mathbf{x}\|_2 \leq \|\mathbf{x}\|_1$ for $\mathbf{x} \in \mathbb{R}^d$, for any $\lambda > 0$,

$$\Pr\left\{\sqrt{\boldsymbol{\zeta}^{\top}\boldsymbol{\Sigma}^2\boldsymbol{\zeta}} \geq w\right\} = \Pr\left\{\|\boldsymbol{\Sigma}\boldsymbol{\zeta}\|_2 \geq w\right\} \tag{67}$$

$$\leq \Pr\left\{\|\boldsymbol{\Sigma}\boldsymbol{\zeta}\|_1 \geq w\right\} \tag{68}$$

$$\leq \exp(-\lambda w) \cdot \prod_{k \in [d]} \mathbb{E}\exp\left(\lambda\sigma_k|\zeta_k|\right) \tag{69}$$

$$\leq \exp(-\lambda w) \cdot 2\prod_{k \in [d]} \mathbb{E}\exp\left(\lambda\sigma_k\zeta_k\right) \tag{70}$$

$$= 2\exp\left(\frac{\lambda^2}{2}\operatorname{Tr}(\boldsymbol{\Sigma}^2) - w\lambda\right), \tag{71}$$

where the second inequality follows from Markov inequality and the last follows from the moment-generating function of Gaussian distribution. Minimizing the upper bound over $\lambda$ gives the desired inequality Eq. 66.

Fix $i, j \in [n]$ and $t, u \in [T]$. For any $v, w > 0$, we have

$$\Pr\left\{|\langle\boldsymbol{\epsilon}_t^{(i)}, \boldsymbol{\epsilon}_u^{(j)}\rangle| > v\right\} \leq \Pr\left\{|\langle\boldsymbol{\epsilon}_t^{(i)}, \boldsymbol{\epsilon}_u^{(j)}\rangle| > v \mid \sqrt{\boldsymbol{\epsilon}_u^{(j)\top}\boldsymbol{\Sigma}\boldsymbol{\epsilon}_u^{(j)}} \leq w\right\} + \Pr\left\{\sqrt{\boldsymbol{\epsilon}_u^{(j)\top}\boldsymbol{\Sigma}\boldsymbol{\epsilon}_u^{(j)}} > w\right\}, \tag{72}$$

where we used the inequality $\Pr(A) = \Pr(B)\Pr(A|B) + \Pr(B^C)\Pr(A|B^C) \leq \Pr(B) + \Pr(A|B^C)$ for the event $A, B$, which gives tighter bound when outlier event $A$ and event $B$ share large common parts.

Under the condition $\boldsymbol{\epsilon}_u^{(j)}$ is fixed, since $\langle\boldsymbol{\epsilon}_t^{(i)}, \boldsymbol{\epsilon}_u^{(j)}\rangle$ follows $N(0, \boldsymbol{\epsilon}_u^{(j)\top}\boldsymbol{\Sigma}\boldsymbol{\epsilon}_u^{(j)})$, Lemma B.4 gives

$$\Pr\left\{|\langle\boldsymbol{\epsilon}_t^{(i)}, \boldsymbol{\epsilon}_u^{(j)}\rangle| > v\right\} \leq \frac{2}{v} \cdot \frac{\sqrt{\boldsymbol{\epsilon}_u^{(j)\top}\boldsymbol{\Sigma}\boldsymbol{\epsilon}_u^{(j)}}}{\sqrt{2\pi}}\exp\left(-\frac{v^2}{2\boldsymbol{\epsilon}_u^{(j)\top}\boldsymbol{\Sigma}\boldsymbol{\epsilon}_u^{(j)}}\right). \tag{73}$$

Thus, the conditional probability is bounded

$$\Pr\left\{|\langle\boldsymbol{\epsilon}_t^{(i)}, \boldsymbol{\epsilon}_u^{(j)}\rangle| > v \mid \sqrt{\boldsymbol{\epsilon}_u^{(j)\top}\boldsymbol{\Sigma}\boldsymbol{\epsilon}_u^{(j)}} \leq w\right\} \leq \frac{2}{v} \cdot \frac{w}{\sqrt{2\pi}}\exp\left(-\frac{v^2}{2w^2}\right). \tag{74}$$

Combining Eqs. 66 and 74 then applying union bound on 72, we obtain

$$\Pr\left\{\exists i, j \in [n], t, u \in [T], (i, t) \neq (j, u), \text{ s.t. } |\langle\boldsymbol{\epsilon}_t^{(i)}, \boldsymbol{\epsilon}_u^{(j)}\rangle| > v\right\}$$

$$\leq \Pr\left\{\exists i, j \in [n], t, u \in [T], (i, t) \neq (j, u), \text{ s.t. } |\langle\boldsymbol{\epsilon}_t^{(i)}, \boldsymbol{\epsilon}_u^{(j)}\rangle| > v \;\middle|\; \forall j \in [n], u \in [T], \sqrt{\boldsymbol{\epsilon}_u^{(j)\top}\boldsymbol{\Sigma}\boldsymbol{\epsilon}_u^{(j)}} \leq w\right\}$$

$$+ \Pr\left\{\exists j \in [n], u \in [T], \text{ s.t. } \sqrt{\boldsymbol{\epsilon}_u^{(j)\top}\boldsymbol{\Sigma}\boldsymbol{\epsilon}_u^{(j)}} > w\right\} \tag{75}$$

$$\leq \frac{2wT^2n^2}{\sqrt{2\pi}v}\exp\left(-\frac{v^2}{2w^2}\right) + 2Tn\exp\left(-\frac{w^2}{2\operatorname{Tr}(\boldsymbol{\Sigma}^2)}\right). \tag{76}$$

Let $w = c_1'\left(\log(Tn/\delta)\right)^{-1/2}v$ for some constant $c_1' > 0$. By Eq. 76, we have

$$\Pr\left\{\exists i, j \in [n], t, u \in [T], (i, t) \neq (j, u), \text{ s.t. } |\langle\boldsymbol{\epsilon}_t^{(i)}, \boldsymbol{\epsilon}_u^{(j)}\rangle| > v\right\}$$

$$\leq \frac{2T^2n^2}{\sqrt{2\pi}} \cdot c_1'\left(\log(Tn/\delta)\right)^{-1/2}\left(\frac{\delta}{Tn}\right)^{1/\left(2c_1'^2\right)} + 2Tn\exp\left(-\frac{c_1'^2v^2}{2\operatorname{Tr}(\boldsymbol{\Sigma}^2)\log(Tn/\delta)}\right). \tag{77}$$

Further, let $v = c_1\sqrt{\text{Tr}(\mathbf{\Sigma})}\log(Tn/\delta)$, where $c_1 > 0$ is some constant. Since $\text{Tr}(\mathbf{\Sigma}^2) \leq \sigma_{max}(\mathbf{\Sigma})\text{Tr}(\mathbf{\Sigma})$, we have

$$\Pr\left\{\exists i, j \in [n], t, u \in [T], (i, t) \neq (j, u), \text{ s.t. } |\langle \boldsymbol{\epsilon}_t^{(i)}, \boldsymbol{\epsilon}_u^{(j)}\rangle| > v\right\}$$

$$\leq \frac{2T^2n^2}{\sqrt{2\pi}} \cdot c_1'\left(\log(Tn/\delta)\right)^{-1/2}\left(\frac{\delta}{Tn}\right)^{1/\left(2c_1'^2\right)} + 2Tn\left(\frac{\delta}{Tn}\right)^{c_1'^2 c_1^2/(2\sigma_{max}(\mathbf{\Sigma}))} \tag{78}$$

$$\leq \frac{\delta}{8} + \frac{\delta}{8} = \frac{\delta}{4}, \tag{79}$$

where the last inequality is satisfied with the appropriate choice of $c_1, c_1' > 0$. $\qquad\square$

**Lemma B.6** (Inner-product of Signal and Noise). *There exists some constant $c_2 > 0$ such that with probability at least $1 - \delta/4$,*

$$|\langle \boldsymbol{\epsilon}_t^{(i)}, \boldsymbol{\mu}_c\rangle| < c_2\|\boldsymbol{\mu}\|_2\sqrt{\log(Tn/\delta)}, \tag{80}$$

*for all $i \in [n], t \in [T], c \in \{\pm 1\}$.*

*Proof.* We will show that the inequality for $\boldsymbol{\mu}_{+1}$ holds with probability at least $1 - \delta/8$. The same discussion applies to $\boldsymbol{\mu}_{-1}$. For the fixed $i \in [n], t \in [T]$, since $\langle \boldsymbol{\epsilon}_t^{(i)}, \boldsymbol{\mu}_{+1}\rangle$ follows the Gaussian distribution $N(0, \boldsymbol{\mu}_{+1}^\top\mathbf{\Sigma}\boldsymbol{\mu}_{+1})$, Lemma B.4 gives

$$\Pr\left\{\langle |\boldsymbol{\epsilon}_t^{(i)}, \boldsymbol{\mu}_{+1}\rangle| > w\right\} \leq \frac{2\sqrt{\boldsymbol{\mu}_{+1}^\top\mathbf{\Sigma}\boldsymbol{\mu}_{+1}}}{\sqrt{2\pi}w}\exp\left(-\frac{w^2}{2\boldsymbol{\mu}_{+1}^\top\mathbf{\Sigma}\boldsymbol{\mu}_{+1}}\right). \tag{81}$$

Let $w = c_2\|\boldsymbol{\mu}\|_2\sqrt{\log(Tn/\delta)}$ for some constant $c_2 > 0$, then applying union bound on Eq. 81 gives

$$\Pr\left\{\exists i \in [n], t \in [T], \text{ s.t. } |\langle \boldsymbol{\epsilon}_t^{(i)}, \boldsymbol{\mu}_{+1}\rangle| > w\right\}$$

$$\leq \frac{2Tn\sqrt{\sigma_{max}(\mathbf{\Sigma})}\|\boldsymbol{\mu}\|_2}{\sqrt{2\pi}w}\exp\left(-\frac{w^2}{2\sigma_{max}(\mathbf{\Sigma})\|\boldsymbol{\mu}\|_2^2}\right) \tag{82}$$

$$\leq \frac{2Tn\sqrt{\sigma_{max}(\mathbf{\Sigma})}}{\sqrt{2\pi}c_2\sqrt{\log(Tn/\delta)}}\left(\frac{\delta}{Tn}\right)^{c_2^2/(2\sigma_{max}(\mathbf{\Sigma}))} \tag{83}$$

$$< \frac{2\sqrt{\sigma_{max}(\mathbf{\Sigma})}}{\sqrt{2\pi}c_2} \cdot \left(\frac{\delta}{Tn}\right)^{c_2^2/(2\sigma_{max}(\mathbf{\Sigma}))-1} < \frac{\delta}{8}, \tag{84}$$

where the second last inequality follows from $\sqrt{\boldsymbol{\mu}_{+1}^\top\mathbf{\Sigma}\boldsymbol{\mu}_{+1}} \leq \sqrt{\sigma_{max}(\mathbf{\Sigma})}\|\boldsymbol{\mu}\|_2$ and the last one is satisfied with the appropriate choice of $c_2 > 0$. $\qquad\square$

**Lemma B.7** (Hoeffding Inequality). *Let $X_1, \ldots, X_n$ be i.i.d. random variables such that $0 \leq X \leq 1$ almost surely. Then for all $w > 0$, we have*

$$\Pr\left\{\left|\frac{1}{n}\sum_{i=1}^n X_i - \mathbb{E}[X]\right| > w\right\} \leq 2\exp\left(-2nw^2\right). \tag{85}$$

**Lemma B.8** (Number of Samples). *For all $c' > 0$, the following hold with probability at least $1 - \delta/4$:*

$$((1-\eta)/2 - c')n \leq |\mathcal{C}_{+1}| \leq ((1-\eta)/2 + c')n, \tag{86}$$

$$((1-\eta)/2 - c')n \leq |\mathcal{C}_{-1}| \leq ((1-\eta)/2 + c')n, \tag{87}$$

$$(\eta/2 - c')n \leq |\mathcal{N}_{+1}| \leq (\eta/2 + c')n, \tag{88}$$

$$(\eta/2 - c')n \leq |\mathcal{N}_{-1}| \leq (\eta/2 + c')n. \tag{89}$$

*Proof.* We show the first equation holds with probability at least $1 - \delta/16$. The proof of remaining cases follows similarly, and the desired result is achieved by using union bound.

The training data $i \in [n]$ belongs to $\mathcal{C}_{+1}$ when its true label $Y^{(i)} = 1$ and label flip does not occur. This event occurs independently, and its probability is calculated as $(1 - \eta)/2$. Since $|\mathcal{C}_{+1}| = \sum_{i \in [n]} \mathbf{1}_{Y^{(i)} = Y^{*(i)} = 1}$, applying Lemma B.7 to $X_i := \mathbf{1}_{Y^{(i)} = Y^{*(i)} = 1}$ leads

$$\Pr\left\{||\mathcal{C}_{+1}| - (1 - \eta)n/2| > c_3\right\} \le 2\exp\left(-2nc_3^2\right) < \frac{\delta}{16}, \tag{90}$$

where the last inequality follows from the assumption (A5) $n \ge C \log(1/\delta)$. $\square$

Finally, although we have incorporated it into the data model setup for simplicity, we show that if the token length $T$ is large to some extent, then $|\mathcal{W}_{+1}| \ge 1$ and $|\mathcal{W}_{-1}| \ge 1$ will hold with high probability.

**Lemma B.9** (Number of Weakly Relevant Tokens). *Suppose the number of weakly relevant tokens* $\zeta_{\mathcal{W}}T$ *satisfies* $\zeta_{\mathcal{W}}T \ge 2\log(n/\delta) + 1$. *Then, with probability at least* $1 - \delta$,

$$|\mathcal{W}_{+1}^{(i)}| \ge 1, \ |\mathcal{W}_{-1}^{(i)}| \ge 1, \tag{91}$$

*for all* $i \in [n]$.

*Proof.* From Definition 3.1 on the data model, we have

$$\Pr\left\{\exists i \in [n] \text{ s.t. } |\mathcal{W}_{+1}^{(i)}| = 0 \text{ or } |\mathcal{W}_{-1}^{(i)}| = 0\right\} \le n\left(\frac{1}{2}\right)^{\zeta_{\mathcal{W}}T - 1}. \tag{92}$$

This follows from union bound and the generation process of weakly relevant tokens. By using the condition on the size of weakly relevant tokens, the right-hand side is upper-bounded by $n(1/2)^{2\log(n/\delta)} = n(\delta/n)^{2\log 2} \le \delta$, where we used $2\log 2 > 1$. $\square$

## C  PROOF OF THEOREM 4.1

Before proceeding with the proof, we define the desirable events for the unseen data similarly as we evaluated the probability for the training data in Lemma A.1.

**Definition C.1.** We define $\mathcal{E}$ as the event that the following inequalities are satisfied for the unseen data $(\mathbf{X}, Y^*) \sim P^*$:

$$\forall t \in [T], \ (1 - 1/\text{Tr}(\mathbf{\Sigma}) - 1/C')\sqrt{\text{Tr}(\mathbf{\Sigma})} \le \|\boldsymbol{\epsilon}_t\|_2 \le (1 + 1/C')\sqrt{\text{Tr}(\mathbf{\Sigma})}, \tag{93}$$

$$\forall i \in [n], t, u \in [T], \ |\langle \boldsymbol{\epsilon}_t, \boldsymbol{\epsilon}_u^{(i)} \rangle| < c_1\sqrt{\text{Tr}(\mathbf{\Sigma})}\log(Tn/\delta), \tag{94}$$

$$\forall t \in [T], c \in \{\pm 1\}, \ |\langle \mathbf{x}_t, \boldsymbol{\mu}_c \rangle| < c_2\|\boldsymbol{\mu}\|_2\sqrt{\log(Tn/\delta)}, \tag{95}$$

where the constants $C', c_1, c_2$ are the same ones appeared in Lemma A.1.

**Lemma C.1.** *We have* $\Pr[\mathcal{E}] > 1 - \delta/n$.

*Proof of Lemma C.1.* Applying union bound on the modified versions of Lemma B.3, B.5, and B.6. Since there is no need to apply the union bound over the $n$ training data points, the outlier probability can be reduced by $1/n$ compared to the original lemma. $\square$

Additionally, we prepare an evaluation of the token scores for the training data on a good run. Note that the same results can be obtained for the test data when conditioned on $\mathcal{E}$. Please recall that $\pm$ is the sign to denote the possible range of the values.

**Lemma C.2** (Token Score). *Suppose that the norm of the linear head* $\boldsymbol{\nu} \propto \boldsymbol{\mu}_{+1} - \boldsymbol{\mu}_{-1}$. *Then, on a good run, for the clean data* $i \in \mathcal{C}$, *we have*

$$Y^{(i)} \cdot \gamma_t^{(i)} \in \frac{\|\boldsymbol{\nu}\|_2}{\sqrt{2}}\left(\|\boldsymbol{\mu}\|_2 \pm 2c_2\sqrt{\log(Tn/\delta)}\right), \quad |\gamma_v^{(i)}| \le \frac{\|\boldsymbol{\nu}\|_2}{\sqrt{2}} \cdot 2c_2\sqrt{\log(Tn/\delta)},$$

$$Y^{(i)} \cdot \gamma_u^{(i)} \in \frac{\|\boldsymbol{\nu}\|_2}{\sqrt{2}}\left(\rho\|\boldsymbol{\mu}\|_2 \pm 2c_2\sqrt{\log(Tn/\delta)}\right), \ Y^{(i)} \cdot \gamma_{u'}^{(i)} \in \frac{\|\boldsymbol{\nu}\|_2}{\sqrt{2}}\left(-\rho\|\boldsymbol{\mu}\|_2 \pm 2c_2\sqrt{\log(Tn/\delta)}\right),$$

*where $t \in \mathcal{R}, u \in \mathcal{W}_{Y^{(i)}}^{(i)}, u' \in \mathcal{W}_{-Y^{(i)}}^{(i)}, v \in \mathcal{I}$, and for the noisy data $j \in \mathcal{N}$, we have*

$$Y^{(j)} \cdot \gamma_t^{(j)} \in \frac{\|\boldsymbol{\nu}\|_2}{\sqrt{2}} \left( -\|\boldsymbol{\mu}\|_2 \pm 2c_2\sqrt{\log(Tn/\delta)} \right), \quad |\gamma_v^{(j)}| \leq \frac{\|\boldsymbol{\nu}\|_2}{\sqrt{2}} \cdot 2c_2\sqrt{\log(Tn/\delta)},$$

$$Y^{(j)} \cdot \gamma_u^{(j)} \in \frac{\|\boldsymbol{\nu}\|_2}{\sqrt{2}} \left( \rho\|\boldsymbol{\mu}\|_2 \pm 2c_2\sqrt{\log(Tn/\delta)} \right), \quad Y^{(j)} \cdot \gamma_{u'}^{(j)} \in \frac{\|\boldsymbol{\nu}\|_2}{\sqrt{2}} \left( -\rho\|\boldsymbol{\mu}\|_2 \pm 2c_2\sqrt{\log(Tn/\delta)} \right),$$

*where $t \in \mathcal{R}, u \in \mathcal{W}_{Y^{(j)}}^{(j)}, u' \in \mathcal{W}_{-Y^{(j)}}^{(j)}, v \in \mathcal{I}$.*

*Proof.* Using Eq 21, we have

$$Y^{(i)} \cdot \gamma_t^{(i)} = Y^{(i)} \cdot \mathbf{x}_t^{(i)\top} \frac{\|\boldsymbol{\nu}\|_2}{\|\boldsymbol{\mu}_{+1} - \boldsymbol{\mu}_{-1}\|_2} (\boldsymbol{\mu}_{+1} - \boldsymbol{\mu}_{-1}) \in \frac{\|\boldsymbol{\nu}\|_2}{\sqrt{2}} \left( \|\boldsymbol{\mu}\|_2 \pm 2c_2\sqrt{\log(Tn/\delta)} \right). \tag{96}$$

The other equations are derived as well. $\qquad\square$

### C.1 OVERFITTING PART

First, we show how clean data fits. On a good run, it is sufficient to show that the model's output becomes deterministically positive when the true label is 1. The same argument applies when the true label is $-1$. For clean data $i \in \mathcal{C}_{+1}$, we have

$$f(\mathbf{X}^{(i)}) = \boldsymbol{\nu}^\top \mathbf{X}^{(i)\top} \mathbb{S} \left( \mathbf{X}^{(i)} \mathbf{W}^\top \mathbf{p} \right) = \sum_{t \in \mathcal{R}} \gamma_t^{(i)} \mathbb{S} \left( \mathbf{X}^{(i)} \mathbf{W}^\top \mathbf{p} \right)_t + \sum_{u \in [T] \setminus \mathcal{R}} \gamma_u^{(i)} \mathbb{S} \left( \mathbf{X}^{(i)} \mathbf{W}^\top \mathbf{p} \right)_u. \tag{97}$$

Substituting $\mathbf{p} = R \cdot \mathbf{W}(\boldsymbol{\mu}_{+1} + \boldsymbol{\mu}_{-1} + \sum_{j \in \mathcal{N}} \beta_j \boldsymbol{\epsilon}_{u_j}^{(j)})$, where $\beta_j = \Theta(1/n) > 0$ and $u_j \in \mathcal{W}_{Y^{(j)}}^{(j)}$, to this equation. For the probability assigned to the relevant tokens, we have

$$\sum_{t \in \mathcal{R}} \mathbb{S}(\mathbf{X}^{(i)} \mathbf{W}^\top \mathbf{p})_t = \sum_{t \in \mathcal{R}} \frac{\exp(R \cdot \mathbf{x}_t^{(i)\top}(\boldsymbol{\mu}_{+1} + \boldsymbol{\mu}_{-1} + \sum_{j \in \mathcal{N}} \beta_j \boldsymbol{\epsilon}_{u_j}^{(j)}))}{\sum_{t' \in [T]} \exp(R \cdot \mathbf{x}_{t'}^{(i)\top}(\boldsymbol{\mu}_{+1} + \boldsymbol{\mu}_{-1} + \sum_{j \in \mathcal{N}} \beta_j \boldsymbol{\epsilon}_{u_j}^{(j)}))} \tag{98}$$

$$= \left( 1 + \frac{\sum_{u \in [T] \setminus \mathcal{R}} \exp \left( R \cdot \mathbf{x}_u^{(i)\top}(\boldsymbol{\mu}_{+1} + \boldsymbol{\mu}_{-1} + \sum_{j \in \mathcal{N}} \beta_j \boldsymbol{\epsilon}_{u_j}^{(j)}) \right)}{\sum_{t \in \mathcal{R}} \exp \left( R \cdot \mathbf{x}_t^{(i)\top}(\boldsymbol{\mu}_{+1} + \boldsymbol{\mu}_{-1} + \sum_{j \in \mathcal{N}} \beta_j \boldsymbol{\epsilon}_{u_j}^{(j)}) \right)} \right)^{-1} \tag{99}$$

$$\geq 1 - \frac{\sum_{u \in [T] \setminus \mathcal{R}} \exp \left( R \cdot \mathbf{x}_u^{(i)\top}(\boldsymbol{\mu}_{+1} + \boldsymbol{\mu}_{-1} + \sum_{j \in \mathcal{N}} \beta_j \boldsymbol{\epsilon}_{u_j}^{(j)}) \right)}{\sum_{t \in \mathcal{R}} \exp \left( R \cdot \mathbf{x}_t^{(i)\top}(\boldsymbol{\mu}_{+1} + \boldsymbol{\mu}_{-1} + \sum_{j \in \mathcal{N}} \beta_j \boldsymbol{\epsilon}_{u_j}^{(j)}) \right)} \tag{100}$$

$$\geq 1 - \frac{1 - \zeta_{\mathcal{R}}}{\zeta_{\mathcal{R}}} \cdot \frac{\max_{u \in [T] \setminus \mathcal{R}} \left\{ \exp \left( R \cdot \mathbf{x}_u^{(i)\top}(\boldsymbol{\mu}_{+1} + \boldsymbol{\mu}_{-1} + \sum_{j \in \mathcal{N}} \beta_j \boldsymbol{\epsilon}_{u_j}^{(j)}) \right) \right\}}{\min_{t \in \mathcal{R}} \left\{ \exp \left( R \cdot \mathbf{x}_t^{(i)\top}(\boldsymbol{\mu}_{+1} + \boldsymbol{\mu}_{-1} + \sum_{j \in \mathcal{N}} \beta_j \boldsymbol{\epsilon}_{u_j}^{(j)}) \right) \right\}}, \tag{101}$$

where the first inequality follows from $1/(1+x) \geq 1 - x, \forall x > -1$, and in the second inequality, we used the fact that the number of relevant tokens is given by $\zeta_{\mathcal{R}} T$.

By Lemma A.1, for all relevant token $t \in \mathcal{R}$, we have

$$\mathbf{x}_t^{(i)\top} \left( \boldsymbol{\mu}_{+1} + \boldsymbol{\mu}_{-1} + \sum_{j \in \mathcal{N}} \beta_j \boldsymbol{\epsilon}_{u_j}^{(j)} \right)$$

$$= \left( \boldsymbol{\mu}_{Y^{(i)}} + \boldsymbol{\epsilon}_t^{(i)} \right)^{\top} \left( \boldsymbol{\mu}_{+1} + \boldsymbol{\mu}_{-1} + \sum_{j \in \mathcal{N}} \beta_j \boldsymbol{\epsilon}_{u_j}^{(j)} \right) \tag{102}$$

$$\geq \|\boldsymbol{\mu}\|_2^2 - \left( 2 + \sum_{j \in \mathcal{N}} \beta_j \right) c_2 \|\boldsymbol{\mu}\|_2 \sqrt{\log(Tn/\delta)} - \left( \sum_{j \in \mathcal{N}} \beta_j \right) c_1 \sqrt{\mathrm{Tr}(\boldsymbol{\Sigma})} \log(Tn/\delta) \tag{103}$$

$$\geq \|\boldsymbol{\mu}\|_2^2/2, \tag{104}$$

where we used the assumption (A2), leading to $\|\boldsymbol{\mu}\|_2/4 > \left( 2 + \sum_{j \in \mathcal{N}} \beta_j \right) c_2 \sqrt{\log(Tn/\delta)}$ and $\|\boldsymbol{\mu}\|_2^2/4 > \left( \sum_{j \in \mathcal{N}} \beta_j \right) c_1 \sqrt{\mathrm{Tr}(\boldsymbol{\Sigma})} \log(Tn/\delta)$. Using the same argument, for all tokens except for relevant tokens: $u \in [T] \setminus \mathcal{R}$, i.e. weakly relevant token or irrelevant token, we have

$$\mathbf{x}_u^{(i)\top} \left( \boldsymbol{\mu}_{+1} + \boldsymbol{\mu}_{-1} + \sum_{j \in \mathcal{N}} \beta_j \boldsymbol{\epsilon}_{u_j}^{(j)} \right)$$

$$\leq \rho \|\boldsymbol{\mu}\|_2^2 + \left( 2 + \rho \sum_{j \in \mathcal{N}} \beta_j \right) c_2 \|\boldsymbol{\mu}\|_2 \sqrt{\log(Tn/\delta)} + \left( \sum_{j \in \mathcal{N}} \beta_j \right) c_1 \sqrt{\mathrm{Tr}(\boldsymbol{\Sigma})} \log(Tn/\delta)$$

$$\leq \|\boldsymbol{\mu}\|_2^2/4, \tag{105}$$

where we used the assumption (A3) $\rho \leq 1/C$. Substituting Eqs. 104 and 105 to Eq.101 yields

$$\sum_{t \in \mathcal{R}} \mathbb{S}(\mathbf{X}^{(i)} \mathbf{W}^{\top} \mathbf{p})_t \geq 1 - \frac{1 - \zeta_{\mathcal{R}}}{\zeta_{\mathcal{R}}} \exp\left( -\frac{R}{4} \|\boldsymbol{\mu}\|_2^2 \right) > \frac{1}{2}, \tag{106}$$

by taking $R > \frac{4 \log(2(1-\zeta_{\mathcal{R}})/\zeta_{\mathcal{R}})}{\|\boldsymbol{\mu}\|_2^2}$.

Combining this probability evaluation with Lemma C.2 gives us

$$\sum_{t \in \mathcal{R}} \gamma_t^{(i)} \mathbb{S}\left( \mathbf{X}^{(i)} \mathbf{W}^{\top} \mathbf{p} \right)_t + \sum_{u \in [T] \setminus \mathcal{R}} \gamma_u^{(i)} \mathbb{S}\left( \mathbf{X}^{(i)} \mathbf{W}^{\top} \mathbf{p} \right)_u$$

$$\geq \frac{\|\boldsymbol{\nu}\|_2}{\sqrt{2}} \left\{ \left( \|\boldsymbol{\mu}\|_2 - 2c_2 \sqrt{\log(Tn/\delta)} \right) \left( \sum_{t \in \mathcal{R}} \mathbb{S}(\mathbf{X}^{(i)} \mathbf{W}^{\top} \mathbf{p})_t \right) \right.$$

$$\left. - \left( \rho \|\boldsymbol{\mu}\|_2 + 2c_2 \sqrt{\log(Tn/\delta)} \right) \left( 1 - \sum_{t \in \mathcal{R}} \mathbb{S}(\mathbf{X}^{(i)} \mathbf{W}^{\top} \mathbf{p})_t \right) \right\} \tag{107}$$

$$= \frac{\|\boldsymbol{\mu}\|_2}{\sqrt{2}} \left( (1 + \rho) \|\boldsymbol{\mu}\|_2 \cdot \left( \sum_{t \in \mathcal{R}} \mathbb{S}(\mathbf{X}^{(i)} \mathbf{W}^{\top} \mathbf{p})_t \right) - \rho \|\boldsymbol{\mu}\|_2 - 2c_2 \sqrt{\log(Tn/\delta)} \right) \tag{108}$$

$$\geq \frac{\|\boldsymbol{\mu}\|_2}{\sqrt{2}} \left( \frac{1 - \rho}{2} \|\boldsymbol{\mu}\|_2 - 2c_2 \sqrt{\log(Tn/\delta)} \right) \tag{109}$$

$$> 0, \tag{110}$$

where the last line follows again from assumptions (A2) and (A3). Thus, the model fits the true labels for the clean data.

Next, we demonstrate that the model successfully fits the label noise for the noisy data $j \in \mathcal{N}$. Without loss of generality, we will show for the case $Y^{*(j)} = -1, Y^{(j)} = 1$. The model output is

given by

$$
\begin{aligned}
f(\mathbf{X}^{(j)}) &= \boldsymbol{\nu}^\top \mathbf{X}^{(j)\top} \mathbb{S}\left(\mathbf{X}^{(j)}\mathbf{W}^\top \mathbf{p}\right) \\
&= \gamma_{u_j}^{(j)} \mathbb{S}\left(\mathbf{X}^{(j)}\mathbf{W}^\top \mathbf{p}\right)_{u_j} + \sum_{t \in [T] \setminus \{u_j\}} \gamma_t^{(j)} \mathbb{S}\left(\mathbf{X}^{(j)}\mathbf{W}^\top \mathbf{p}\right)_t.
\end{aligned}
\tag{111}
$$

Since $u_j \in \mathcal{W}_{Y^{(j)}}^{(j)}$, we bound its token score by using Lemma C.2

$$
\gamma_{u_j}^{(j)} \geq \frac{\|\boldsymbol{\nu}\|_2}{\sqrt{2}} \cdot \left(\rho\|\boldsymbol{\mu}\|_2 - 2c_2\sqrt{\log(Tn/\delta)}\right).
\tag{112}
$$

Regarding the softmax probability, we obtain the following bound for the $u_j$-th token by a similar procedure to Eq. 101:

$$
\mathbb{S}\left(\mathbf{X}^{(j)}\mathbf{W}^\top \mathbf{p}\right)_{u_j} \geq 1 - \sum_{t \in [T] \setminus \{u_j\}} \exp\left(R \cdot \left(\mathbf{x}_t^{(j)} - \mathbf{x}_{u_j}^{(j)}\right)^\top \left(\boldsymbol{\mu}_{+1} + \boldsymbol{\mu}_{-1} + \sum_{k \in \mathcal{N}} \beta_k \boldsymbol{\epsilon}_{u_k}^{(k)}\right)\right).
\tag{113}
$$

The softmax probabilities of other terms can be bounded as

$$
\begin{aligned}
\mathbb{S}\left(\mathbf{X}^{(j)}\mathbf{W}^\top \mathbf{p}\right)_t &= \frac{\exp(R \cdot \mathbf{x}_t^{(j)\top}(\boldsymbol{\mu}_{+1} + \boldsymbol{\mu}_{-1} + \sum_{k \in \mathcal{N}} \beta_k \boldsymbol{\epsilon}_{u_k}^{(k)}))}{\sum_{t' \in [T]} \exp(R \cdot \mathbf{x}_{t'}^{(j)\top}(\boldsymbol{\mu}_{+1} + \boldsymbol{\mu}_{-1} + \sum_{k \in \mathcal{N}} \beta_k \boldsymbol{\epsilon}_{u_k}^{(k)}))} \\
&\leq \exp\left(R \cdot \left(\mathbf{x}_t^{(j)} - \mathbf{x}_{u_j}^{(j)}\right)^\top \left(\boldsymbol{\mu}_{+1} + \boldsymbol{\mu}_{-1} + \sum_{k \in \mathcal{N}} \beta_k \boldsymbol{\epsilon}_{u_k}^{(k)}\right)\right),
\end{aligned}
\tag{114}
$$

for $t \in [T] \setminus \{u_j\}$. In the remainder of this proof, we will show that the softmax probability assigned to $u_j$ becomes sufficiently large, leading to $f(\mathbf{X}^{(j)}) > 0$.

We will evaluate the term inside softmax of Eqs. 113 and 114. By Lemma A.1, we have for all tokens except for $u_j$: $t \in [T] \setminus \{u_j\}$:

$$
\begin{aligned}
&\left(\mathbf{x}_t^{(j)} - \mathbf{x}_{u_j}^{(j)}\right)^\top \left(\boldsymbol{\mu}_{+1} + \boldsymbol{\mu}_{-1} + \sum_{k \in \mathcal{N}} \beta_k \boldsymbol{\epsilon}_{u_k}^{(k)}\right) \\
&\leq -\beta_j \left(1 - 1/\operatorname{Tr}(\boldsymbol{\Sigma}) - 1/C'\right)^2 \operatorname{Tr}(\boldsymbol{\Sigma}) + (1 - \rho)\|\boldsymbol{\mu}\|_2^2 \\
&\quad + \left(4 + (1 + \rho)\sum_{k \in \mathcal{N}} \beta_k\right) c_2\|\boldsymbol{\mu}\|_2\sqrt{\log(Tn/\delta)} + 2c_1\left(\sum_{k \in \mathcal{N}} \beta_k\right)\sqrt{\operatorname{Tr}(\boldsymbol{\Sigma})}\log(Tn/\delta)
\end{aligned}
\tag{115}
$$

$$
\leq -\frac{\beta_j}{2}\operatorname{Tr}(\boldsymbol{\Sigma}),
\tag{116}
$$

where we used the assumptions (A1), (A2) and (A3), which implies $\beta_j \operatorname{Tr}(\boldsymbol{\Sigma})/8$ is greater than the other three terms. Here, the result of Lemma A.8, that is, Eq.44, was used in the last term. By substituting Eq. 116 to Eq. 113, we have

$$
\mathbb{S}\left(\mathbf{X}^{(j)}\mathbf{W}^\top \mathbf{p}\right)_{u_j} \geq 1 - (T - 1)\exp\left(-\frac{R\beta_j}{2}\operatorname{Tr}(\boldsymbol{\Sigma})\right),
\tag{117}
$$

and similarly, substituting Eq. 116 to Eq. 114 implies

$$
\mathbb{S}\left(\mathbf{X}^{(j)}\mathbf{W}^\top \mathbf{p}\right)_t \leq \exp\left(-\frac{R\beta_j}{2}\operatorname{Tr}(\boldsymbol{\Sigma})\right),
\tag{118}
$$

for $t \in [T] \setminus \{u_j\}$. Using these inequalities for softmax probability and lower-bounds for token scores in Eq. 112 and Lemma C.2, Eq. 111 gives us

$$\gamma_{u_j}^{(j)} \mathbb{S}\left(\mathbf{X}^{(j)}\mathbf{W}^\top \mathbf{p}\right)_{u_j} + \sum_{t \in [T] \setminus \{u_j\}} \gamma_t^{(j)} \mathbb{S}\left(\mathbf{X}^{(j)}\mathbf{W}^\top \mathbf{p}\right)_t$$

$$\geq \frac{\|\boldsymbol{\nu}\|_2}{\sqrt{2}} \Bigg\{ \left(\rho\|\boldsymbol{\mu}\|_2 - 2c_2\sqrt{\log(Tn/\delta)}\right) \mathbb{S}\left(\mathbf{X}^{(j)}\mathbf{W}^\top \mathbf{p}\right)_{u_j}$$

$$- \left(\|\boldsymbol{\mu}\|_2 + 2c_2\sqrt{\log(Tn/\delta)}\right) \sum_{t \in [T] \setminus \{u_j\}} \mathbb{S}\left(\mathbf{X}^{(j)}\mathbf{W}^\top \mathbf{p}\right)_t \Bigg\} \tag{119}$$

$$\geq \frac{\|\boldsymbol{\nu}\|_2}{\sqrt{2}} \left( \rho\|\boldsymbol{\mu}\|_2 - 2c_2\sqrt{\log(Tn/\delta)} - (1+\rho)\|\boldsymbol{\mu}\|_2 \cdot (T-1) \exp\left(-\frac{R\beta_j}{2}\operatorname{Tr}(\boldsymbol{\Sigma})\right) \right) \tag{120}$$

$$\geq \frac{\|\boldsymbol{\nu}\|_2}{\sqrt{2}} \cdot \sqrt{\log(Tn/\delta)} > 0, \tag{121}$$

where the inequality in the last line follows by using the assumption (A3) $\rho \geq C\sqrt{\log(Tn/\delta)}/\|\boldsymbol{\mu}\|_2$ and taking $R > 0$ sufficiently large so that the third term in Eq.120 becomes smaller than $c_2\sqrt{\log(Tn/\delta)}$. Specifically, we choose $R$ to satisfy

$$R > \frac{2}{\max_{j \in \mathcal{N}}\{\beta_j\}\operatorname{Tr}(\boldsymbol{\Sigma})} \log\left(\frac{(1+\rho)(T-1)\|\boldsymbol{\mu}\|_2}{c_2\sqrt{\log(Tn/\delta)}}\right).$$

Consequently, we have $\operatorname{sign}\left(f(\mathbf{X}^{(j)})\right) = Y^{(j)}$ for all noisy data $j \in \mathcal{N}$, indicating that the model successfully fits the label noise.

## C.2 GENERALIZATION PART

The generalization error of the model $f(\mathbf{X}) = \boldsymbol{\nu}^\top \mathbf{X}\mathbb{S}(\mathbf{X}\mathbf{W}^\top \mathbf{p})$ is given by

$$\Pr_{(\mathbf{X}, Y^*) \sim P^*}\left[\operatorname{sign}(f(\mathbf{X})) \neq Y^*\right]$$

$$= \frac{1}{2}\Pr_{\mathbf{X} \sim P^*_{\mathbf{X}|Y^*=1}}\left[\boldsymbol{\nu}^\top \mathbf{X}^\top \mathbb{S}(\mathbf{X}\mathbf{W}^\top \mathbf{p}) < 0\right] + \frac{1}{2}\Pr_{\mathbf{X} \sim P^*_{\mathbf{X}|Y^*=-1}}\left[\boldsymbol{\nu}^\top \mathbf{X}^\top \mathbb{S}(\mathbf{X}\mathbf{W}^\top \mathbf{p}) > 0\right]. \tag{122}$$

We will discuss the first term in the following. A similar argument applies to the second term as well. By conditioning on the event $\mathcal{E}$, we have

$$\Pr_{\mathbf{X} \sim P^*_{\mathbf{X}|Y^*=1}}\left[\boldsymbol{\nu}^\top \mathbf{X}^\top \mathbb{S}(\mathbf{X}\mathbf{W}^\top \mathbf{p}) < 0\right]$$

$$\leq \Pr_{\mathbf{X} \sim P^*_{\mathbf{X}|Y^*=1}}\left[\boldsymbol{\nu}^\top \mathbf{X}^\top \mathbb{S}(\mathbf{X}\mathbf{W}^\top \mathbf{p}) < 0 \mid \mathcal{E}\right] + \Pr_{\mathbf{X} \sim P^*_{\mathbf{X}|Y=1}}\left[\mathcal{E}^c\right] \tag{123}$$

$$\leq \Pr_{\mathbf{X} \sim P^*_{\mathbf{X}|Y^*=1}}\left[\boldsymbol{\nu}^\top \mathbf{X}^\top \mathbb{S}(\mathbf{X}\mathbf{W}^\top \mathbf{p}) < 0 \mid \mathcal{E}\right] + \delta/n, \tag{124}$$

where we used $\Pr(A) \leq \Pr(B) + \Pr(A|B^C)$, and the last line follows from Lemma C.1.

We will show that the output $f(\mathbf{X})$ becomes positive under the conditioning on $\mathcal{E}$, leading to the first term in Eq. 124 becoming zero. This is essentially the same argument as in the fitting of clean data in Section C.1, with the only difference being the application of conditions in $\mathcal{E}$ instead of Lemma A.1. Therefore, we avoid repeating the same discussion here. From Eqs. 122 and 124, the desired generalization error is bounded by $\delta/n$.

## D LEMMAS FOR THEOREM 4.2

In this section, we will present lemmas concerning the gradient descent dynamics of token selection, which are used for proving Theorem 4.2. For clarity, the essential lemmas are put first, and minor trivial lemmas are placed at the end of this section.

### D.1 PRELIMINARY LEMMAS

**Lemma D.1.** *Let* $\mathbf{p}(\tau)$ *be a gradient iteration at* $\tau$*-th time step. Then, there exists unique coefficients such that*

$$\mathbf{p}(\tau) = \lambda_{+1}(\tau)\mathbf{W}\boldsymbol{\mu}_{+1} + \lambda_{-1}(\tau)\mathbf{W}\boldsymbol{\mu}_{-1} + \sum_{i\in[n]}\sum_{t\in[T]}\rho_{i,t}(\tau)\mathbf{W}\boldsymbol{\epsilon}_t^{(i)}.$$

*Furthermore, these coefficients satisfy the following recurrence equations.*

*The initialization is*

$$\lambda_c(0) = \rho_{i,t}(0) = 0,$$

*for any* $c \in \{\pm 1\}, i \in [n], t \in [T]$*, and the signal updates are given by*

$$
\begin{aligned}
\lambda_c(\tau+1) = \lambda_c(\tau) &+ \frac{\alpha}{n}\sum_{i\in\mathcal{C}_c}(-\ell_i'(\tau))\sum_{t\in\mathcal{R}}s^{(i)}(\tau)_t\left(\gamma_t^{(i)} - \sum_{u\in[T]}s^{(i)}(\tau)_u\gamma_u^{(i)}\right) \\
&- \frac{\alpha}{n}\sum_{j\in\mathcal{N}_{-c}}(-\ell_j'(\tau))\sum_{t\in\mathcal{R}}s^{(j)}(\tau)_t\left(\gamma_t^{(j)} - \sum_{u\in[T]}s^{(j)}(\tau)_u\gamma_u^{(j)}\right) \\
&+ \frac{\alpha}{n}\sum_{i\in\mathcal{C}_c\cup\mathcal{N}_c}(-\ell_i'(\tau))\sum_{t\in\mathcal{W}_c^{(i)}}s^{(i)}(\tau)_t\cdot\rho\left(\gamma_t^{(i)} - \sum_{u\in[T]}s^{(i)}(\tau)_u\gamma_u^{(i)}\right) \\
&- \frac{\alpha}{n}\sum_{i\in\mathcal{C}_{-c}\cup\mathcal{N}_{-c}}(-\ell_i'(\tau))\sum_{t\in\mathcal{W}_c^{(i)}}s^{(i)}(\tau)_t\cdot\rho\left(\gamma_t^{(i)} - \sum_{u\in[T]}s^{(i)}(\tau)_u\gamma_u^{(i)}\right),
\end{aligned}
$$

*for any* $c \in \{\pm 1\}$*, and the noise updates are given by*

$$\rho_{i,t}(\tau+1) = \rho_{i,t}(\tau) + \frac{\alpha}{n}\cdot(-\ell_i'(\tau))\cdot Y^{(i)}\cdot s^{(i)}(\tau)_t\left(\gamma_t^{(i)} - \sum_{u\in[T]}s^{(i)}(\tau)_u\gamma_u^{(i)}\right),$$

*for all* $i \in [n], t \in [T]$*.*

*Proof of Lemma D.1.* Since the noise vectors $\{\boldsymbol{\epsilon}_t^{(i)}\}_{i\in[n],t\in[T]}$ follow the continuous distribution, $\{\boldsymbol{\mu}_{+1}, \boldsymbol{\mu}_{-1}\} \cup \{\boldsymbol{\epsilon}_t^{(i)}\}_{i\in[n],t\in[T]}$ are linearly independent with probability 1. Therefore, the learned parameter $\mathbf{p}(\tau)$ can be uniquely decomposed. It remains to show that the coefficients satisfying the recurrence equations in the statement match $\mathbf{p}(\tau)$ updated by the gradient descent.

The equality holds at $\tau = 0$ because the parameter is initialized as $\mathbf{p}(0) = \mathbf{0}$, and the coefficients are set to zero. Suppose that the equality holds at the time step $\tau$, then Eq. 18 provides

$$\mathbf{p}(\tau+1) - \mathbf{p}(\tau) = \frac{\alpha}{n}\sum_{i\in[n]}(-\ell_i'(\tau))\cdot Y^{(i)}\cdot\left(\sum_{t\in[T]}s^{(i)}(\tau)_t\left(\gamma_t^{(i)} - \sum_{u\in[T]}s^{(i)}(\tau)_u\gamma_u^{(i)}\right)\mathbf{W}\mathbf{x}_t^{(i)}\right). \tag{125}$$

This is further decomposed using the problem setting:

$$\mathbf{x}_t^{(i)} = \begin{cases} \boldsymbol{\mu}_{Y^{*(i)}} + \boldsymbol{\epsilon}_t^{(i)}, & t \in \mathcal{R}, \\ \rho\boldsymbol{\mu}_{+1} + \boldsymbol{\epsilon}_t^{(i)}, & t \in \mathcal{W}_{+1}^{(i)}, \\ \rho\boldsymbol{\mu}_{-1} + \boldsymbol{\epsilon}_t^{(i)}, & t \in \mathcal{W}_{-1}^{(i)}, \\ \boldsymbol{\epsilon}_t^{(i)}, & t \in \mathcal{I}, \end{cases}$$

and we have

$$
\begin{aligned}
\mathbf{p}(\tau+1) - \mathbf{p}(\tau) &= \frac{\alpha}{n} \sum_{i \in [n]} (-\ell_i'(\tau)) \cdot Y^{(i)} \cdot \left( \sum_{t \in \mathcal{R}} s^{(i)}(\tau)_t \left( \gamma_t^{(i)} - \sum_{u \in [T]} s^{(i)}(\tau)_u \gamma_u^{(i)} \right) \mathbf{W} \boldsymbol{\mu}_{Y^{*(i)}} \right) \\
&+ \frac{\alpha}{n} \sum_{i \in [n]} (-\ell_i'(\tau)) \cdot Y^{(i)} \cdot \left( \sum_{t \in \mathcal{W}_{+1}^{(i)}} s^{(i)}(\tau)_t \left( \gamma_t^{(i)} - \sum_{u \in [T]} s^{(i)}(\tau)_u \gamma_u^{(i)} \right) \rho \mathbf{W} \boldsymbol{\mu}_{+1} \right) \\
&+ \frac{\alpha}{n} \sum_{i \in [n]} (-\ell_i'(\tau)) \cdot Y^{(i)} \cdot \left( \sum_{t \in \mathcal{W}_{-1}^{(i)}} s^{(i)}(\tau)_t \left( \gamma_t^{(i)} - \sum_{u \in [T]} s^{(i)}(\tau)_u \gamma_u^{(i)} \right) \rho \mathbf{W} \boldsymbol{\mu}_{-1} \right) \\
&+ \frac{\alpha}{n} \sum_{i \in [n]} (-\ell_i'(\tau)) \cdot Y^{(i)} \cdot \left( \sum_{t \in [T]} s^{(i)}(\tau)_t \left( \gamma_t^{(i)} - \sum_{u \in [T]} s^{(i)}(\tau)_u \gamma_u^{(i)} \right) \mathbf{W} \boldsymbol{\epsilon}_t^{(i)} \right) .
\end{aligned}
\tag{126}
$$

Therefore, the coefficients updated with the equation in the statement and the parameter updated by the gradient descent (Eq. 126) match at the time step $\tau + 1$. $\qquad \square$

We denote the updates of $\lambda_{+1}(\tau)$, $\lambda_{-1}(\tau)$, $\{\rho_{i,t}(\tau)\}_{i \in [n], t \in [T]}$ via gradient descent by $\Delta\lambda_{+1}(\tau)$, $\Delta\lambda_{-1}(\tau)$, $\{\Delta\rho_{i,t}(\tau)\}_{i \in [n], t \in [T]}$, respectively. By using this notation, the 1-step gradient update can be expressed as follows:

$$
\mathbf{p}(\tau+1) - \mathbf{p}(\tau) = -\alpha \nabla_{\mathbf{p}} \widehat{\mathcal{L}}(\mathbf{p}(\tau)) \tag{127}
$$

$$
= \Delta\lambda_{+1}(\tau) \mathbf{W} \boldsymbol{\mu}_{+1} + \Delta\lambda_{-1}(\tau) \mathbf{W} \boldsymbol{\mu}_{-1} + \sum_{i \in [n]} \sum_{t \in [T]} \Delta\rho_{i,t}(\tau) \mathbf{W} \boldsymbol{\epsilon}_t^{(i)}. \tag{128}
$$

In the following, we will proceed with the convergence analysis based on the sign and comparison of the updates of these coefficients.

The following corollary gives us the range of these updates. Although somewhat complicated, it can be obtained simply by substituting the concentration evaluation for the token score. In the signal learning of class $c \in \{\pm 1\}$, the balance of contribution between the clean data $\mathcal{C}_c$ and the noisy data $\mathcal{N}_{-c}$: the data sampled as class $c$ and added label noise, is significant. Specifically, we are primarily interested in the first and second terms of Eq.129. For noise learning, there are two terms: $\|\boldsymbol{\mu}\|_2$ term and $\sqrt{\log(Tn/\delta)}$ term, but our primary focus is on the first term. We will carefully examine the sign and magnitude of the first term.

**Corollary D.1.** *For the signal updates at time step $\tau$: $\Delta\lambda_c(\tau), c \in \{\pm 1\}$, on a good run, we have*

$$\Delta\lambda_c(\tau) \in$$

$$\frac{\alpha\|\boldsymbol{\nu}\|_2}{\sqrt{2}n} \sum_{i\in\mathcal{C}_c} (-\ell_i'(\tau)) \left(\sum_{t\in\mathcal{R}} s^{(i)}(\tau)_t\right) \left(1 - \sum_{t\in\mathcal{R}} s^{(i)}(\tau)_t\right) (1 \pm \rho)\|\boldsymbol{\mu}\|_2$$

$$- \frac{\alpha\|\boldsymbol{\nu}\|_2}{\sqrt{2}n} \sum_{j\in\mathcal{N}_{-c}} (-\ell_j'(\tau)) \left(\sum_{t\in\mathcal{R}} s^{(j)}(\tau)_t\right) \left(1 - \sum_{t\in\mathcal{R}} s^{(j)}(\tau)_t\right) (1 \pm \rho)\|\boldsymbol{\mu}\|_2$$

$$\pm \frac{\alpha\|\boldsymbol{\nu}\|_2}{\sqrt{2}n} \sum_{i\in\mathcal{C}_c\cup\mathcal{N}_{-c}} (-\ell_i'(\tau)) \left(\sum_{t\in\mathcal{R}} s^{(i)}(\tau)_t \left(1 - s^{(i)}(\tau)_t\right)\right) \cdot 4c_2\sqrt{\log(Tn/\delta)}$$

$$+ \frac{\alpha\|\boldsymbol{\nu}\|_2}{\sqrt{2}n} \sum_{i\in\mathcal{C}_c} (-\ell_i'(\tau)) \left(\sum_{t\in\mathcal{W}_c^{(i)}} s^{(i)}(\tau)_t\right) \left(-\sum_{t\in\mathcal{R}} s^{(i)}(\tau)_t + 2\rho \sum_{t\in\mathcal{W}_{-c}^{(i)}} s^{(i)}(\tau)_t + \rho \sum_{t\in\mathcal{R}\cup\mathcal{I}} s^{(i)}(\tau)_t\right) \rho\|\boldsymbol{\mu}\|_2$$

$$+ \frac{\alpha\|\boldsymbol{\nu}\|_2}{\sqrt{2}n} \sum_{i\in\mathcal{N}_c} (-\ell_i'(\tau)) \left(\sum_{t\in\mathcal{W}_c^{(i)}} s^{(i)}(\tau)_t\right) \left(\sum_{t\in\mathcal{R}} s^{(i)}(\tau)_t + 2\rho \sum_{t\in\mathcal{W}_{-c}^{(i)}} s^{(i)}(\tau)_t + \rho \sum_{t\in\mathcal{R}\cup\mathcal{I}} s^{(i)}(\tau)_t\right) \rho\|\boldsymbol{\mu}\|_2$$

$$- \frac{\alpha\|\boldsymbol{\nu}\|_2}{\sqrt{2}n} \sum_{i\in\mathcal{C}_{-c}} (-\ell_i'(\tau)) \left(\sum_{t\in\mathcal{W}_c^{(i)}} s^{(i)}(\tau)_t\right) \left(\sum_{t\in\mathcal{R}} s^{(i)}(\tau)_t + 2\rho \sum_{t\in\mathcal{W}_{-c}^{(i)}} s^{(i)}(\tau)_t + \rho \sum_{t\in\mathcal{R}\cup\mathcal{I}} s^{(i)}(\tau)_t\right) \rho\|\boldsymbol{\mu}\|_2$$

$$- \frac{\alpha\|\boldsymbol{\nu}\|_2}{\sqrt{2}n} \sum_{i\in\mathcal{N}_{-c}} (-\ell_i'(\tau)) \left(\sum_{t\in\mathcal{W}_c^{(i)}} s^{(i)}(\tau)_t\right) \left(-\sum_{t\in\mathcal{R}} s^{(i)}(\tau)_t + 2\rho \sum_{t\in\mathcal{W}_{-c}^{(i)}} s^{(i)}(\tau)_t + \rho \sum_{t\in\mathcal{R}\cup\mathcal{I}} s^{(i)}(\tau)_t\right) \rho\|\boldsymbol{\mu}\|_2$$

$$\pm \frac{\alpha\|\boldsymbol{\nu}\|_2}{\sqrt{2}n} \sum_{i\in[n]} (-\ell_i'(\tau)) \left(\sum_{t\in\mathcal{W}_c^{(i)}} s^{(i)}(\tau)_t \left(1 - s^{(i)}(\tau)_t\right)\right) \rho \cdot 4c_2\sqrt{\log(Tn/\delta)}, \tag{129}$$

*and for noise updates $\{\Delta\rho_{i,t}(\tau)\}_{i\in[n],t\in[T]}$, we have*

$$\Delta\rho_{i,t}(\tau) \in \frac{\alpha\|\boldsymbol{\nu}\|_2}{\sqrt{2}n}\cdot(-\ell_i'(\tau))\cdot s^{(i)}(\tau)_t\left(1-\sum_{t'\in\mathcal{R}}s^{(i)}(\tau)_{t'}\right)(1\pm\rho)\|\boldsymbol{\mu}\|_2$$
$$\pm\frac{\alpha\|\boldsymbol{\nu}\|_2}{\sqrt{2}n}\cdot(-\ell_i'(\tau))\cdot s^{(i)}(\tau)_t\left(1-s^{(i)}(\tau)_t\right)\cdot 4c_2\sqrt{\log(Tn/\delta)}, \tag{130}$$

$$\Delta\rho_{i,u}(\tau) \in \frac{\alpha\|\boldsymbol{\nu}\|_2}{\sqrt{2}n}\cdot(-\ell_i'(\tau))\cdot s^{(i)}(\tau)_u\left(-\sum_{t\in\mathcal{R}}s^{(i)}(\tau)_t+2\rho\sum_{t\in\mathcal{W}_{-Y^{(i)}}^{(i)}}s^{(i)}(\tau)_t+\rho\sum_{t\in\mathcal{R}\cup\mathcal{I}}s^{(i)}(\tau)_t\right)\|\boldsymbol{\mu}\|_2$$
$$\pm\frac{\alpha\|\boldsymbol{\nu}\|_2}{\sqrt{2}n}\cdot(-\ell_i'(\tau))\cdot s^{(i)}(\tau)_u\left(1-s^{(i)}(\tau)_u\right)\cdot 4c_2\sqrt{\log(Tn/\delta)}, \tag{131}$$

$$\Delta\rho_{i,u'}(\tau) \in \frac{\alpha\|\boldsymbol{\nu}\|_2}{\sqrt{2}n}\cdot(-\ell_i'(\tau))\cdot s^{(i)}(\tau)_{u'}\left(-\sum_{t\in\mathcal{R}}s^{(i)}(\tau)_t-2\rho\sum_{t\in\mathcal{W}_{Y^{(i)}}^{(i)}}s^{(i)}(\tau)_t-\rho\sum_{t\in\mathcal{R}\cup\mathcal{I}}s^{(i)}(\tau)_t\right)\|\boldsymbol{\mu}\|_2$$
$$\pm\frac{\alpha\|\boldsymbol{\nu}\|_2}{\sqrt{2}n}\cdot(-\ell_i'(\tau))\cdot s^{(i)}(\tau)_{u'}\left(1-s^{(i)}(\tau)_{u'}\right)\cdot 4c_2\sqrt{\log(Tn/\delta)}, \tag{132}$$

$$\Delta\rho_{i,v}(\tau) \in \frac{\alpha\|\boldsymbol{\nu}\|_2}{\sqrt{2}n}\cdot(-\ell_i'(\tau))\cdot s^{(i)}(\tau)_v\left(-\sum_{t\in\mathcal{R}}s^{(i)}(\tau)_t-\rho\sum_{t\in\mathcal{W}_{Y^{(i)}}^{(i)}}s^{(i)}(\tau)_t+\rho\sum_{t\in\mathcal{W}_{-Y^{(i)}}^{(i)}}s^{(i)}(\tau)_t\right)\|\boldsymbol{\mu}\|_2$$
$$\pm\frac{\alpha\|\boldsymbol{\nu}\|_2}{\sqrt{2}n}\cdot(-\ell_i'(\tau))\cdot s^{(i)}(\tau)_v\left(1-s^{(i)}(\tau)_v\right)\cdot 4c_2\sqrt{\log(Tn/\delta)}, \tag{133}$$

*where $i\in\mathcal{C}, t\in\mathcal{R}, u\in\mathcal{W}_{Y^{(i)}}^{(i)}, u'\in\mathcal{W}_{-Y^{(i)}}^{(i)}, v\in\mathcal{I}$, and for noisy data $j\in\mathcal{N}$, we have*

$$\Delta\rho_{j,t}(\tau) \in -\frac{\alpha\|\boldsymbol{\nu}\|_2}{\sqrt{2}n}\cdot(-\ell_j'(\tau))\cdot s^{(j)}(\tau)_t\left(1-\sum_{t'\in\mathcal{R}}s^{(j)}(\tau)_{t'}\right)(1\pm\rho)\|\boldsymbol{\mu}\|_2$$
$$\pm\frac{\alpha\|\boldsymbol{\nu}\|_2}{\sqrt{2}n}\cdot(-\ell_j'(\tau))\cdot s^{(j)}(\tau)_t\left(1-s^{(j)}(\tau)_t\right)\cdot 4c_2\sqrt{\log(Tn/\delta)}, \tag{134}$$

$$\Delta\rho_{j,u}(\tau) \in \frac{\alpha\|\boldsymbol{\nu}\|_2}{\sqrt{2}n}\cdot(-\ell_j'(\tau))\cdot s^{(j)}(\tau)_u\left(\sum_{t\in\mathcal{R}}s^{(j)}(\tau)_t+2\rho\sum_{t\in\mathcal{W}_{-Y^{(j)}}^{(j)}}s^{(j)}(\tau)_t+\rho\sum_{t\in\mathcal{R}\cup\mathcal{I}}s^{(j)}(\tau)_t\right)\|\boldsymbol{\mu}\|_2$$
$$\pm\frac{\alpha\|\boldsymbol{\nu}\|_2}{\sqrt{2}n}\cdot(-\ell_j'(\tau))\cdot s^{(j)}(\tau)_u\left(1-s^{(j)}(\tau)_u\right)\cdot 4c_2\sqrt{\log(Tn/\delta)}, \tag{135}$$

$$\Delta\rho_{j,u'}(\tau) \in \frac{\alpha\|\boldsymbol{\nu}\|_2}{\sqrt{2}n}\cdot(-\ell_j'(\tau))\cdot s^{(j)}(\tau)_{u'}\left(\sum_{t\in\mathcal{R}}s^{(j)}(\tau)_t-2\rho\sum_{t\in\mathcal{W}_{Y^{(j)}}^{(j)}}s^{(j)}(\tau)_t-\rho\sum_{t\in\mathcal{R}\cup\mathcal{I}}s^{(j)}(\tau)_t\right)\|\boldsymbol{\mu}\|_2$$
$$\pm\frac{\alpha\|\boldsymbol{\nu}\|_2}{\sqrt{2}n}\cdot(-\ell_j'(\tau))\cdot s^{(j)}(\tau)_{u'}\left(1-s^{(j)}(\tau)_{u'}\right)\cdot 4c_2\sqrt{\log(Tn/\delta)}, \tag{136}$$

$$\Delta\rho_{j,v}(\tau) \in \frac{\alpha\|\boldsymbol{\nu}\|_2}{\sqrt{2}n}\cdot(-\ell_j'(\tau))\cdot s^{(j)}(\tau)_v\left(\sum_{t\in\mathcal{R}}s^{(j)}(\tau)_t-\rho\sum_{t\in\mathcal{W}_{Y^{(j)}}^{(j)}}s^{(j)}(\tau)_t+\rho\sum_{t\in\mathcal{W}_{-Y^{(j)}}^{(j)}}s^{(j)}(\tau)_t\right)\|\boldsymbol{\mu}\|_2$$
$$\pm\frac{\alpha\|\boldsymbol{\nu}\|_2}{\sqrt{2}n}\cdot(-\ell_j'(\tau))\cdot s^{(j)}(\tau)_v\left(1-s^{(j)}(\tau)_v\right)\cdot 4c_2\sqrt{\log(Tn/\delta)}, \tag{137}$$

*where $t\in\mathcal{R}, u\in\mathcal{W}_{Y^{(j)}}^{(j)}, u'\in\mathcal{W}_{-Y^{(j)}}^{(j)}, v\in\mathcal{I}$.*

*Proof.* The proof is completed by substituting the range of token score on a good run obtained in Lemma C.2 into the update equations in Lemma D.1. Here, note that for $i \in \mathcal{C}_{+1} \cup \mathcal{N}_{-1}$ and any relevant token $t \in \mathcal{R}$, we have

$$\gamma_t^{(i)} - \sum_{u \in [T]} s^{(i)}(\tau)_u \gamma_u^{(i)} = \sum_{u \in [T] \setminus \{t\}} s^{(i)}(\tau)_u \left( \gamma_t^{(i)} - \gamma_u^{(i)} \right) \tag{138}$$

$$\in \sum_{u \in \mathcal{R} \setminus \{t\}} s^{(i)}(\tau)_u \cdot \left( \pm 4c_2 \sqrt{\log(Tn/\delta)} \right) + \sum_{u \in \mathcal{I}} s^{(i)}(\tau)_u \left( \|\boldsymbol{\mu}\|_2 \pm 4c_2 \sqrt{\log(Tn/\delta)} \right)$$

$$+ \sum_{u \in \mathcal{W}_{+1}^{(i)}} s^{(i)}(\tau)_u \left( (1-\rho)\|\boldsymbol{\mu}\|_2 \pm 4c_2 \sqrt{\log(Tn/\delta)} \right)$$

$$+ \sum_{u \in \mathcal{W}_{-1}^{(i)}} s^{(i)}(\tau)_u \left( (1+\rho)\|\boldsymbol{\mu}\|_2 \pm 4c_2 \sqrt{\log(Tn/\delta)} \right) \tag{139}$$

$$\subseteq \left( 1 - \sum_{t \in \mathcal{R}} s^{(i)}(\tau)_t \right) (1 \pm \rho) \|\boldsymbol{\mu}\|_2 + \left( 1 - s^{(i)}(\tau)_t \right) \left( \pm 4c_2 \sqrt{\log(Tn/\delta)} \right). \tag{140}$$

In the same way, for any weakly relevant token $u \in \mathcal{W}_{+1}^{(i)}$, we have

$$\gamma_u^{(i)} - \sum_{w \in [T]} s^{(i)}(\tau)_w \gamma_w^{(i)} \in \left( -\sum_{t \in \mathcal{R}} s^{(i)}(\tau)_t + \rho \left( 2 \sum_{t \in \mathcal{W}_{-1}^{(i)}} s^{(i)}(\tau)_t + \sum_{t \in \mathcal{R} \cup \mathcal{I}} s^{(i)}(\tau)_t \right) \right) \|\boldsymbol{\mu}\|_2$$

$$+ \left( 1 - s^{(i)}(\tau)_u \right) \left( \pm 4c_2 \sqrt{\log(Tn/\delta)} \right). \tag{141}$$

and for $u \in \mathcal{W}_{-1}^{(i)}$,

$$\gamma_u^{(i)} - \sum_{w \in [T]} s^{(i)}(\tau)_w \gamma_w^{(i)} \in \left( -\sum_{t \in \mathcal{R}} s^{(i)}(\tau)_t - \rho \left( 2 \sum_{t \in \mathcal{W}_{+1}^{(i)}} s^{(i)}(\tau)_t + \sum_{t \in \mathcal{R} \cup \mathcal{I}} s^{(i)}(\tau)_t \right) \right) \|\boldsymbol{\mu}\|_2$$

$$+ \left( 1 - s^{(i)}(\tau)_u \right) \left( \pm 4c_2 \sqrt{\log(Tn/\delta)} \right). \tag{142}$$

Finally, for any irrelevant token $v \in \mathcal{I}$, we have

$$\gamma_v^{(i)} - \sum_{w \in [T]} s^{(i)}(\tau)_w \gamma_w^{(i)} \in \left( -\sum_{t \in \mathcal{R}} s^{(i)}(\tau)_t - \rho \sum_{t \in \mathcal{W}_{+1}^{(i)}} s^{(i)}(\tau)_t + \rho \sum_{t \in \mathcal{W}_{-1}^{(i)}} s^{(i)}(\tau)_t \right) \|\boldsymbol{\mu}\|_2$$

$$+ \left( 1 - s^{(i)}(\tau)_v \right) \left( \pm 4c_2 \sqrt{\log(Tn/\delta)} \right). \tag{143}$$

Substituting them to the updates in Lemma D.1 leads to the desired equations. $\square$

**Lemma D.2** (Ratio of Loss Derivative). *Suppose that the norm of the linear head $\boldsymbol{\nu} \propto \boldsymbol{\mu}_{+1} - \boldsymbol{\mu}_{-1}$ scales as $\Theta(1/\|\boldsymbol{\mu}\|_2)$. There exists an absolute constant $C_\ell > 0$ such that on a good run, we have for all time step $\tau \geq 0$,*

$$\max_{i,j \in [n]} \frac{\ell_i'(\tau)}{\ell_j'(\tau)} < C_\ell, \tag{144}$$

*Proof.* Recall that the derivative of the loss function is given by

$$-\ell_i'(\tau) = \frac{1}{1 + \exp\left( \sum_{t \in [T]} s^{(i)}(\tau)_t \gamma_t^{(i)} \right)}, \tag{145}$$

for any $i \in [n], \tau \geq 0$. On a good run, Lemma C.2 gives us the score of relevant token, for all $i \in \mathcal{C}, j \in \mathcal{N}, t \in \mathcal{R}$:

$$\gamma_t^{(i)} = \Theta\left(1 \pm \frac{2c_2\sqrt{\log(Tn/\delta)}}{\|\boldsymbol{\mu}\|_2}\right), \ \gamma_t^{(j)} = \Theta\left(-1 \pm \frac{2c_2\sqrt{\log(Tn/\delta)}}{\|\boldsymbol{\mu}\|_2}\right), \quad (146)$$

and for $i \in [n]$, the score of weakly relevant token $u \in \mathcal{W}_{+1}, u' \in \mathcal{W}_{-1}$ and irrelevant token $v \in \mathcal{I}$ are given by

$$\gamma_u^{(i)} = \Theta\left(\rho \pm \frac{2c_2\sqrt{\log(Tn/\delta)}}{\|\boldsymbol{\mu}\|_2}\right), \ \gamma_{u'}^{(i)} = \Theta\left(-\rho \pm \frac{2c_2\sqrt{\log(Tn/\delta)}}{\|\boldsymbol{\mu}\|_2}\right), \quad (147)$$

$$|\gamma_v^{(i)}| \leq \Theta\left(\frac{\sqrt{\log(Tn/\delta)}}{\|\boldsymbol{\mu}\|_2}\right). \quad (148)$$

The assumption (A2) leads to $\sqrt{\log(Tn/\delta)}/\|\boldsymbol{\mu}\|_2 = o(1)$; therefore there exists some constant $c > 0$ such that $|\gamma_t^{(i)}| < c$ for any $i \in [n], t \in [T]$. Since $-\ell_i'$ is monotonically decreasing, we have

$$\frac{1}{1 + \exp(c)} < -\ell_i'(\tau) < \frac{1}{1 + \exp(-c)}. \quad (149)$$

This leads to the conclusion with the constant $C_\ell = 1 + \exp(c)/(1 + \exp(-c))$. $\qquad \square$

**Remark 4** (Ratio of Loss Derivative). This lemma, which shows that the gradients of loss function for clean data and noisy data remain within a constant factor of each other at every time step, is a critical component of the proof in the existing analyses of linear classifiers and two-layer neural networks (Chatterji & Long, 2021; Frei et al., 2022; Xu & Gu, 2023). However, in the learning of token selection, the output is always an affine combination of the token scores, and the output scale is not changed. Therefore, as long as the balance of the loss derivatives in the token scores is maintained, the training process itself need not be considered. To ensure that the derivative of the loss function for each token remains within a constant factor, a small linear head scale, as described in Lemma D.2, is required. If the scale of the linear head is too large, little gradient will be generated for clean data even at the initial weights, and learning the signal vectors will not progress.

**Lemma D.3.** *Suppose that the gradient of loss function satisfies* $\lim_{\tau \to \infty} \|\nabla_{\mathbf{p}}\widehat{\mathcal{L}}(\mathbf{p}(\tau))\|_2 = 0$*, and the assumption (A4) for the step size* $\alpha$ *holds. Then there exists the token index* $t_i^* \in [T]$ *for each* $i \in [n]$*, and we have*

$$\lim_{\tau \to \infty} s_{t_i^*}^{(i)}(\tau) = 1, \ \text{and} \ \lim_{\tau \to \infty} s_t^{(i)}(\tau) = 0, \quad (150)$$

*for all* $i \in [n], t \in [T] \setminus \{t_i^*\}$*.*

*Proof.* We first show the technical result that for linearly independent vectors $\mathbf{v}_1, \ldots, \mathbf{v}_m \in \mathbb{R}^d$ and coefficients $a_1, \ldots, a_m \in \mathbb{R}$, there exists a constant $c > 0$ such that

$$\left\|\sum_{i \in [m]} a_i \mathbf{v}_i\right\|_2 \geq c \sum_{i \in [m]} |a_i|. \quad (151)$$

We can discuss the case for $\sum_{i \in [m]} |a_i| = 1$ without loss of generality. Considering the map from the unit sphere of $\ell_1^m$ to $\mathbb{R}$ via $(a_1, \ldots, a_m) \mapsto \|\sum_{i \in [m]} a_i \mathbf{v}_i\|_2$, since this map is continuous and the domain it compact, this function attains a minimum value. The linear independence of $\mathbf{v}_1, \ldots, \mathbf{v}_m$ implies this minimum value is positive, and we take this value as $c > 0$ and thus conclude Eq.151.

Recall that the gradient of the empirical loss function is given by the linear combination of $\{\mathbf{W}\mathbf{x}_t^{(i)}\}_{i \in [n], t \in [T]}$:

$$\nabla_{\mathbf{p}}\widehat{\mathcal{L}}(\mathbf{p}) = \frac{1}{n} \sum_{i=1}^{n} \ell_i' \cdot Y^{(i)} \cdot \left(\sum_{t \in [T]} s_t^{(i)}\left(\gamma_t^{(i)} - \sum_{u \in [T]} s_u^{(i)}\gamma_u^{(i)}\right)\mathbf{W}\mathbf{x}_t^{(i)}\right). \quad (152)$$

Since this norm converges to zero, we have

$$\forall \epsilon > 0, \exists T_0 > 0 \text{ s.t. } \forall \tau \geq T_0, \|\nabla_{\mathbf{p}} \widehat{\mathcal{L}}(\mathbf{p}(\tau))\|_2 < \epsilon. \tag{153}$$

Combining Eq.151 and the fact that $\{\mathbf{W}\mathbf{x}_t^{(i)}\}_{i\in[n],t\in[T]}$ are linearly independent with probability 1, if $\|\nabla_{\mathbf{p}} \widehat{\mathcal{L}}(\mathbf{p}(\tau))\|_2 < \epsilon$ holds for some $\epsilon > 0, \tau > 0$, then we have

$$\frac{|\ell_i'|}{n} \cdot s^{(i)}(\tau)_t \cdot \left| \gamma_t^{(i)} - \sum_{u\in[T]} s^{(i)}(\tau)_u \gamma_u^{(i)} \right| < \epsilon/c, \tag{154}$$

for all $i \in [n], t \in [T]$. Given that the linear head $\boldsymbol{\nu}$ is fixed and the output scale remains unchanged, note that there exists some constant $0 < c_1, c_2 < 1$ such that $c_1 < |\ell_i'| < c_2$. For $\epsilon' := n\epsilon/(c|\ell_i'|)$, Eq.154 gives us

$$s^{(i)}(\tau)_t < \sqrt{\epsilon'}, \text{ or } \left| \gamma_t^{(i)} - \sum_{u\in[T]} s^{(i)}(\tau)_u \gamma_u^{(i)} \right| < \sqrt{\epsilon'}. \tag{155}$$

To proceed, suppose that the second equation holds for multiple $t_1, t_2 \in [T]$, then we have

$$|\gamma_{t_1}^{(i)} - \gamma_{t_2}^{(i)}| < 2\sqrt{\epsilon'}. \tag{156}$$

Here, since $\{\boldsymbol{\epsilon}_t^{(i)}\}_{i\in[n],t\in[T]}$ are continuous random variables, these realizations take distinct values almost surely. For any sufficiently small $\epsilon$ such that $2\sqrt{\epsilon'} < \min_{i\in[n],t_1,t_2\in[T]} |\gamma_{t_1}^{(i)} - \gamma_{t_2}^{(i)}|$, Eq.156 leads to a contradiction, so the second equation does not hold for multiple $t$. Using $\sum_t s^{(i)}(\tau)_t = 1$, there exists some token $t_i^*(\tau) \in [T]$, we have $s^{(i)}(\tau)_t < \sqrt{\epsilon'}$ for $t \in [T] \setminus \{t_i^*(\tau)\}$, and $s^{(i)}(\tau)_{t_i^*(\tau)} > 1 - (T-1)\sqrt{\epsilon'}$. Here, from the step size assumption (A4) and Lemma D.4, since $s^{(i)}(\tau)_t$ changes by at most a constant factor in a single gradient descent step, the token index $t_i^*(\tau)$ is determined without depending on the time step $\tau$ for sufficiently small $\epsilon$. We denote this value as $t_i^*$, which appears in the statement. Consequently, for any sufficiently small $\epsilon_1 := \sqrt{\epsilon'}$ and $\epsilon_2 := (T-1)\epsilon_1$, from Eq.153 ,there exists $T_0 > 0$ such that $\forall \tau \geq T_0$

$$s^{(i)}(\tau)_t < \epsilon_1, \quad s^{(i)}(\tau)_{t_i^*} > 1 - \epsilon_2, \tag{157}$$

which completes the proof. $\qquad \square$

Based on the preparation of the above lemmas, we are ready to provide proofs for Lemmas A.2, A.3, and A.4 introduced in the proof sketch in section A.2. For convenience, we restate each proposition and provide the proof below.

## D.2 Proof of Lemma A.2

**Lemma A.2.** *Suppose the step size $\alpha$ satisfies the assumption (A4). Then, there exists the token index $t_i^* \in [T]$ for each $i \in [n]$, and we have*

$$\lim_{\tau\to\infty} s_{t_i^*}^{(i)}(\tau) = 1, \text{ and } \lim_{\tau\to\infty} s_t^{(i)}(\tau) = 0, \tag{25}$$

*for all $i \in [n], t \in [T] \setminus \{t_i^*\}$.*

*Proof.* Combining the step-size condition in assumption (A4) with Lemma A.7 and D.3 gives us the desired result. $\qquad \square$

## D.3 Proof of Lemma A.3

**Lemma A.3.** *Suppose that the assumptions (A'1)-(A'3), (A4)-(A6) hold, and the norm of the linear head $\boldsymbol{\nu} \propto \boldsymbol{\mu}_{+1} - \boldsymbol{\mu}_{-1}$ scales as $\Theta(1/\|\boldsymbol{\mu}\|_2)$. For any clean data $i \in \mathcal{C}$, on a good run, for all time step $\tau \geq 0$ and all weakly relevant or irrelevant token $u \in \mathcal{W} \cup \mathcal{I}$, we have*

$$s^{(i)}(\tau)_u \leq \max_{t\in\mathcal{R}} \left\{ s^{(i)}(\tau)_t \right\}. \tag{26}$$

*Proof.* We will proceed by induction. The equation holds at initialization because all elements are equal to $1/T$. By taking the ratio of the softmax probability, we have

$$\frac{\max_{t \in \mathcal{R}} \left\{ s^{(i)}(\tau+1)_t \right\}}{s^{(i)}(\tau+1)_u} = \frac{\max_{t \in \mathcal{R}} \left\{ \exp\left( \mathbf{x}_t^{(i)\top} \mathbf{W}^\top \mathbf{p}(\tau+1) \right) \right\}}{\exp\left( \mathbf{x}_u^{(i)\top} \mathbf{W}^\top \mathbf{p}(\tau+1) \right)} \tag{158}$$

$$= \frac{\max_{t \in \mathcal{R}} \left\{ \exp\left( \mathbf{x}_t^{(i)\top} \mathbf{W}^\top \mathbf{p}(\tau) \right) \exp\left( \mathbf{x}_t^{(i)\top} \mathbf{W}^\top \left( -\alpha \nabla_\mathbf{p} \widehat{\mathcal{L}}\left( \mathbf{p}(\tau) \right) \right) \right) \right\}}{\exp\left( \mathbf{x}_u^{(i)\top} \mathbf{W}^\top \mathbf{p}(\tau) \right) \exp\left( \mathbf{x}_u^{(i)\top} \mathbf{W}^\top \left( -\alpha \nabla_\mathbf{p} \widehat{\mathcal{L}}\left( \mathbf{p}(\tau) \right) \right) \right)} \tag{159}$$

$$\geq \frac{s^{(i)}(\tau)_{t'}}{s^{(i)}(\tau)_u} \cdot \exp\left( \left( \mathbf{x}_{t'}^{(i)} - \mathbf{x}_u^{(i)} \right)^\top \mathbf{W}^\top \left( -\alpha \nabla_\mathbf{p} \widehat{\mathcal{L}}\left( \mathbf{p}(\tau) \right) \right) \right), \tag{160}$$

where $t' = \arg\max_{t \in \mathcal{R}} \left\{ s^{(i)}(\tau)_t \right\}$. Under the induction hypothesis at time step $\tau$, if we can show the inside the exponential of Eq.160 is positive, then the right-hand side of this equation becomes greater than 1, which proves the induction at $\tau+1$.

To begin with, note that we only need to consider the case where $\sum_{t \in \mathcal{R}} s^{(i)}(\tau)_t < 1 - 1/(4T)$. This is because, from Lemma D.4, the token probability can only change at most by a factor between $1/2$ and 2 in one step of gradient descent under the step size assumption. When $\sum_{t \in \mathcal{R}} s^{(i)}(\tau)_t > 1 - 1/(4T)$ holds, the probability of not selecting the relevant token is less than $1/(4T)$. Using Lemma D.4, the probability of not selecting relevant tokens after a single step is at most $1/(2T)$ in total. Therefore, $\sum_{t \in \mathcal{R}} s^{(i)}(\tau+1)_t > 1 - 1/(2T)$ holds, meaning that

$$s^{(i)}(\tau+1)_u < \frac{1}{2T} < \frac{1}{\zeta_\mathcal{R} T}\left( 1 - \frac{1}{2T} \right) < \max_{t \in \mathcal{R}} \left\{ s^{(i)}(\tau+1)_t \right\}, \tag{161}$$

where recall that $\zeta_\mathcal{R} \in (0,1)$ is a ratio of the relevant tokens. Thus, the inductive hypothesis holds at $\tau+1$-th step in this case.

In the rest of the proof, suppose that we have $\sum_{t \in \mathcal{R}} s^{(i)}(\tau)_t < 1 - 1/(4T)$. Additionally, we have $s^{(i)}(\tau)_{t'} \geq 1/T$ because if we assume that it does not hold, then it contradicts that $s^{(i)}(\tau)_{t'}$ is the maximum probability at $\tau$-th step, which is derived from induction hypothesis at time step $\tau$. Recall that $t' \in \mathcal{R}$ and $u \in \mathcal{W} \cup \mathcal{I}$, and from Eq.128, the inside the exponential in Eq. 160 becomes

$$\left( \mathbf{x}_{t'}^{(i)} - \mathbf{x}_u^{(i)} \right)^\top \mathbf{W}^\top \left( -\alpha \nabla_\mathbf{p} \widehat{\mathcal{L}}(\mathbf{p}(\tau)) \right)$$

$$\geq \left( \Delta\rho_{i,t'}(\tau) - \Delta\rho_{i,u}(\tau) \right) \Theta\left( \text{Tr}(\boldsymbol{\Sigma}) \right) - |\Delta\lambda_{Y^{(i)}}(\tau)| \|\boldsymbol{\mu}\|_2^2 - \rho|\Delta\lambda_{-Y^{(i)}}(\tau)| \|\boldsymbol{\mu}\|_2^2$$

$$- \Theta\left( \left( |\Delta\lambda_{+1}(\tau)| + |\Delta\lambda_{-1}(\tau)| \right) \cdot \|\boldsymbol{\mu}\|_2 \sqrt{(\log(Tn/\delta)} \right)$$

$$- \Theta\left( \left( \sum_{k \in [n], u \in [T]} |\Delta\rho_{k,u}(\tau)| \right) \cdot \sqrt{\text{Tr}(\boldsymbol{\Sigma})} \log(Tn/\delta) \right), \tag{162}$$

which follows from the definition of the data model and the high-probability events in Lemma A.1 that occur in a good run. We will show that the first term is dominant, resulting in the positive left-hand side. Using Corollary D.1 and $\|\boldsymbol{\nu}\|_2 = \Theta(1/\|\boldsymbol{\mu}\|_2)$, if $u \in \mathcal{I}$, then we have

$$\left( \Delta\rho_{i,t'}(\tau) - \Delta\rho_{i,u}(\tau) \right) \Theta\left( \text{Tr}(\boldsymbol{\Sigma}) \right)$$

$$\sim \frac{\alpha}{n} \cdot \left\{ s^{(i)}(\tau)_{t'}\left( 1 - \sum_{t \in \mathcal{R}} s^{(i)}(\tau)_t \right) \right.$$

$$\left. + s^{(i)}(\tau)_u\left( \sum_{t \in \mathcal{R}} s^{(i)}(\tau)_t + \rho \sum_{t \in \mathcal{W}_{Y^{(i)}}^{(i)}} s^{(i)}(\tau)_t - \rho \sum_{t \in \mathcal{W}_{-Y^{(i)}}^{(i)}} s^{(i)}(\tau)_t \right) \right\} \text{Tr}(\boldsymbol{\Sigma}) \tag{163}$$

$$\gtrsim \frac{\alpha}{n} \cdot \left( s^{(i)}(\tau)_{t'}\left( 1 - \sum_{t \in \mathcal{R}} s^{(i)}(\tau)_t \right) \right) \text{Tr}(\boldsymbol{\Sigma}) \tag{164}$$

$$\gtrsim \frac{\alpha}{T^2 n} \text{Tr}(\boldsymbol{\Sigma}), \tag{165}$$

where the second last inequality Eq.164 follows from the fact that since $s^{(i)}(\tau)_{t'}$ is the maximum probability, leading to $s^{(i)}(\tau)_{t'} \geq 1/T$, the token $t'$ belongs to $\mathcal{R}$, and the assumption (A3) implies $\rho \leq 1/(CT)$, the second term in Eq.163 becomes positive. The last line follows from $1/T \leq s^{(i)}(\tau)_{t'}$ and $\sum_{t \in \mathcal{R}} s^{(i)}(\tau)_t < 1 - 1/(4T)$, leading to $s^{(i)}(\tau)_{t'} \left(1 - \sum_{t \in \mathcal{R}} s^{(i)}(\tau)_t\right) \gtrsim 1/T^2$. At this time, note that from Corollary D.1, the coefficient of the small order term $\Theta(\sqrt{\log(Tn/\delta)}/\|\boldsymbol{\mu}\|_2)$ in $\Delta\rho$ is at most $T$ times the coefficient of $\Theta(1)$. Therefore, from Eq.46, which is derived by assumption (A'2), we have $\sqrt{\log(Tn/\delta)}/\|\boldsymbol{\mu}\|_2 < 1/(CTn)$, meaning that we can ignore this small order term.

We show that the same order of lower bound as Eq.165 is obtained when $u \in \mathcal{W}$. Since we have a smaller lower bound for $u \in \mathcal{W}^{(i)}_{Y^{(i)}}$ than $u \in \mathcal{W}^{(i)}_{-Y^{(i)}}$, we only consider $\mathcal{W}^{(i)}_{Y^{(i)}}$ case and get

$$-\Delta\rho_{i,u}(\tau) \sim \frac{\alpha}{n} \cdot \left( s^{(i)}(\tau)_u \left( (1-\rho) \sum_{t \in \mathcal{R}} s^{(i)}(\tau)_t - 2\rho \sum_{t \in \mathcal{W}^{(i)}_{-Y^{(i)}}} s^{(i)}(\tau)_t - \rho \sum_{t \in \mathcal{I}} s^{(i)}(\tau)_t \right) \right) \tag{166}$$

$$\geq \frac{\alpha}{n} \cdot \left( s^{(i)}(\tau)_u \left( \left( (1-\rho) - 2\rho \left( |\mathcal{W}^{(i)}_{-Y^{(i)}}| + |\mathcal{I}| \right) \right) s^{(i)}(\tau)_{t'} \right) \right) > 0, \tag{167}$$

where the last inequality follows from the assumption (A3) $\rho \leq 1/(CT)$. Therefore, we can get the same lower bound as Eq.165 for $u \in \mathcal{W}$.

Furthermore, we will bound the other terms in Eq.162. Corollary D.1 and Lemma D.5 gives us the upper bound for other terms:

$$|\Delta\lambda_{+1}(\tau)| \lesssim \alpha, \quad |\Delta\lambda_{-1}(\tau)| \lesssim \alpha, \quad \sum_{k,u} |\Delta\rho_{k,u}(\tau)| \lesssim \alpha. \tag{168}$$

Using the assumption (A'1) and Eq.45 in Lemma A.8, which implies $\mathrm{Tr}(\boldsymbol{\Sigma}) \geq CT^2 n \|\boldsymbol{\mu}\|_2^2$ and $\sqrt{\mathrm{Tr}(\boldsymbol{\Sigma})} \geq CT^2 n \log(Tn/\delta)$, we can see that the first noise memorization term in Eq. 162 is dominant, which concludes that the inside the exponential in Eq.160 is greater than 1. Combining this and the induction hypothesis provides $s^{(i)}(\tau+1)_u \leq \max_{t \in \mathcal{R}}\{s^{(i)}(\tau+1)_t\}$, for any weakly relevant or irrelevant token $u \in \mathcal{W} \cup \mathcal{I}$. $\qquad\square$

**Remark 5.** We explain the reason for considering the worst-case scenario where the signal update $\Delta\lambda(\tau)$ becomes negative, as in Eq. 162. In the initial training phase, the signal learning progresses in a positive direction because of the noise ratio assumption, assigning more probability to the relevant token of the clean data. If we can show that *for all clean data $i \in \mathcal{C}$, $\max_{t \in \mathcal{R}}\left\{s^{(i)}(\tau)_t\right\}$* becomes a constant order after this initial phase, for example, greater than $1/2$ instead of around the initial value $\Theta(1/T)$, it would be possible to prove Lemma A.3 under the weaker strength of memorization, $\mathrm{Tr}(\boldsymbol{\Sigma}) \geq CTn\|\boldsymbol{\mu}\|_2^2$. However, considering the case where most clean data pick the relevant token very quickly (i.e., $\sum_{t \in \mathcal{R}} s^{(i)}(\tau)_t \gg 1 - 1/T$), while some clean data is learned slowly (i.e., $\sum_{t \in \mathcal{R}} s^{(i')}(\tau)_t = \Theta(|\mathcal{R}|/T)$), the signal update might be a negative direction because of the discussion in Section 4.3. This leads to a situation where the relevant token will not be picked for these slowly learned clean data. Therefore, we proceed with the inequality $\mathrm{Tr}(\boldsymbol{\Sigma}) \geq CT^2 n\|\boldsymbol{\mu}\|_2^2$ to ensure that the relevant token is selected for all clean data regardless of the positive or negative of the signal update.

## D.4 PROOF OF LEMMA A.4

In this section, using proof similar to the previous lemma, we will show that the model picks weakly relevant tokens for noisy data with a signal corresponding to the noisy labels.

**Lemma A.4.** *Suppose that the assumptions (A'1)-(A'3), (A4)-(A6) hold, and the norm of the linear head $\boldsymbol{\nu} \propto \boldsymbol{\mu}_{+1} - \boldsymbol{\mu}_{-1}$ scales as $\Theta(1/\|\boldsymbol{\mu}\|_2)$. For any noisy data $j \in \mathcal{N}$, on a good run, for all time step $\tau \geq 0$ and all tokens except for the weakly relevant tokens that align with the label noise: $t \in [T] \setminus \mathcal{W}^{(j)}_{Y^{(j)}}$, we have*

$$s^{(j)}(\tau)_t \leq \max_{u \in \mathcal{W}^{(j)}_{Y^{(j)}}} \left\{ s^{(j)}(\tau)_u \right\}. \tag{27}$$

*Proof.* We essentially proceed with the same proof as Lemma A.3 and use an induction argument. At initialization, the equality holds with $1/T$. In the rest of the proof, let $t$ be in $[T] \setminus \mathcal{W}_{Y^{(j)}}^{(j)}$. Since we have

$$\frac{\max_{u \in \mathcal{W}_{Y^{(j)}}^{(j)}} \left\{ s^{(j)}(\tau+1)_u \right\}}{s^{(j)}(\tau+1)_t} \geq \frac{s^{(j)}(\tau)_{u'}}{s^{(j)}(\tau)_t} \cdot \exp\left( \left(\mathbf{x}_{u'}^{(j)} - \mathbf{x}_t^{(j)}\right)^\top \mathbf{W}^\top \left(-\alpha \nabla_{\mathbf{p}} \widehat{\mathcal{L}}\left(\mathbf{p}(\tau)\right)\right) \right), \tag{169}$$

where $u' = \arg\max_{u \in \mathcal{W}_{Y^{(j)}}^{(j)}} \left\{ s^{(j)}(\tau)_u \right\}$. The goal below is to show that the inside exponential of this equation becomes positive because with the induction hypothesis at time step $\tau$, it proves the induction at $\tau + 1$. Here, note that we only need to consider the case where $\sum_{u \in \mathcal{W}_{Y^{(j)}}^{(j)}} s^{(j)}(\tau)_u < 1 - 1/(4T)$ because otherwise, the induction still holds at time step $\tau + 1$ from Lemma D.4 by the same discussion as in Lemma A.3. Therefore, in the following proof, we assume $\sum_{u \in \mathcal{W}_{Y^{(j)}}^{(j)}} s^{(j)}(\tau)_u < 1 - 1/(4T)$, and by the induction hypothesis at time step $\tau$, we have $s^{(j)}(\tau)_{u'} \geq 1/T$. We will consider three cases based on the membership of $t \in [T] \setminus \mathcal{W}_{Y^{(j)}}^{(j)}$.

**Case 1:** Member of relevant token: $t \in \mathcal{R}$.
From the high-probability events in Lemma A.1, the inside the exponential in Eq. 169 becomes

$$\left(\mathbf{x}_{u'}^{(j)} - \mathbf{x}_t^{(j)}\right)^\top \mathbf{W}^\top \left(-\alpha \nabla_{\mathbf{p}} \widehat{\mathcal{L}}(\mathbf{p}(\tau))\right)$$
$$\geq \left(\Delta\rho_{j,u'}(\tau) - \Delta\rho_{j,t}(\tau)\right) \Theta\left(\mathrm{Tr}(\boldsymbol{\Sigma})\right) - \rho|\Delta\lambda_{Y^{(j)}}(\tau)| \|\boldsymbol{\mu}\|_2^2 - |\Delta\lambda_{-Y^{(j)}}(\tau)| \|\boldsymbol{\mu}\|_2^2$$
$$- \Theta\left( \left(|\Delta\lambda_{+1}(\tau)| + |\Delta\lambda_{-1}(\tau)|\right) \cdot \|\boldsymbol{\mu}\|_2 \sqrt{(\log(Tn/\delta))} \right)$$
$$- \Theta\left( \left(\sum_{k \in [n], u \in [T]} |\Delta\rho_{k,u}(\tau)|\right) \cdot \sqrt{\mathrm{Tr}(\boldsymbol{\Sigma})} \log(Tn/\delta) \right). \tag{170}$$

We will show that the first noise memorization term is dominant. Using Corollary D.1 and $\|\boldsymbol{\nu}\|_2 = \Theta(1/\|\boldsymbol{\mu}\|_2)$ gives us:

$$\left(\Delta\rho_{j,u'}(\tau) - \Delta\rho_{j,t}(\tau)\right) \Theta\left(\mathrm{Tr}(\boldsymbol{\Sigma})\right)$$
$$\sim \frac{\alpha}{n} \cdot \left\{ s^{(j)}(\tau)_{u'} \left( \sum_{t' \in \mathcal{R}} s^{(j)}(\tau)_{t'} + \rho \left( 2 \sum_{u \in \mathcal{W}_{-Y^{(j)}}^{(j)}} s^{(j)}(\tau)_u + \sum_{u \in \mathcal{R} \cup \mathcal{I}} s^{(j)}(\tau)_u \right) \right) \right.$$
$$\left. + s^{(j)}(\tau)_t \left(1 - \sum_{t' \in \mathcal{R}} s^{(j)}(\tau)_{t'}\right) \right\} \mathrm{Tr}(\boldsymbol{\Sigma}). \tag{171}$$

Now we only need to handle the case where $\sum_{t' \in \mathcal{R}} s^{(j)}(\tau)_{t'} > 1/(4T)$ because if we assume otherwise, then again by Lemma D.4, we have

$$s^{(j)}(\tau+1)_t \leq \sum_{t' \in \mathcal{R}} s^{(j)}(\tau+1)_{t'} < \frac{1}{2T} \leq s^{(j)}(\tau+1)_{u'}, \tag{172}$$

which establishes the induction at time step $\tau + 1$. Using $s^{(j)}(\tau)_{u'} \geq 1/T$ and $\sum_{t' \in \mathcal{R}} s^{(j)}(\tau)_{t'} > 1/(4T)$, Eq.171 becomes

$$\left(\Delta\rho_{j,u'}(\tau) - \Delta\rho_{j,t}(\tau)\right) \Theta\left(\mathrm{Tr}(\boldsymbol{\Sigma})\right) \gtrsim \frac{\alpha}{T^2 n} \mathrm{Tr}(\boldsymbol{\Sigma}), \tag{173}$$

where we used the fact that the second and third terms in Eq.171 are positive.

Since Corollary D.1 and Lemma D.5 provides

$$|\Delta\lambda_{+1}(\tau)| \lesssim \alpha, \quad |\Delta\lambda_{-1}(\tau)| \lesssim \alpha, \quad \sum_{k,u} |\Delta\rho_{k,u}(\tau)| \lesssim \alpha, \tag{174}$$

by combining Eq.173 and the assumption (A'1) and Eq.45, which implies $\mathrm{Tr}(\boldsymbol{\Sigma}) \geq CT^2 n \|\boldsymbol{\mu}\|_2^2$ and $\sqrt{\mathrm{Tr}(\boldsymbol{\Sigma})} \geq CT^2 n \log(Tn/\delta)$, we can see that the first noise memorization term in Eq. 170 is dominant, and the inside exponential becomes positive.

**Case 2:** Member of other weakly relevant token: $t \in \mathcal{W} \setminus \mathcal{W}_{Y^{(j)}}^{(j)} = \mathcal{W}_{-Y^{(j)}}^{(j)}$.
The inside the exponential in Eq. 169 becomes

$$
\left(\mathbf{x}_{u'}^{(j)} - \mathbf{x}_t^{(j)}\right)^\top \mathbf{W}^\top \left(-\alpha \nabla_{\mathbf{p}} \widehat{\mathcal{L}}(\mathbf{p}(\tau))\right)
$$

$$
\geq (\Delta \rho_{j,u'}(\tau) - \Delta \rho_{j,t}(\tau)) \, \Theta \left(\mathrm{Tr}(\boldsymbol{\Sigma})\right) - \rho \left(|\Delta \lambda_{+1}(\tau)| + |\Delta \lambda_{-1}(\tau)|\right) \|\boldsymbol{\mu}\|_2^2
$$

$$
- \Theta \left(\left(|\Delta \lambda_{+1}(\tau)| + |\Delta \lambda_{-1}(\tau)|\right) \cdot \|\boldsymbol{\mu}\|_2 \sqrt{(\log(Tn/\delta))}\right)
$$

$$
- \Theta \left(\left(\sum_{k \in [n], u \in [T]} |\Delta \rho_{k,u}(\tau)|\right) \cdot \sqrt{\mathrm{Tr}(\boldsymbol{\Sigma})} \log(Tn/\delta)\right). \tag{175}
$$

Note that the $\|\boldsymbol{\mu}\|_2^2$ term is different from Eq.170. Using Corollary D.1 and $\|\boldsymbol{\nu}\|_2 = \Theta(1/\|\boldsymbol{\mu}\|_2)$ gives us:

$$
(\Delta \rho_{j,u'}(\tau) - \Delta \rho_{j,t}(\tau)) \, \Theta \left(\mathrm{Tr}(\boldsymbol{\Sigma})\right)
$$

$$
\gtrsim \frac{\alpha}{n} \cdot \left\{ s^{(j)}(\tau)_{u'} \left( \sum_{t' \in \mathcal{R}} s^{(j)}(\tau)_{t'} + \rho \left( 1 - \sum_{u \in \mathcal{W}_{Y^{(j)}}^{(j)}} s^{(j)}(\tau)_u \right) \right) \right.
$$

$$
+ s^{(j)}(\tau)_t \left( -\sum_{t' \in \mathcal{R}} s^{(j)}(\tau)_{t'} + \rho \left( 1 - \sum_{u \in \mathcal{W}_{-Y^{(j)}}^{(j)}} s^{(j)}(\tau)_u \right) \right)
$$

$$
\left. - \left( s^{(j)}(\tau)_{u'} \left( 1 - s^{(j)}(\tau)_{u'} \right) + s^{(j)}(\tau)_t \left( 1 - s^{(j)}(\tau)_t \right) \right) \frac{\sqrt{\log(Tn/\delta)}}{\|\boldsymbol{\mu}\|_2} \right\} \mathrm{Tr}(\boldsymbol{\Sigma}). \tag{176}
$$

Here, the term multiplied $\rho$ can be dominant, unlike in Case 1, so we have to account for the lower order terms $\sqrt{\log(Tn/\delta)}/\|\boldsymbol{\mu}\|_2$ in Corollary D.1. Since we have $s^{(j)}(\tau)_{u'} \geq s^{(j)}(\tau)_t$ from the induction hypothesis, the sum of the first and third terms: $(s^{(j)}(\tau)_{u'} - s^{(j)}(\tau)_t) \sum_{t' \in \mathcal{R}} s^{(j)}(\tau)_{t'}$ is positive. Now we have $s^{(j)}(\tau)_{u'} \geq 1/T$ and $\sum_{u \in \mathcal{W}_{Y^{(j)}}^{(j)}} s^{(j)}(\tau)_u < 1 - (4T)$; therefore, the lower bound of the second term is given by

$$
s^{(j)}(\tau)_{u'} \cdot \rho \left( 1 - \sum_{u \in \mathcal{W}_{Y^{(j)}}^{(j)}} s^{(j)}(\tau)_u \right) \gtrsim \frac{\rho}{T^2}. \tag{177}
$$

As for the fifth and sixth terms, by using the assumption (A'3): $\rho \geq CT\sqrt{\log(Tn/\delta)}/\|\boldsymbol{\mu}\|_2$ and Lemma D.6, we can bound the effect of these small order terms by the second term as:

$$
\left( s^{(j)}(\tau)_{u'} \left( 1 - s^{(j)}(\tau)_{u'} \right) + s^{(j)}(\tau)_t \left( 1 - s^{(j)}(\tau)_t \right) \right) \frac{\sqrt{\log(Tn/\delta)}}{\|\boldsymbol{\mu}\|_2}
$$

$$
\leq 2 s^{(j)}(\tau)_{u'} \left( 1 - s^{(j)}(\tau)_{u'} \right) \frac{\sqrt{\log(Tn/\delta)}}{\|\boldsymbol{\mu}\|_2} \tag{178}
$$

$$
\leq 2 s^{(j)}(\tau)_{u'} \left( 1 - \sum_{u \in \mathcal{W}_{Y^{(j)}}^{(j)}} s^{(j)}(\tau)_u \right) \frac{4T\sqrt{\log(Tn/\delta)}}{\|\boldsymbol{\mu}\|_2} \tag{179}
$$

$$
\leq s^{(j)}(\tau)_{u'} \cdot \rho \left( 1 - \sum_{u \in \mathcal{W}_{Y^{(j)}}^{(j)}} s^{(j)}(\tau)_u \right), \tag{180}
$$

where the second inequality follows from $1 - \sum_{u \in \mathcal{W}_{Y^{(j)}}^{(j)}} s^{(j)}(\tau)_u \geq 1/(4T)$. Consequently, the second term in Eq. 176 becomes dominant, and we have

$$
(\Delta \rho_{j,u'}(\tau) - \Delta \rho_{j,t}(\tau)) \, \Theta \left(\mathrm{Tr}(\boldsymbol{\Sigma})\right) \gtrsim \frac{\alpha \rho}{T^2 n} \mathrm{Tr}(\boldsymbol{\Sigma}). \tag{181}
$$

Eq. 174 provides the upper bound of other terms in Eq.175 as follows:

$$\rho \left( |\Delta\lambda_{+1}(\tau)| + |\Delta\lambda_{-1}(\tau)| \right) \|\boldsymbol{\mu}\|_2^2 \lesssim \alpha\rho\|\boldsymbol{\mu}\|_2^2, \tag{182}$$

$$\left( |\Delta\lambda_{+1}(\tau)| + |\Delta\lambda_{-1}(\tau)| \right) \cdot \|\boldsymbol{\mu}\|_2 \sqrt{(\log(Tn/\delta)} \lesssim \alpha \|\boldsymbol{\mu}\|_2 \sqrt{\log(Tn/\delta)} \lesssim \alpha\rho\|\boldsymbol{\mu}\|_2^2, \tag{183}$$

$$\left( \sum_{k\in[n],u\in[T]} |\Delta\rho_{k,u}(\tau)| \right) \cdot \sqrt{\text{Tr}(\boldsymbol{\Sigma})} \log(Tn/\delta) \lesssim \alpha \sqrt{\text{Tr}(\boldsymbol{\Sigma})} \log(Tn/\delta) \lesssim \frac{\alpha\rho}{T^2 n} \text{Tr}(\boldsymbol{\Sigma}), \tag{184}$$

where we used the assumption (A'3), which implies $C\sqrt{\log(Tn/\delta)}/\|\boldsymbol{\mu}\|_2 \le \rho$, in the second line. The last line follows from Eq.45, which is derived from the assumption (A'1), i.e., $\sqrt{\text{Tr}(\boldsymbol{\Sigma})} \ge CT^2 n \log(Tn/\delta)/\rho$. From the assumption (A'1), which leads to $\text{Tr}(\boldsymbol{\Sigma}) \ge CT^2 n\|\boldsymbol{\mu}\|_2^2$, we can show that the noise memorization term Eq.181 becomes dominant.

**Case 3:** Member of irrelevant token: $t \in \mathcal{I}$.
We repeat the same discussion as in Case 2, so we only discuss the different parts. The noise memorization term becomes

$$\left( \Delta\rho_{j,u'}(\tau) - \Delta\rho_{j,t}(\tau) \right) \Theta\left( \text{Tr}(\boldsymbol{\Sigma}) \right)$$

$$\gtrsim \frac{\alpha}{n} \cdot \left\{ s^{(j)}(\tau)_{u'} \left( \sum_{t'\in\mathcal{R}} s^{(j)}(\tau)_{t'} + \rho \left( 2 \sum_{u\in\mathcal{W}_{-Y^{(j)}}^{(j)}} s^{(j)}(\tau)_u + \sum_{u\in\mathcal{R}\cup\mathcal{I}} s^{(j)}(\tau)_u \right) \right) \right.$$

$$+ s^{(j)}(\tau)_t \left( -\sum_{t'\in\mathcal{R}} s^{(j)}(\tau)_{t'} + \rho \left( \sum_{u\in\mathcal{W}_{Y^{(j)}}^{(j)}} s^{(j)}(\tau)_u - \sum_{u\in\mathcal{W}_{-Y^{(j)}}^{(j)}} s^{(j)}(\tau)_u \right) \right)$$

$$\left. - \left( s^{(j)}(\tau)_{u'} \left( 1 - s^{(j)}(\tau)_{u'} \right) + s^{(j)}(\tau)_t \left( 1 - s^{(j)}(\tau)_t \right) \right) \frac{\sqrt{\log(Tn/\delta)}}{\|\boldsymbol{\mu}\|_2} \right\} \text{Tr}(\boldsymbol{\Sigma}). \tag{185}$$

Since we have $s^{(j)}(\tau)_{u'} \ge s^{(j)}(\tau)_t$ from the induction hypothesis, we have

$$s^{(j)}(\tau)_{u'} \left( \sum_{t'\in\mathcal{R}} s^{(j)}(\tau)_{t'} + \rho \left( 2 \sum_{u\in\mathcal{W}_{-Y^{(j)}}^{(j)}} s^{(j)}(\tau)_u + \sum_{u\in\mathcal{R}\cup\mathcal{I}} s^{(j)}(\tau)_u \right) \right)$$

$$+ s^{(j)}(\tau)_t \left( -\sum_{t'\in\mathcal{R}} s^{(j)}(\tau)_{t'} + \rho \left( \sum_{u\in\mathcal{W}_{Y^{(j)}}^{(j)}} s^{(j)}(\tau)_u - \sum_{u\in\mathcal{W}_{-Y^{(j)}}^{(j)}} s^{(j)}(\tau)_u \right) \right)$$

$$\ge \rho \cdot s^{(j)}(\tau)_{u'} \left( 1 - \sum_{u\in\mathcal{W}_{Y^{(j)}}^{(j)}} s^{(j)}(\tau)_u \right) + \rho \cdot s^{(j)}(\tau)_t \left( \sum_{u\in\mathcal{W}_{Y^{(j)}}^{(j)}} s^{(j)}(\tau)_u \right) \tag{186}$$

$$\ge \rho \cdot s^{(j)}(\tau)_{u'} \left( 1 - \sum_{u\in\mathcal{W}_{Y^{(j)}}^{(j)}} s^{(j)}(\tau)_u \right), \tag{187}$$

where the first inequality follows from $\left( s^{(j)}(\tau)_{u'} - s^{(j)}(\tau)_t \right) \sum_{t'\in\mathcal{R}} s^{(j)}(\tau)_{t'} \ge 0$ and $s^{(j)}(\tau)_{u'} \sum_{u\in\mathcal{W}_{-Y^{(j)}}^{(j)}} s^{(j)}(\tau)_u \ge s^{(j)}(\tau)_t \sum_{u\in\mathcal{W}_{-Y^{(j)}}^{(j)}} s^{(j)}(\tau)_u$. The remainder follows the same reasoning as in Case 2, concluding that the noise memorization term is dominant.

By combining the above three cases, we conclude that the inside exponential of Eq.169 is positive, leading to the induction at time step $\tau + 1$. Therefore, we conclude the statement. $\qquad\square$

## D.5 MINOR TECHNICAL LEMMA

We will see that with the assumption of a sufficiently small step size, the softmax probabilities do not change significantly in a single step of gradient descent.

**Lemma D.4.** *Suppose that the norm of the linear head $\boldsymbol{\nu} \propto \boldsymbol{\mu}_{+1} - \boldsymbol{\mu}_{-1}$ scales as $\Theta(1/\|\boldsymbol{\mu}\|_2)$, and the step size of gradient descent is small enough: $\alpha \lesssim n/\operatorname{Tr}(\boldsymbol{\Sigma})$. Then, the probability assigned to each token only changes at most by a constant factor; in other words, we have*

$$\forall \tau \geq 0, \ \frac{1}{2} s^{(i)}(\tau)_t < s^{(i)}(\tau+1)_t < 2 s^{(i)}(\tau)_t, \tag{188}$$

*for all $i \in [n], t \in [T]$.*

*Proof.* By the update of gradient descent, we have

$$s^{(i)}(\tau+1)_t = \frac{\exp\left(\mathbf{x}_t^{(i)\top}\mathbf{W}^\top \mathbf{p}(\tau)\right)\exp\left(\mathbf{x}_t^{(i)\top}\mathbf{W}^\top\left(-\alpha\nabla_{\mathbf{p}}\widehat{\mathcal{L}}(\mathbf{p}(\tau))\right)\right)}{\sum_{u\in[T]}\exp\left(\mathbf{x}_u^{(i)\top}\mathbf{W}^\top \mathbf{p}(\tau)\right)\exp\left(\mathbf{x}_u^{(i)\top}\mathbf{W}^\top\left(-\alpha\nabla_{\mathbf{p}}\widehat{\mathcal{L}}(\mathbf{p}(\tau))\right)\right)}. \tag{189}$$

Then, we have

$$\frac{s^{(i)}(\tau+1)_t}{s^{(i)}(\tau)_t}$$
$$\leq \exp\left(\mathbf{x}_t^{(i)\top}\mathbf{W}^\top\left(-\alpha\nabla_{\mathbf{p}}\widehat{\mathcal{L}}(\mathbf{p}(\tau))\right)\right)\left(\min_{u\in[T]}\left\{\exp\left(\mathbf{x}_u^{(i)\top}\mathbf{W}^\top\left(-\alpha\nabla_{\mathbf{p}}\widehat{\mathcal{L}}(\mathbf{p}(\tau))\right)\right)\right\}\right)^{-1}, \tag{190}$$

$$\frac{s^{(i)}(\tau+1)_t}{s^{(i)}(\tau)_t}$$
$$\geq \exp\left(\mathbf{x}_t^{(i)\top}\mathbf{W}^\top\left(-\alpha\nabla_{\mathbf{p}}\widehat{\mathcal{L}}(\mathbf{p}(\tau))\right)\right)\left(\max_{u\in[T]}\left\{\exp\left(\mathbf{x}_u^{(i)\top}\mathbf{W}^\top\left(-\alpha\nabla_{\mathbf{p}}\widehat{\mathcal{L}}(\mathbf{p}(\tau))\right)\right)\right\}\right)^{-1}. \tag{191}$$

Therefore, it suffices to show that

$$\frac{1}{2} < \exp\left(\left(\mathbf{x}_t^{(i)} - \mathbf{x}_u^{(i)}\right)^\top\mathbf{W}^\top\left(-\alpha\nabla_{\mathbf{p}}\widehat{\mathcal{L}}(\mathbf{p}(\tau))\right)\right) < 2, \tag{192}$$

for any token $u \in [T]$. The inside exponential of this is bounded as:

$$\left|\left(\mathbf{x}_t^{(i)} - \mathbf{x}_u^{(i)}\right)^\top\mathbf{W}^\top\left(-\alpha\nabla_{\mathbf{p}}\widehat{\mathcal{L}}(\mathbf{p}(\tau))\right)\right|$$
$$\leq 2\max_{u\in[T]}\left\{\left|\mathbf{x}_u^{(i)\top}\mathbf{W}^\top\left(-\alpha\nabla_{\mathbf{p}}\widehat{\mathcal{L}}(\mathbf{p}(\tau))\right)\right|\right\} \tag{193}$$
$$\lesssim |\Delta\rho_{i,u}(\tau)|\operatorname{Tr}(\boldsymbol{\Sigma}) + |\Delta\lambda(\tau)|\|\boldsymbol{\mu}\|_2^2$$
$$+ \left(|\Delta\lambda_{+1}(\tau)| + |\Delta\lambda_{-1}(\tau)|\right)\|\boldsymbol{\mu}\|_2\sqrt{\log(Tn/\delta)} + \sum_{k,u}|\Delta\rho_{k,u}(\tau)|\sqrt{\operatorname{Tr}(\boldsymbol{\Sigma})}\log(Tn/\delta), \tag{194}$$

where the last line follows from Eq.128 and the high-probability events in Lemma A.1. Here, from Corollary D.1, Lemma D.2, and $\|\boldsymbol{\nu}\|_2 = \Theta(1/\|\boldsymbol{\mu}\|_2)$, each coefficient update is at most the following order:

$$|\Delta\lambda_{+1}(\tau)| \lesssim \alpha, \ |\Delta\lambda_{-1}(\tau)| \lesssim \alpha, \ |\Delta\rho_{i,t}(\tau)| \lesssim \alpha/n, \ \sum_{k,u}|\Delta\rho_{k,u}(\tau)| \lesssim \alpha, \tag{195}$$

where the last one follows from Lemma D.5. By assumption on the step size $\alpha \lesssim n/\operatorname{Tr}(\boldsymbol{\Sigma})$ and assumption (A'1), the upper-bound in Eq. 194 can be bounded with $\Theta(1)$. Therefore, choosing $\alpha$ sufficiently small can make it smaller than $\log 2$, thus satisfying Eq. 192. It concludes the proof. $\square$

Next, we evaluate the maximum updates of noise terms, which are helpful in analyzing the influence of small-order terms in the dynamics analysis. By using Corollary D.1, we obtain a naive evaluation for $\sum_{i\in[n],t\in[T]}|\Delta\rho_{i,t}(\tau)| \lesssim Tn\max|\Delta\rho|$, summing over $i\in[n]$ and $t\in[T]$. However, a more detailed analysis of the softmax probability gives us a tighter upper bound without the dependence on $T$.

**Lemma D.5.** *On a good run, for any time step $\tau \geq 0$, we have*

$$\sum_{i\in[n],t\in[T]}|\Delta\rho_{i,t}(\tau)| \lesssim \alpha\|\boldsymbol{\nu}\|_2\|\boldsymbol{\mu}\|_2. \tag{196}$$

*Proof.* From Corollary D.1, we proceed by summing up each two order term: $\|\boldsymbol{\mu}\|_2$ and $\sqrt{\log(Tn/\delta)}$ in the noise update $\Delta\rho_{i,t}(\tau)$.

We first examine $\Theta(\|\boldsymbol{\mu}\|_2)$ term. The softmax probabilities in the upper bound are given by: $s_t(1-\sum_{t'\in\mathcal{R}}s_{t'})$ for the relevant token $t\in\mathcal{R}$, $s_u(\sum_{t\in\mathcal{R}}s_t + 2\rho\sum_{t\in\mathcal{W}_{-Y^{(i)}}^{(i)}}s_t + \rho\sum_{t\in\mathcal{R}\cup\mathcal{I}}s_t)$ and $s_{u'}(\sum_{t\in\mathcal{R}}s_t + 2\rho\sum_{t\in\mathcal{W}_{Y^{(i)}}^{(i)}}s_t + \rho\sum_{t\in\mathcal{R}\cup\mathcal{I}}s_t)$ for the weakly relevant token $u\in\mathcal{W}_{Y^{(i)}}^{(i)}, u'\in\mathcal{W}_{-Y^{(i)}}^{(i)}$, and $s_v\left(\sum_{t\in\mathcal{R}}s_t + \rho\sum_{t\in\mathcal{W}}s_t\right)$ for irrelevant token $v\in\mathcal{I}$. Thus, we have

$$\sum_{i\in[n]}\frac{\alpha\|\boldsymbol{\nu}\|_2}{\sqrt{2}n}\left\{2\left(\sum_{t\in\mathcal{R}}s^{(i)}(\tau)_t\right)\left(1-\sum_{t\in\mathcal{R}}s^{(i)}(\tau)_t\right) + 4\rho\left(\sum_{t\in\mathcal{W}_{+1}^{(i)}}s^{(i)}(\tau)_t\right)\left(\sum_{t\in\mathcal{W}_{-1}^{(i)}}s^{(i)}(\tau)_t\right)\right.$$

$$\left.+ \rho\left(\sum_{t\in\mathcal{W}}s^{(i)}(\tau)_t\right)\left(\sum_{t\in\mathcal{R}\cup\mathcal{I}}s^{(i)}(\tau)_t\right) + \rho\left(\sum_{t\in\mathcal{W}}s^{(i)}(\tau)_t\right)\left(\sum_{t\in\mathcal{I}}s^{(i)}(\tau)_t\right)\right\}\|\boldsymbol{\mu}\|_2 \tag{197}$$

$$\lesssim \alpha\|\boldsymbol{\nu}\|_2\|\boldsymbol{\mu}\|_2, \tag{198}$$

where we used the fact that the loss derivative becomes constant order by Lemma D.2.

Next, for $\Theta(\sqrt{\log(Tn/\delta)})$ term, the softmax probabilities in the upper bound are $s_t(1-s_t)$ for all token $t\in[T]$, so we have

$$\sum_{i\in[n]}\frac{\alpha\|\boldsymbol{\nu}\|_2}{\sqrt{2}n}\left(\sum_{t\in[T]}s^{(i)}(\tau)_t\left(1-s^{(i)}(\tau)_t\right)\right)\sqrt{\log(Tn/\delta)} \tag{199}$$

$$\lesssim \alpha\|\boldsymbol{\nu}\|_2\sqrt{\log(Tn/\delta)}, \tag{200}$$

where the inequality follows from $\sum_{t\in[T]}s^{(i)}(\tau)_1(1-s^{(i)}(\tau)_1) \leq 1$. Combining Eqs.198 and 200 provides the desired result. $\square$

In the next lemma, we confirm that the significant term in tracking the gradient descent dynamics, $s(1-s)$, is dominated by the token with the highest assigned probability.

**Lemma D.6.** *Let $\mathbf{s}\in\mathbb{R}^T$ be a probability vector, and for $t\in[T]$, $s_t$ is the vector's largest element. Then we have*

$$s_u(1-s_u) \leq s_t(1-s_t), \ \forall u\in[T]\setminus\{t\}. \tag{201}$$

*Proof.* The function $f(x)=x-x^2$ defined on $x\in[0,1]$ is monotonically increasing in $[0,1/2]$, and $f(x)=f(1-x)$ holds because of the symmetry at $x=1/2$. When $0\leq s_t\leq 1/2$, the claim holds because of the monotonicity over $[0,1/2]$. For the remaining case $1/2\leq s_t\leq 1$, since $f(s_t)=f(1-s_t)$ and

$$s_u \leq \sum_{v\in[T]\setminus\{t\}}s_v = 1-s_t \tag{202}$$

hold, the claim follows from the monotonicity over $[0,1/2]$ again. $\square$

# E    PROOF OF THEOREM 4.2

Under the preparation of the lemmas in Section D, we provide the proof of Theorem 4.2 in this section. The proof is divided into two parts: i) we show the gradient descent leads to overfitting (Section E.1), and then ii) we discuss the generalization ability based on the behavior of the softmax probability during the training (Section E.2).

## E.1    OVERFITTING PART

*Proof.* Lemma A.2 gives us that there there exists the token index $t_i^* \in [T]$ for each $i \in [n]$, and we have

$$\lim_{\tau \to \infty} s_{t_i^*}^{(i)}(\tau) = 1, \text{ and } \lim_{\tau \to \infty} s_t^{(i)}(\tau) = 0, \tag{203}$$

for all $i \in [n], t \in [T] \setminus \{t_i^*\}$. In the case of $i \in \mathcal{C}$, Lemma A.3 implies that $t_i^* \in \mathcal{R}$ because if we suppose $t_i^* \notin \mathcal{R}$, then it contradicts to Lemma A.3, which states for any time step $\tau \geq 0$, $s^{(i)}(\tau)_{t_i^*} \leq \max_{t \in \mathcal{R}} s^{(i)}(\tau)_t$. Then, we have

$$Y^{(i)} \cdot f(\mathbf{X}^{(i)}) = Y^{(i)} \cdot \mathbf{x}_{t_i^*}^{(i)\top} \boldsymbol{\nu} \sim 1 \pm 2c_2 \frac{\sqrt{\log(Tn/\delta)}}{\|\boldsymbol{\mu}\|_2} > 0, \tag{204}$$

where the last inequality follows from the assumption (A'2) on the signal strength.

Finally, in the case of $i \in \mathcal{N}$, Lemma A.4 implies that $t_i^* \in \mathcal{W}_{Y^{(i)}}^{(i)}$ by the similar argument. Then, we have

$$Y^{(i)} \cdot f(\mathbf{X}^{(i)}) = Y^{(i)} \cdot \mathbf{x}_{t_i^*}^{(i)\top} \boldsymbol{\nu} \sim \rho \pm 2c_2 \frac{\sqrt{\log(Tn/\delta)}}{\|\boldsymbol{\mu}\|_2} > 0, \tag{205}$$

where we use the assumption (A3) on the signal scale $\rho$ of weakly relevant token: $\rho \geq CT\sqrt{\log(Tn/\delta)}/\|\boldsymbol{\mu}\|_2$. Combining Eqs.204 and 205 concludes that the converged solution fits the training data $\mathcal{S}$ perfectly. $\qquad\square$

## E.2    GENERALIZATION PART

### E.2.1    PRELIMINARY LEMMAS

Before delving into the proof, we introduce the lemma about the direction of the signal learning.

**Lemma E.1** (Direction of Signal Learning). *Suppose that the conditions in Theorem 4.2: Eq.9 and 10 hold. If the training trajectory satisfies Eq.11, then we have*

$$\lambda_c(\tau) \sim \frac{\alpha \|\boldsymbol{\nu}\|_2}{n} \left( \sum_{i \in \mathcal{C}_c} \sum_{0 \leq \tau' \leq \tau} \mathfrak{S}^{(i)}(\tau') \right) \|\boldsymbol{\mu}\|_2 > 0, \tag{206}$$

*for all $c \in \{\pm 1\}$. Furthermore, if the training trajectory satisfies Eq.13, then we have*

$$\lambda_c(\tau) \sim -\frac{\alpha \|\boldsymbol{\nu}\|_2}{n} \left( \sum_{j \in \mathcal{N}_{-c}} \sum_{0 \leq \tau' \leq \tau} \mathfrak{S}^{(j)}(\tau') \right) \|\boldsymbol{\mu}\|_2 < 0, \tag{207}$$

*for all $c \in \{\pm 1\}$.*

*Proof.* The proof is obtained by using $\lambda_c(\tau) = \sum_{0 \leq \tau' \leq \tau} \Delta \lambda_c(\tau')$ and carefully examining the formula for this update in Corollary D.1. We will show that among the eight terms in Eq.129, the first term becomes dominant, leading to Eq.206. Eq.207 can be shown in the same way. We will analyze each term separately.

**Second term:** The difference between the first and second terms is given by:

$$
\frac{\alpha\|\boldsymbol{\nu}\|_2}{\sqrt{2n}} \left( \sum_{i\in\mathcal{C}_c, 0\leq\tau'\leq\tau} (-\ell_i'(\tau))\mathfrak{S}^{(i)}(\tau')(1\pm\rho)\|\boldsymbol{\mu}\|_2 - \sum_{j\in\mathcal{N}_{-c}, 0\leq\tau'\leq\tau} (-\ell_j'(\tau))\mathfrak{S}^{(j)}(\tau')(1\pm\rho)\|\boldsymbol{\mu}\|_2 \right)
$$

$$
\geq \frac{\alpha\|\boldsymbol{\nu}\|_2}{\sqrt{2n}} \min_{i\in\mathcal{C}_c} \{(-\ell_i'(\tau))\} \left( \sum_{i\in\mathcal{C}_c, 0\leq\tau'\leq\tau} \mathfrak{S}^{(i)}(\tau') - C_\ell \sum_{j\in\mathcal{N}_{-c}, 0\leq\tau'\leq\tau} \mathfrak{S}^{(j)}(\tau') \right) (1\pm\rho)\|\boldsymbol{\mu}\|_2,
$$

$$(208)$$

where $C_\ell > 0$ is the constant in Lemma D.2. If Eq.11 is satisfied with $C_3 = 2C_\ell > 0$, then Eq.208 equals to $\alpha\|\boldsymbol{\nu}\|_2/n \cdot \sum_{i\in\mathcal{C}_c, \tau'\leq\tau} \mathfrak{S}^{(i)}(\tau')\|\boldsymbol{\mu}\|_2$, ignoring the constants.

**Third term:** The absolute value of the third term in Eq.129 is bounded, ignoring constants, as follows:

$$
\frac{\alpha\|\boldsymbol{\nu}\|_2}{n} \cdot \sum_{i\in\mathcal{C}_c\cup\mathcal{N}_{-c}} \sum_{0\leq\tau'\leq\tau} \left( \sum_{t\in\mathcal{R}} s^{(i)}(\tau')_t \left(1 - s^{(i)}(\tau')_t\right) \right) \cdot \sqrt{\log(Tn/\delta)} \tag{209}
$$

$$
< \frac{\alpha\|\boldsymbol{\nu}\|_2}{n} \cdot \frac{Tn}{C_1} \left( \sum_{i\in\mathcal{C}_c} \sum_{0\leq\tau'\leq\tau} \mathfrak{S}^{(i)}(\tau') \right) \cdot \sqrt{\log(Tn/\delta)} \tag{210}
$$

$$
\lesssim \frac{\alpha\|\boldsymbol{\nu}\|_2}{n} \cdot \left( \sum_{i\in\mathcal{C}_c} \sum_{0\leq\tau'\leq\tau} \mathfrak{S}^{(i)}(\tau') \right) \|\boldsymbol{\mu}\|_2, \tag{211}
$$

where the second last inequality follows from the condition Eq.9, and the last line follows from Eq.46, which is derived by assumption (A'2). This leads to the desired bound.

**Fourth - Seventh term:** Here, we further divide into two cases based on the degrees of $\rho$ and evaluate the negative terms. For the negative linear terms of $\rho$, we have the following bound without constant:

$$
\frac{\alpha\|\boldsymbol{\nu}\|_2}{n} \sum_{i\in\mathcal{C}_c\cup\mathcal{C}_{-c}} \sum_{0\leq\tau'\leq\tau} \left( \sum_{t\in\mathcal{W}_c^{(i)}} s^{(i)}(\tau')_t \right) \left( \sum_{t\in\mathcal{R}} s^{(i)}(\tau')_t \right) \cdot \rho\|\boldsymbol{\mu}\|_2 \tag{212}
$$

$$
\lesssim \frac{\alpha\|\boldsymbol{\nu}\|_2}{n} \cdot \left( \sum_{i\in\mathcal{C}_c} \sum_{0\leq\tau'\leq\tau} \mathfrak{S}^{(i)}(\tau') \right) \rho\|\boldsymbol{\mu}\|_2, \tag{213}
$$

where the last line follows from $\sum_{t\in\mathcal{W}_c^{(i)}} s^{(i)}(\tau')_t \leq 1 - \sum_{t\in\mathcal{R}} s^{(i)}(\tau')_t$ and the condition on the class balance of $\mathfrak{S}$, i.e., Eq.10. Since $\rho = \Omega(1)$ is a small constant, Eq.213 is bounded by the order of the first term.

Next, for the negative quadratic terms of $\rho$, we have the following bound without constant:

$$
\frac{\alpha\|\boldsymbol{\nu}\|_2}{n} \sum_{i\in\mathcal{C}_{-c}\cup\mathcal{N}_{-c}} \sum_{0\leq\tau'\leq\tau} \left( \sum_{t\in\mathcal{W}_c^{(i)}} s^{(i)}(\tau')_t \right) \left( 1 - \sum_{t\in\mathcal{W}_c^{(i)}} s^{(i)}(\tau')_t \right) \cdot \rho^2\|\boldsymbol{\mu}\|_2 \tag{214}
$$

$$
\leq \frac{\alpha\|\boldsymbol{\nu}\|_2}{n} \sum_{i\in\mathcal{C}_{-c}\cup\mathcal{N}_{-c}} \sum_{0\leq\tau'\leq\tau} \sum_{t\in\mathcal{W}_c^{(i)}} s^{(i)}(\tau')_t \left(1 - s^{(i)}(\tau')_t\right) \cdot \rho^2\|\boldsymbol{\mu}\|_2 \tag{215}
$$

$$
< \frac{\alpha\|\boldsymbol{\nu}\|_2}{n} \cdot \frac{Tn}{C_1} \left( \sum_{i\in\mathcal{C}_c} \sum_{0\leq\tau'\leq\tau} \mathfrak{S}^{(i)}(\tau') \right) \cdot \rho^2\|\boldsymbol{\mu}\|_2 \tag{216}
$$

$$
\lesssim \frac{\alpha\|\boldsymbol{\nu}\|_2}{n} \cdot \left( \sum_{i\in\mathcal{C}_c} \sum_{0\leq\tau'\leq\tau} \mathfrak{S}^{(i)}(\tau') \right) \|\boldsymbol{\mu}\|_2, \tag{217}
$$

where the first inequality follows from

$$\left( \sum_{t \in \mathcal{W}_c^{(i)}} s^{(i)}(\tau')_t \right) \left( 1 - \sum_{t \in \mathcal{W}_c^{(i)}} s^{(i)}(\tau')_t \right) \leq \left( \sum_{t \in \mathcal{W}_c^{(i)}} s^{(i)}(\tau')_t \right) \left( 1 - \max_{t \in \mathcal{W}_c^{(i)}} \left\{ s^{(i)}(\tau')_t \right\} \right) \tag{218}$$

$$\leq \sum_{t \in \mathcal{W}_c^{(i)}} s^{(i)}(\tau')_t \left( 1 - s^{(i)}(\tau')_t \right). \tag{219}$$

We used the condition Eq.9 in Eq.216. The inequality 217 follows the assumption (A'3): $\rho^2 \leq 1/(CTn)$, which provides the desired result.

**Last term:** The last term's absolute value is bounded similarly to the third term, so we omit the same discussion.

Consequently, combining Eq.208 with the bounds for other terms concludes the proof. $\qquad\square$

Additionally, we provide a lemma on how the concentration of attention probability implies the norm of the learnable parameter $\mathbf{p}$.

**Lemma E.2.** *Suppose that we have $s(\tau)_{t^*} > 1 - \epsilon$ for some $\tau \geq 0, t^* \in [T], \epsilon > 0$. Then, on a good run, we have*

$$\|\mathbf{p}(\tau)\|_2 \geq \frac{1}{2 \left( \|\boldsymbol{\mu}\|_2 + 2\sqrt{\mathrm{Tr}(\boldsymbol{\Sigma})} \right)} \log \left( \frac{1 - \epsilon}{\epsilon}(T - 1) \right). \tag{220}$$

*Proof.* Since we have $s(\tau)_{t^*} > 1 - \epsilon$, we obtain

$$\frac{\exp \left( \mathbf{x}_{t^*}^\top \mathbf{W}^\top \mathbf{p}(\tau) \right)}{\sum_{u \in [T]} \exp \left( \mathbf{x}_u \mathbf{W}^\top \mathbf{p}(\tau) \right)} \geq 1 - \epsilon. \tag{221}$$

By rearranging terms and using the inequality $\mathbf{x}^\top \mathbf{W}^\top \mathbf{p}(\tau) \leq \|\mathbf{x}\|_2 \|\mathbf{p}(\tau)\|_2$, which follows from Cauchy-Schwarz inequality and the assumption B, we have

$$\exp \left( \max_{t \in [T]} \{\|\mathbf{x}_t\|_2\} \|\mathbf{p}(\tau)\|_2 \right) \geq \frac{1 - \epsilon}{\epsilon}(T - 1) \exp \left( -\max_{t \in [T]} \{\|\mathbf{x}_t\|_2\} \|\mathbf{p}(\tau)\|_2 \right). \tag{222}$$

Since we have $\max_{t \in [T]} \{\|\mathbf{x}_t\|_2\} \leq \|\boldsymbol{\mu}\|_2 + (1 + 1/C')\sqrt{\mathrm{Tr}(\boldsymbol{\Sigma})}$ on a good run, we obtain the desired result from Eq.222. $\qquad\square$

### E.2.2 PROOF OF LEMMA A.5

Lemma A.5, introduced in the proof sketch, extracts the essential part of the generalization analysis in this section and is not used in the complete proof provided in this section. It can be shown using Lemma E.1, the condition Eq.10, and the order evaluation of the noise component, which will be discussed in the next section.

In the following, we first provide proof of benign overfitting and move on to discussing the not-benign case.

### E.2.3 BENIGN OVERFITTING CASE

*Proof.* Since the trained model has already been shown to overfit the training data in Section E.1, we proceed to show that the generalization error gets sufficiently small, in a similar way to Section C.2.

From Eq.124, it suffices to show that the output $f_\tau(\mathbf{X})$ with the true label $Y^* = 1$ is deterministically positive under the event $\mathcal{E}$, which was defined in Definition C.1. By following the same argument as in Section C.1, if we can show

$$\mathbf{x}_t^\top \mathbf{W}^\top \mathbf{p}(\tau) \geq b_1 \|\boldsymbol{\mu}\|_2^2, \quad \mathbf{x}_u^\top \mathbf{W}^\top \mathbf{p}(\tau) \leq b_2 \|\boldsymbol{\mu}\|_2^2, \tag{223}$$

for all $t \in \mathcal{R}, u \in [T] \setminus \mathcal{R}$ and some constant $b_1 > b_2 > 0$, and this gap $b_1 - b_2 > 0$, i.e., the norm of $\mathbf{p}(\tau)$, becomes sufficiently large, then we can show $f_\tau(\mathbf{X}) > 0$ as in Eq.110.

Lemma D.1 implies that the trained model $\mathbf{p}(\tau)$ is uniquely decomposed, and we have

$$
\mathbf{x}_t^\top \mathbf{W}^\top \mathbf{p}(\tau) \geq \lambda_{+1}(\tau)\|\boldsymbol{\mu}\|_2^2 - \left( |\lambda_{+1}(\tau)| + |\lambda_{-1}(\tau)| + \sum_{i,t} |\rho_{i,t}(\tau)| \right) c_2 \|\boldsymbol{\mu}\|_2 \sqrt{\log(Tn/\delta)}
$$

$$
- \left( \sum_{i,t} |\rho_{i,t}(\tau)| \right) c_1 \sqrt{\mathrm{Tr}(\boldsymbol{\Sigma})} \log(Tn/\delta), \tag{224}
$$

$$
\mathbf{x}_u^\top \mathbf{W}^\top \mathbf{p}(\tau) \leq \rho \cdot \max\{\lambda_{+1}(\tau), \lambda_{-1}(\tau), 0\} \|\boldsymbol{\mu}\|_2^2
$$

$$
+ \left( |\lambda_{+1}(\tau)| + |\lambda_{-1}(\tau)| + \sum_{i,t} |\rho_{i,t}(\tau)| \right) c_2 \|\boldsymbol{\mu}\|_2 \sqrt{\log(Tn/\delta)}
$$

$$
+ \left( \sum_{i,t} |\rho_{i,t}(\tau)| \right) c_1 \sqrt{\mathrm{Tr}(\boldsymbol{\Sigma})} \log(Tn/\delta), \tag{225}
$$

where the inequality follows from $t \in \mathcal{R}, u \in \mathcal{W} \cup \mathcal{I}$ and the definition of the event $\mathcal{E}$. Combining with the case of $Y^* = -1$, for the small generalization error, we need

1. The signal coefficients are sufficiently large and balanced:

$$
\min\{\lambda_{+1}(\tau), \lambda_{-1}(\tau)\} > C'' \log T / \|\boldsymbol{\mu}\|_2^2, \tag{226}
$$
$$
\lambda_{+1}(\tau) \gtrsim \rho \cdot \lambda_{-1}(\tau), \ \lambda_{-1}(\tau) \gtrsim \rho \cdot \lambda_{+1}(\tau), \tag{227}
$$

where $C'' > 0$ is some constant and Eq.226 comes from the same argument as in Eq.106.

2. The noise coefficients are not too dominant:

$$
\min\{\lambda_{+1}(\tau), \lambda_{-1}(\tau)\} \gtrsim \left( \sum_{i,t} |\rho_{i,t}(\tau)| \right) \max\left\{ \frac{\sqrt{\log(Tn/\delta)}}{\|\boldsymbol{\mu}\|_2}, \frac{\sqrt{\mathrm{Tr}(\boldsymbol{\Sigma})} \log(Tn/\delta)}{\|\boldsymbol{\mu}\|_2^2} \right\}. \tag{228}
$$

Since the inside max is upper-bounded by $\Theta(1/\sqrt{Tn})$ from assumption (A'2), it is sufficient to show the left-hand side is greater than $\sum_{i,t} |\rho_{i,t}(\tau)|/\sqrt{Tn}$ ignoring constants.

In the rest of the proof, we will show Eqs.226, 227, and 228 hold. By using Lemma E.1, for all $c \in \{\pm 1\}$, we have

$$
\lambda_c(\tau) \sim \frac{\alpha \|\boldsymbol{\nu}\|_2}{n} \left( \sum_{i \in \mathcal{C}_c} \sum_{0 \leq \tau' \leq \tau} \mathfrak{S}^{(i)}(\tau') \right) \|\boldsymbol{\mu}\|_2 > 0, \tag{229}
$$

and combined with the class-balance condition Eq.10, we obtain Eq.227.

Next, we evaluate $\sum_{i,t} |\rho_{i,t}(\tau)|$ using a similar procedure to Lemma D.5 and compare it with Eq.228. Corollary D.1 gives us:

$$\sum_{i,t} |\rho_{i,t}(\tau)| \leq \sum_{0 \leq \tau' \leq \tau} \sum_{i,t} |\Delta\rho_{i,t}(\tau')| \tag{230}$$

$$\lesssim \frac{\alpha\|\boldsymbol{\nu}\|_2}{n} \sum_{\substack{0 \leq \tau' \leq \tau, \\ i \in [n]}} \left\{ 2\left(\sum_{t \in \mathcal{R}} s^{(i)}(\tau)_t\right)\left(1 - \sum_{t \in \mathcal{R}} s^{(i)}(\tau)_t\right) + 4\rho\left(\sum_{t \in \mathcal{W}_{+1}^{(i)}} s^{(i)}(\tau)_t\right)\left(\sum_{t \in \mathcal{W}_{-1}^{(i)}} s^{(i)}(\tau)_t\right) \right.$$

$$\left. + \rho\left(\sum_{t \in \mathcal{W}} s^{(i)}(\tau)_t\right)\left(\sum_{t \in \mathcal{R} \cup \mathcal{I}} s^{(i)}(\tau)_t\right) + \rho\left(\sum_{t \in \mathcal{W}} s^{(i)}(\tau)_t\right)\left(\sum_{t \in \mathcal{I}} s^{(i)}(\tau)_t\right) \right\} \|\boldsymbol{\mu}\|_2$$

$$+ \frac{\alpha\|\boldsymbol{\nu}\|_2}{n} \sum_{\substack{0 \leq \tau' \leq \tau, \\ i \in [n]}} \left(\sum_{t \in [T]} s^{(i)}(\tau)_t \left(1 - s^{(i)}(\tau)_t\right)\right) \sqrt{\log(Tn/\delta)} \tag{231}$$

$$\leq \frac{\alpha\|\boldsymbol{\nu}\|_2}{n}\left\{ \sum_{\substack{0 \leq \tau' \leq \tau, \\ i \in [n],}} 2\mathfrak{S}^{(i)}(\tau') + 4\rho \sum_{\substack{0 \leq \tau' \leq \tau, \\ i \in [n]}} \left(\sum_{t \in \mathcal{W}_{+1}^{(i)}} s^{(i)}(\tau)_t\right)\left(1 - \sum_{t \in \mathcal{W}_{+1}^{(i)}} s^{(i)}(\tau)_t\right) \right.$$

$$\left. + 2\rho \sum_{\substack{0 \leq \tau' \leq \tau, \\ i \in [n]}} \left(\sum_{t \in \mathcal{W}} s^{(i)}(\tau)_t\right)\left(1 - \sum_{t \in \mathcal{W}} s^{(i)}(\tau)_t\right) \right\} \|\boldsymbol{\mu}\|_2$$

$$+ \frac{\alpha\|\boldsymbol{\nu}\|_2}{n} \cdot \frac{Tn}{C_1}\left(\sum_{i \in \mathcal{C}_c} \sum_{0 \leq \tau' \leq \tau} \mathfrak{S}^{(i)}(\tau')\right) \cdot \sqrt{\log(Tn/\delta)} \tag{232}$$

$$\lesssim \frac{\alpha\|\boldsymbol{\nu}\|_2}{n}\left(\sum_{i \in \mathcal{C}_c} \sum_{0 \leq \tau' \leq \tau} \mathfrak{S}^{(i)}(\tau')\right)\left(\left(1 + C_2 + \frac{1 + C_2}{C_3}\right) + \sqrt{Tn} + 1\right)\|\boldsymbol{\mu}\|_2 \tag{233}$$

$$\lesssim \frac{\alpha\|\boldsymbol{\nu}\|_2}{n}\left(\sum_{i \in \mathcal{C}_c} \sum_{0 \leq \tau' \leq \tau} \mathfrak{S}^{(i)}(\tau')\right)\sqrt{Tn}\|\boldsymbol{\mu}\|_2, \tag{234}$$

where the second line follows from the same calculation as Eqs.197 and 199. In the third line, we used the definition of $\mathfrak{S}$ and the condition Eq.9 in the last term. The next line follows from the condition Eqs.9 and 10 again, and is further based on assumptions (A'2), (A'3) and Eq.46, which implies $\|\boldsymbol{\mu}\|_2 \geq CTn\sqrt{\log(Tn/\delta)}$ and $\rho \leq 1/(C\sqrt{Tn})$. Note that the same argument as in Eq.219 is applied to the second and third term in Eq.232.

Therefore, by Eq.234 and assumption (A'2), we have

$$\left(\sum_{i,t} |\rho_{i,t}(\tau)|\right) \max\left\{\frac{\sqrt{\log(Tn/\delta)}}{\|\boldsymbol{\mu}\|_2}, \frac{\sqrt{\mathrm{Tr}(\boldsymbol{\Sigma})}\log(Tn/\delta)}{\|\boldsymbol{\mu}\|_2^2}\right\}$$

$$\lesssim \frac{1}{\sqrt{Tn}} \sum_{i,t} |\rho_{i,t}(\tau)| \tag{235}$$

$$\lesssim \frac{\alpha\|\boldsymbol{\nu}\|_2}{n}\left(\sum_{i \in \mathcal{C}_c} \sum_{0 \leq \tau' \leq \tau} \mathfrak{S}^{(i)}(\tau')\right)\|\boldsymbol{\mu}\|_2, \tag{236}$$

which concludes Eq.228 combined with Eq.229.

Finally, we move on to proving Eq.226 for $\tau \geq T_0$, where $T_0$ is a sufficiently large time step appeared in the statement of Theorem 4.2. Using the convergence of the attention probability in Eq.203, for

some $i \in [n]$, we obtain that for any $\epsilon > 0$, there exists $T_0$ such that $\forall \tau \geq T_0$, $s^{(i)}(\tau)_t > 1 - \epsilon$. Then, from Lemma E.2, we have

$$\|\mathbf{p}(\tau)\|_2 \geq \frac{1}{2\left(\|\boldsymbol{\mu}\|_2 + 2\sqrt{\mathrm{Tr}(\boldsymbol{\Sigma})}\right)} \log\left(\frac{1-\epsilon}{\epsilon}(T-1)\right). \tag{237}$$

Lemma D.1 implies that $\mathbf{p}(\tau)$ can be uniquely decomposed into $\mathbf{W}\boldsymbol{\mu}_1, \mathbf{W}\boldsymbol{\mu}_2, \{\mathbf{W}\boldsymbol{\epsilon}_t^{(i)}\}_{i\in[n], t\in[T]}$, and from the class balance assumption Eq.10 and Eq.229, we can see that the scales of $\lambda_{+1}(\tau)$ and $\lambda_{-1}(\tau)$ are equal. Additionally, the sum of the noise coefficients is bounded from above by $\sqrt{Tn}\lambda_c(\tau)$ ignoring constants from Eq.234. Therefore, using triangle inequality, we have

$$\|\mathbf{p}(\tau)\|_2 \lesssim \left(2 + \sqrt{Tn}\right) \lambda_c(\tau) \max\left\{\|\boldsymbol{\mu}\|_2, (1 + 1/C')\sqrt{\mathrm{Tr}(\boldsymbol{\Sigma})}\right\}, \tag{238}$$

on a good run. By combining Eqs.237 and 238, we can increase $\lambda_c(\tau)$ by making $\epsilon$ sufficiently small, so that there exists large $T_0$ such that for all $\tau \geq T_0$, Eq.226. This concludes the proof. $\quad\square$

### E.2.4 HARMFUL OVERFITTING CASE

*Proof.* By applying Eq.13 to Lemma E.1, we have that both $\lambda_{+1}(\tau)$ and $\lambda_{-1}(\tau)$ become negative and are of the same order from the class-balance condition Eq.10. We follow the same reasoning as in the benign overfitting case in Section E.2.3, and we can show that for any $\epsilon > 0$, there exists a sufficiently large $T_0$, such that for all $\tau \geq T_0$,

$$\sum_{t\in\mathcal{R}} s(\tau)_t \leq \epsilon, \tag{239}$$

for the unseen input data $\mathbf{X}$. This is because, following a similar argument as in Eq.223 and 238, when the signal coefficients $\lambda_{+1}(\tau), \lambda_{-1}(\tau)$ are negative, taking a large $T_0$ can make the probability assigned to the relevant token $t \in \mathcal{R}$ sufficiently small.

In the rest of the proof, we assume $\lambda_{+1}(\tau) \geq \lambda_{-1}(\tau)$ for the fixed $\tau \geq T_0$, without loss of generality. Although we established that the relevant tokens are suppressed, the presence of weakly relevant tokens makes it challenging to conclude that the model's output becomes random. Here, the softmax probability for weakly relevant tokens is not necessarily suppressed because, due to the small scale of $\rho$, the negative $\lambda$ suppressing weakly relevant tokens is hidden in the effects of other noise components. We first define the following three events for the randomness of the unknown input $\mathbf{X}$:

$$\mathcal{E}_= := \left\{ \left| \sum_{u\in\mathcal{W}_{+1}} s_u(\tau) - \sum_{u\in\mathcal{W}_{-1}} s_u(\tau) \right| < \frac{4c_2\sqrt{\log(Tn/\delta)}}{\rho\|\boldsymbol{\mu}\|_2} \right\}, \tag{240}$$

$$\mathcal{E}_{+1} := \left\{ \sum_{u\in\mathcal{W}_{+1}} s_u(\tau) - \sum_{u\in\mathcal{W}_{-1}} s_u(\tau) \geq \frac{4c_2\sqrt{\log(Tn/\delta)}}{\rho\|\boldsymbol{\mu}\|_2} \right\}, \tag{241}$$

$$\mathcal{E}_{-1} := \left\{ \sum_{u\in\mathcal{W}_{-1}} s_u(\tau) - \sum_{u\in\mathcal{W}_{+1}} s_u(\tau) \geq \frac{4c_2\sqrt{\log(Tn/\delta)}}{\rho\|\boldsymbol{\mu}\|_2} \right\}, \tag{242}$$

where recall that $c_2$ is the constant in Lemma A.1. Recall that the event $\mathcal{E}$ was defined in Definition C.1, and we have

$$\Pr_{P^*_{\mathbf{X}|Y^*=1}}[f_\tau(\mathbf{X}) < 0 \mid \mathcal{E}] = \Pr[f_\tau(\mathbf{X}) < 0 \mid \mathcal{E} \cup \mathcal{E}_=]\Pr[\mathcal{E}_=] + \Pr[f_\tau(\mathbf{X}) < 0 \mid \mathcal{E} \cup \mathcal{E}_{+1}]\Pr[\mathcal{E}_{+1}]$$

$$+ \Pr[f_\tau(\mathbf{X}) < 0 \mid \mathcal{E} \cup \mathcal{E}_{-1}]\Pr[\mathcal{E}_{-1}]. \tag{243}$$

We first evaluate the second and third terms. Since the total probability assigned to the relevant token is bounded as in Eq.239, by conditioning on $\mathcal{E} \cup \mathcal{E}_{-1}$, we have

$$f_\tau(\mathbf{X}) = \sum_{t\in[T]} s_t(\tau)\gamma_t \leq \frac{\|\boldsymbol{\nu}\|_2}{\sqrt{2}}\left(\epsilon\|\boldsymbol{\mu}\|_2 - \frac{4c_2\sqrt{\log(Tn/\delta)}}{\rho\|\boldsymbol{\mu}\|_2}\rho\|\boldsymbol{\mu}\|_2 + 2c_2\sqrt{\log(Tn/\delta)}\right) \tag{244}$$

$$\leq \frac{\|\boldsymbol{\nu}\|_2}{\sqrt{2}}\left(\epsilon\|\boldsymbol{\mu}\|_2 - 2c_2\sqrt{\log(Tn/\delta)}\right) < 0, \tag{245}$$

where the last line follows from the selection of sufficiently large $T_0$ so that $\epsilon$ becomes small enough, for example $\epsilon < c_2 \sqrt{\log(Tn/\delta)}/\|\boldsymbol{\mu}\|_2$. Thus, the model output becomes deterministically negative, and the third term of Eq.243 equals to $\Pr[\mathcal{E}_{-1}]$. Similarly, by the same discussion, the model output $f_\tau(\mathbf{X})$ becomes deterministically positive under the event $\mathcal{E} \cup \mathcal{E}_{+1}$; therefore, the second term becomes zero.

Next, we evaluate the conditional probability in the first term of Eq.243. Since the model output under $\mathcal{E} \cup \mathcal{E}_=$ is given by:

$$f_\tau(\mathbf{X})$$

$$= \frac{\|\boldsymbol{\nu}\|_2}{\sqrt{2}} \left( \left( \sum_{t \in \mathcal{R}} s_t(\tau) + \sum_{u \in \mathcal{W}_{+1}} s_u(\tau)\rho - \sum_{u \in \mathcal{W}_{-1}} s_u(\tau)\rho \right) \|\boldsymbol{\mu}\|_2 + \sum_{t \in [T]} s_t(\tau) \cdot \boldsymbol{\epsilon}_t^\top \left( \boldsymbol{\mu}_{+1} - \boldsymbol{\mu}_{-1} \right) \right),$$
(246)

with choice of $T_0$ such that $\epsilon < c_2 \sqrt{\log(Tn/\delta)}/\|\boldsymbol{\mu}\|_2$, we can get the following lower-bound for the first term:

$$\Pr_{P^*_{\mathbf{X}|Y^*=1}} [f_\tau(\mathbf{X}) < 0 \mid \mathcal{E} \cup \mathcal{E}_=] \geq \Pr \left[ \sum_{t \in [T]} s_t(\tau) \cdot \boldsymbol{\epsilon}_t^\top \left( \boldsymbol{\mu}_{+1} - \boldsymbol{\mu}_{-1} \right) < -5c_2 \sqrt{\log(Tn/\delta)} \right].$$
(247)

Here, the left-hand side follows the Gaussian distribution with mean zero and variance $2\|\mathbf{s}(\tau)\|_2^2 \|\boldsymbol{\mu}\|_\Sigma^2$, which is bounded below by $2\|\mathbf{s}(\tau)\|_1^2 \|\boldsymbol{\mu}\|_\Sigma^2 / T = 2\|\boldsymbol{\mu}\|_\Sigma^2 / T$. Given that the diagonal elements of $\boldsymbol{\Sigma}$ are of constant order and by Eq.46, which is derived from assumption (A'2), the deviation on the right-hand side becomes negligible relative to the standard deviation: $\Theta(\|\boldsymbol{\mu}\|_2 / \sqrt{T})$. Using the cumulative distribution function $\Phi$ of the standard Gaussian distribution, we bound Eq.247 from below by $\Phi(-0.1)$. In summary, by applying the similar discussion for $P^*_{\mathbf{X}|Y^*=-1}$, we can obtain the following lower-bound of generalization error from Eq.243:

$$\Pr_{P^*_{(\mathbf{X},Y^*)}} [Y^* \cdot f_\tau(\mathbf{X}) < 0]$$

$$\geq \Pr_{P^*_{(\mathbf{X},Y^*)}} [Y^* \cdot f_\tau(\mathbf{X}) < 0 \mid \mathcal{E}] \Pr_{P^*_{(\mathbf{X},Y^*)}} [\mathcal{E}]$$
(248)

$$\geq \frac{1}{2} \left( \Pr_{P^*_{\mathbf{X}|Y^*=1}} [f_\tau(\mathbf{X}) < 0 \mid \mathcal{E}] + \Pr_{P^*_{\mathbf{X}|Y^*=-1}} [f_\tau(\mathbf{X}) > 0 \mid \mathcal{E}] \right) \left( 1 - \frac{\delta}{n} \right)$$
(249)

$$\geq \left( \Phi(-0.1) \Pr_{P^*_{(\mathbf{X},Y^*)}} [\mathcal{E}_=] + \frac{1}{2} \Pr_{P^*_{\mathbf{X}|Y^*=1}} [\mathcal{E}_{-1}] + \frac{1}{2} \Pr_{P^*_{\mathbf{X}|Y^*=-1}} [\mathcal{E}_{+1}] \right) \left( 1 - \frac{\delta}{n} \right).$$
(250)

We used Lemma C.1 to evaluate the probability of $\mathcal{E}$.

In the rest of the proof, we examine the three probability terms in the obtained lower bound and show that at least one can be bounded below by a constant. Note that the events $\mathcal{E}_{+1}$ and $\mathcal{E}_{-1}$ are symmetrical, as they become identical when $\mathcal{W}_{+1}$ and $\mathcal{W}_{-1}$ are swapped. Furthermore, in the generation process of weakly relevant tokens, each token belongs to $\mathcal{W}_{+1}$ or $\mathcal{W}_{-1}$ with equal probability. Considering that $\rho > 0$ and currently $\lambda_{+1}(\tau) \geq \lambda_{-1}(\tau)$, and that noise vectors are generated independently of these factors, the symmetry gives us:

$$\Pr_{P^*_{(\mathbf{X},Y^*)}} [\mathcal{E}_{+1}] \geq \Pr_{P^*_{(\mathbf{X},Y^*)}} [\mathcal{E}_{-1}].$$
(251)

Next, we move on to the comparison between $\Pr_{P^*_{\mathbf{X}|Y^*=+1}}[\mathcal{E}_{+1}]$ and $\Pr_{P^*_{\mathbf{X}|Y^*=-1}}[\mathcal{E}_{+1}]$. Rephrasing the inside $\mathcal{E}_{+1}$, we obtain

$$\sum_{u \in \mathcal{W}_{+1}} \exp\left( \mathbf{x}_u^\top \mathbf{W}^\top \mathbf{p}(\tau) \right) - \sum_{u \in \mathcal{W}_{-1}} \exp\left( \mathbf{x}_u^\top \mathbf{W}^\top \mathbf{p}(\tau) \right) \geq$$

$$\frac{4c_2 \sqrt{\log(Tn/\delta)}}{\rho\|\boldsymbol{\mu}\|_2} \left( \sum_{t \in \mathcal{R}} \exp\left( \mathbf{x}_t^\top \mathbf{W}^\top \mathbf{p}(\tau) \right) + \sum_{u \in [T]\setminus\mathcal{R}} \exp\left( \mathbf{x}_u^\top \mathbf{W}^\top \mathbf{p}(\tau) \right) \right). \quad (252)$$

Since we have $\lambda_{+1}(\tau) \geq \lambda_{-1}(\tau)$ holds, considering the contribution of the relevant token in the right-hand side, the case of $Y^* = 1$ imposes stricter conditions compared to $Y^* = -1$ in terms of the randomness in the data generation, i.e., the membership of weakly relevant tokens and the generation of the noise vectors. Therefore, we have

$$\Pr_{P^*_{\mathbf{X}|Y^*=-1}}[\mathcal{E}_{+1}] \geq \Pr_{P^*_{\mathbf{X}|Y^*=+1}}[\mathcal{E}_{+1}]. \tag{253}$$

By combining Eq.251 and Eq.253, we obtain

$$\Pr_{P^*_{\mathbf{X}|Y^*=-1}}[\mathcal{E}_{+1}] \geq \Pr_{P^*_{(\mathbf{X},Y^*)}}[\mathcal{E}_{+1}] \geq \frac{1}{2}\left(1 - \Pr_{P^*_{(\mathbf{X},Y^*)}}[\mathcal{E}_{=}]\right), \tag{254}$$

where in the last inequality, we used the fact that $\Pr[\mathcal{E}_{=} \cup \mathcal{E}_{+1} \cup \mathcal{E}_{-1}] = 1$. Substituting this to Eq.250 gives us:

$$\Pr_{P^*_{(\mathbf{X},Y^*)}}[Y^* \cdot f_\tau(\mathbf{X}) < 0] \geq \left(\frac{1}{4} + \Pr_{P^*_{(\mathbf{X},Y^*)}}[\mathcal{E}_{=}]\left(\Phi(-0.1) - \frac{1}{4}\right)\right)\left(1 - \frac{\delta}{n}\right) \tag{255}$$

$$> 1/4 \cdot 1/2 = \Theta(1) > 0, \tag{256}$$

which concludes the proof. $\qquad\square$

## F  MULTI-CLASS SETTING

In this section, we will see that the analysis in the multi-class setting is a straightforward extension of the binary setting studied in the main paper.

Let $K$ be the number of classes and $\mathbf{W}_V = (\boldsymbol{\nu}_1, \ldots, \boldsymbol{\nu}_K) \in \mathbb{R}^{d \times K}$ be the weight matrix. The model output is given by

$$f(\mathbf{X}) = \mathbf{W}_V^\top \mathbf{X}^\top \mathbb{S}\left(\mathbf{X}\mathbf{W}^\top \mathbf{p}\right) \in \mathbb{R}^K. \tag{257}$$

Let $\{\boldsymbol{\mu}_c\}_{c \in [K]}$ be signal vectors corresponding to each class, and we consider the data distribution $P$, which is modified for the multi-class setting from Definition 3.1. Here, we assume the orthogonality and norm equality among signal vectors similar to assumption A. Furthermore, we assume the well-pretrained linear head $\boldsymbol{\nu}_c \propto \boldsymbol{\mu}_c - \bar{\boldsymbol{\mu}}$ for all $c \in [K]$, where $\bar{\boldsymbol{\mu}} := \sum_{c \in [K]} \boldsymbol{\mu}_c / K$, which is also discussed in Section G.

The results of the existence of benign overfitting, as in Theorem 4.1, mainly rely on the results of concentration inequalities in Lemma A.1. By taking a union bound for class numbers and appropriately updating the assumptions on parameters to depend on $K$, this proof can be easily extended to the multi-class setting.

The convergence results as in Theorem 4.2 is based on the analysis of the empirical loss function on the training data $\mathcal{S}$ sampled i.i.d. from $P$. The empirical loss function when using cross-entropy loss with softmax output is:

$$\widehat{\mathcal{L}}(\mathbf{p}) = -\frac{1}{n}\sum_{i \in [n]}\log\left(\frac{\exp\left(f(\mathbf{X}^{(i)})_{Y^{(i)}}\right)}{\sum_{c \in [K]}\exp\left(f(\mathbf{X}^{(i)})_c\right)}\right) \tag{258}$$

$$= \frac{1}{n}\sum_{i \in [n]}\log\left(\sum_{c \in [K]}\exp\left((\boldsymbol{\nu}_c - \boldsymbol{\nu}_{Y^{(i)}})^\top \mathbf{X}^{(i)\top}\mathbb{S}\left(\mathbf{X}^{(i)}\mathbf{W}^\top\mathbf{p}\right)\right)\right). \tag{259}$$

The derivative of the empirical loss function is given by

$$
\nabla_{\mathbf{p}}\widehat{\mathcal{L}}(\mathbf{p})
$$

$$
= \frac{1}{n}\sum_{i\in[n]}\frac{\sum_{c\in[K]}\exp\left((\boldsymbol{\nu}_c - \boldsymbol{\nu}_{Y^{(i)}})^\top \mathbf{X}^{(i)\top}\mathbb{S}\left(\mathbf{X}^{(i)}\mathbf{W}^\top\mathbf{p}\right)\right)\nabla_{\mathbf{p}}\left(\boldsymbol{\nu}_c - \boldsymbol{\nu}_{Y^{(i)}}\right)^\top\mathbf{X}^{(i)\top}\mathbb{S}\left(\mathbf{X}^{(i)}\mathbf{W}^\top\mathbf{p}\right)}{\sum_{k\in[K]}\exp\left((\boldsymbol{\nu}_k - \boldsymbol{\nu}_{Y^{(i)}})^\top\mathbf{X}^{(i)\top}\mathbb{S}\left(\mathbf{X}^{(i)}\mathbf{W}^\top\mathbf{p}\right)\right)}
$$

$$(260)$$

$$
= \frac{1}{n}\sum_{i\in[n]}\sum_{c\in[K]}\frac{\mathbf{W}\mathbf{X}^{(i)\top}\left(\mathrm{diag}(\mathbb{S}(\mathbf{X}^{(i)}\mathbf{W}^\top\mathbf{p})) - \mathbb{S}(\mathbf{X}^{(i)}\mathbf{W}^\top\mathbf{p})\mathbb{S}(\mathbf{X}^{(i)}\mathbf{W}^\top\mathbf{p})^\top\right)\mathbf{X}^{(i)}\left(\boldsymbol{\nu}_c - \boldsymbol{\nu}_{Y^{(i)}}\right)}{\sum_{k\in[K]}\exp\left((\boldsymbol{\nu}_k - \boldsymbol{\nu}_c)^\top\mathbf{X}^{(i)\top}\mathbb{S}\left(\mathbf{X}^{(i)}\mathbf{W}^\top\mathbf{p}\right)\right)}.
$$

$$(261)$$

Since we have

$$
\frac{1}{\sum_{k\in[K]}\exp\left((\boldsymbol{\nu}_k - \boldsymbol{\nu}_c)^\top\mathbf{X}^{(i)\top}\mathbb{S}\left(\mathbf{X}^{(i)}\mathbf{W}^\top\mathbf{p}\right)\right)}
$$

$$
= \frac{1}{1 + \sum_{k\in[K]\setminus\{c\}}\exp\left((\boldsymbol{\nu}_k - \boldsymbol{\nu}_c)^\top\mathbf{X}^{(i)\top}\mathbb{S}\left(\mathbf{X}^{(i)}\mathbf{W}^\top\mathbf{p}\right)\right)},
$$

$$(262)$$

it corresponds to $-\ell'$ in Eq.18. Under the scale condition of the linear head, this term becomes constant order as discussed in Lemma D.2. The remainder is given by

$$
\sum_{c\in[K]}\mathbf{W}\mathbf{X}^{(i)\top}\left(\mathrm{diag}(\mathbb{S}(\mathbf{X}^{(i)}\mathbf{W}^\top\mathbf{p})) - \mathbb{S}(\mathbf{X}^{(i)}\mathbf{W}^\top\mathbf{p})\mathbb{S}(\mathbf{X}^{(i)}\mathbf{W}^\top\mathbf{p})^\top\right)\mathbf{X}^{(i)}\left(\boldsymbol{\nu}_{Y^{(i)}} - \boldsymbol{\nu}_c\right)
$$

$$
= \mathbf{W}\mathbf{X}^{(i)\top}\left(\mathrm{diag}(\mathbb{S}(\mathbf{X}^{(i)}\mathbf{W}^\top\mathbf{p})) - \mathbb{S}(\mathbf{X}^{(i)}\mathbf{W}^\top\mathbf{p})\mathbb{S}(\mathbf{X}^{(i)}\mathbf{W}^\top\mathbf{p})^\top\right)\mathbf{X}^{(i)}\cdot K\left(\boldsymbol{\nu}_{Y^{(i)}} - \frac{1}{K}\sum_{c\in[K]}\boldsymbol{\nu}_c\right).
$$

$$(263)$$

When applying the concentration inequalities to this gradient update in Corollary D.1, the results depend only on whether the $\boldsymbol{\nu}$ belongs to the target class $Y^{(i)}$ or not; therefore, the statement essentially does not change in the multi-class case. Finally, note that for the case $K = 2$, this gradient update reduces to the setting of the main text: $\boldsymbol{\nu} \propto \boldsymbol{\mu}_{+1} - \boldsymbol{\mu}_{-1}$.

# G  SUPPLEMENT TO FIXED LINEAR HEAD

In this section, we consider the $K$-class classification setting as in Section F for the sake of generality.

As mentioned in the main text, assuming $\mathbf{W}_V \propto (\boldsymbol{\mu}_1 - \bar{\boldsymbol{\mu}}, \ldots, \boldsymbol{\mu}_K - \bar{\boldsymbol{\mu}})$ is validated with the fact that this fixed weight is the direction of the gradient of expected risk under the initial value of $\mathbf{p}(0) = \mathbf{0}$, i.e., under uniform attention. The following lemma confirms this fact.

**Lemma G.1.** *Suppose that $\mathbf{p} = \mathbf{p}(0) = \mathbf{0}$. Then, the gradient descent direction of the expected risk at $\mathbf{W}_V = \mathbf{0}$ forms ETF geometry composed of signal vectors, i.e., we have*

$$
-\left(\nabla_{\mathbf{W}_V}\mathcal{L}(\mathbf{W}_V = \mathbf{0})\right)^\top \propto \left(\mathbf{I}_K - \frac{1}{K}\mathbf{1}_K\mathbf{1}_K^\top\right)(\boldsymbol{\mu}_1, \ldots, \boldsymbol{\mu}_K)^\top, \tag{264}
$$

*where $\mathcal{L}(\mathbf{W}_V) := \mathbb{E}_{(\mathbf{X},Y)\sim P}\left[\widehat{\mathcal{L}}(\mathbf{W}_V)\right]$. Note that the signal vectors have equal lengths and orthogonality from assumption A.*

*Proof.* Under the initial value $\mathbf{W}_V = \mathbf{0}$, from Eq.259, the gradient of expected loss is given by:

$$-\nabla_{\boldsymbol{\nu}_c}\mathcal{L}\left(\mathbf{W}_V = \mathbf{0}\right) = -\mathbb{E}_{(\mathbf{x},Y)\sim P}\left[\sum_{c'\in[K]}\frac{\nabla_{\boldsymbol{\nu}_c}f(\mathbf{X})}{\sum_{k\in[K]}\exp\left((\boldsymbol{\nu}_k-\boldsymbol{\nu}_{c'})^\top\mathbf{X}^\top\mathbb{S}\left(\mathbf{X}\mathbf{W}^\top\mathbf{p}\right)\right)}\Bigg|_{\mathbf{W}_V=0}\right] \quad (265)$$

$$= -\frac{1}{K}\mathbb{E}_{(\mathbf{x},Y)\sim P}\left[\frac{1}{T}\mathbf{X}^\top\mathbf{1}\cdot(1-K\mathbf{1}_{c=Y})\right] \quad (266)$$

$$= -\frac{1}{KT}\left(\frac{1}{K}(1-K)\cdot\mathbb{E}\left[\sum_t\mathbf{x}_t\mid Y=c\right] + \frac{K-1}{K}\mathbb{E}\left[\sum_t\mathbf{x}_t\mid Y\neq c\right]\right) \quad (267)$$

$$= \frac{K-1}{K^2 T}\left(\mathbb{E}\left[\sum_t\mathbf{x}_t\mid Y=c\right] - \mathbb{E}\left[\sum_t\mathbf{x}_t\mid Y\neq c\right]\right), \quad (268)$$

where we changed the order of gradient and integral in the first line, and in the second equality, we denote by $\mathbf{1}_A$ the indicator function which returns 1 if the event $A$ is satisfied and otherwise returns 0. The third line follows that $Y^*$ is sampled from a uniform distribution over $[K]$, and label noise is added to different labels uniformly.

Let $\bar{\boldsymbol{\mu}}$ be the mean of class signals $\sum_{c\in[K]}\boldsymbol{\mu}_c/K$. Since we have

$$\mathbb{E}\left[\sum_t\mathbf{x}_t\mid Y=c\right] = (1-\eta)\mathbb{E}\left[\sum_t\mathbf{x}_t\mid Y^*=c\right] + \frac{\eta}{K-1}\sum_{c'\in[K]\setminus\{c\}}\mathbb{E}\left[\sum_t\mathbf{x}_t\mid Y^*=c'\right] \quad (269)$$

$$= (1-\eta)T(\zeta_{\mathcal{R}}\boldsymbol{\mu}_c + \zeta_{\mathcal{W}}\rho\bar{\boldsymbol{\mu}}) + \frac{\eta T}{K-1}\sum_{c'\in[K]\setminus\{c\}}(\zeta_{\mathcal{R}}\boldsymbol{\mu}_{c'} + \zeta_{\mathcal{W}}\rho\bar{\boldsymbol{\mu}}) \quad (270)$$

$$= (1-\eta)T\zeta_{\mathcal{R}}\boldsymbol{\mu}_c + \frac{\eta T\zeta_{\mathcal{R}}}{K-1}(K\bar{\boldsymbol{\mu}} - \boldsymbol{\mu}_c) + T\zeta_{\mathcal{W}}\rho\bar{\boldsymbol{\mu}} \quad (271)$$

$$= T\zeta_{\mathcal{R}}\left(1 - \frac{K}{K-1}\eta\right)\boldsymbol{\mu}_c + T\left(\frac{K}{K-1}\zeta_{\mathcal{R}}\eta + \zeta_{\mathcal{W}}\rho\right)\bar{\boldsymbol{\mu}}, \quad (272)$$

where we used the Definition 3.1 on the data model configuration.

Therefore, Eq.268 leads to

$$-\nabla_{\boldsymbol{\nu}_c}\mathcal{L}\left(\mathbf{W}_V = \mathbf{0}\right) = \frac{K-1}{K^2}\Bigg\{\zeta_{\mathcal{R}}\left(1 - \frac{K}{K-1}\eta\right)\boldsymbol{\mu}_c + \left(\frac{K}{K-1}\zeta_{\mathcal{R}}\eta + \zeta_{\mathcal{W}}\rho\right)\bar{\boldsymbol{\mu}}$$

$$- \frac{1}{K-1}\sum_{c'\in[K]\setminus\{c\}}\left(\zeta_{\mathcal{R}}\left(1 - \frac{K}{K-1}\eta\right)\boldsymbol{\mu}_{c'} + \left(\frac{K}{K-1}\zeta_{\mathcal{R}}\eta + \zeta_{\mathcal{W}}\rho\right)\bar{\boldsymbol{\mu}}\right)\Bigg\} \quad (273)$$

$$= \frac{K-1}{K^2}\cdot\zeta_{\mathcal{R}}\left(1 - \frac{K}{K-1}\eta\right)\left(\boldsymbol{\mu}_c - \frac{1}{K-1}(K\bar{\boldsymbol{\mu}} - \boldsymbol{\mu}_c)\right) \quad (274)$$

$$= \zeta_{\mathcal{R}}\frac{K-1}{K^2}\left(1 - \frac{K}{K-1}\eta\right)\frac{K}{K-1}(\boldsymbol{\mu}_c - \bar{\boldsymbol{\mu}}). \quad (275)$$

Consequently, we have $-\nabla_{\boldsymbol{\nu}_c}\mathcal{L}\left(\mathbf{W}_V = \mathbf{0}\right) \propto \boldsymbol{\mu}_c - \bar{\boldsymbol{\mu}}$, which concludes the proof. $\square$

## H  ADDITIONAL MINOR NOTES ON MAIN TEXT

### H.1  FURTHER DETAILS ON CONSTANT SIZE ASSUMPTION OF $\mathcal{R}$ AND $\mathcal{W}$

In this section, we show that we can remove the assumptions introduced in our data model defined in Section 3.3, that the numbers of relevant tokens and weakly relevant tokens are constant across all examples. This assumption was introduced in the main text for the sake of notational simplicity.

To relax this simplification, we denote the index set of relevant, weakly relevant, and irrelevant tokens for the input $\mathbf{X}^{(i)}$ by $\mathcal{R}^{(i)} \in [T]$, $\mathcal{W}^{(i)} \in [T] \setminus \mathcal{R}^{(i)}$, and $\mathcal{I}^{(i)} = [T] \setminus (\mathcal{R}^{(i)} \cup \mathcal{W}^{(i)})$, respectively. We still assume that each token set contains at least one token as in the original setting. We then provide comments on the necessary adjustments to the arguments of the paper.

First, no modifications are required for the proof of Theorem 4.1. In the part of demonstrating overfitting, the claim remains valid because at least one relevant token exists for each example. Regarding the discussion on convergence, the metric defined in Definition 4.1, which is related to the softmax probability of relevant tokens, is updated as follows:

$$\mathfrak{S}^{(i)}(\tau) \coloneqq \left( \sum_{t \in \mathcal{R}^{(i)}} s^{(i)}(\tau)_t \right) \left( 1 - \sum_{t \in \mathcal{R}^{(i)}} s^{(i)}(\tau)_t \right), \tag{276}$$

which is slightly modified from the original definition using $\mathcal{R}^{(i)}$ instead of $\mathcal{R}$. We have to update this part; however, no substantial changes other than notations are required for the proof of Theorem 4.2. This is because the convergence discussion relies on the conditions $1 \leq |\mathcal{R}| \leq T - 1$ (similarly for $\mathcal{W}$ and $\mathcal{I}$), and the discussion is based on the inequalities up to constant factors for the worst-case scenario. The assumption that $\mathcal{R}$, $\mathcal{W}$, and $\mathcal{I}$ are common across examples is entirely unnecessary for such asymptotic evaluations.

## H.2 FAILURE OF TRAINING LINEAR CLASSIFIERS WITH UNIFORM ATTENTION

In this section, we demonstrate that training a linear head $\boldsymbol{\nu}$ on top of uniform attention ($\mathbf{p} = \mathbf{0}$) does not work well within our data model, from the perspective of benign overfitting. This discussion supports our data model setting because it is a sufficiently challenging setting so that a fixed linear head $\boldsymbol{\nu}$ alone is inadequate.

We present two scenarios to explain this situation. Since we consider the setting of uniform attention, the model is given by $f(\mathbf{X}; \boldsymbol{\nu}) = \boldsymbol{\nu}^\top \mathbf{X}^\top \mathbf{1}/T$. Let $\bar{\mathbf{x}} \coloneqq \sum_{t \in [T]} \mathbf{x}_i / T$ be the average of input sequence $\mathbf{X}$, where $\mathbf{X}$ is defined in Definition 3.1. We can write $f(\mathbf{X}; \boldsymbol{\nu}) = \boldsymbol{\nu}^\top \bar{\mathbf{x}}$ with this notation. For simplicity, in the discussion of this section, we assume that the input tokens have zero means and treat $\boldsymbol{\mu}_{+1} - \boldsymbol{\mu}_{-1}, -\boldsymbol{\mu}_{+1} + \boldsymbol{\mu}_{-1}$ as the class signals for $Y = +1, -1$, respectively. Please note that this does not change the signal strength ignoring constants.

**Scenario 1: One Relevant Token and Irrelevant Tokens.** In this scenario, we consider the setting where every input $\mathbf{X}$ has one relevant token, and all other tokens are irrelevant tokens. The following proposition is based on the result of (Cao et al. (2021), Theorem 3.2). It gives the lower bound of the generalization error of the linear max-margin classifier.

**Proposition H.1.** *Let $\hat{\boldsymbol{\nu}}_{SVM}$ be the maximum margin linear classifier on $(\bar{\mathbf{x}}^{(i)}, Y^{(i)})_{i=1}^n$, where each input is averaged from the original training set $\mathcal{S} = (\mathbf{X}^{(i)}, Y^{(i)})_{i=1}^n$. Suppose that the assumption (A1) holds. If the signal strength satisfies $\|\boldsymbol{\mu}\|_2 = \Theta\left( \mathrm{Tr}(\boldsymbol{\Sigma})^{1/4} \sqrt{\log(Tn/\delta)} \right)$, which is specific case of assumption (A2), then there exists absolute constants $C, C'$, such that with probability at least $1 - n^{-1}$,*

$$\Pr_{(\mathbf{X}, Y^*) \sim P^*} \left[ \mathrm{sign}(f(\mathbf{X}; \hat{\boldsymbol{\nu}}_{SVM})) \neq Y^* \right] \geq C' \exp\left( -C \frac{n \log^2(Tn/\delta)}{T^2} \right). \tag{277}$$

This result shows that a max-margin linear classifier on averaged tokens fails to generalize as the sequence length $T$ increases and the noise components are dominant. Specifically, when $T \gtrsim n^{1/2} \log(Tn/\delta)$, the generalization error can be bounded from below by a constant, leading to harmful overfitting. However, under the same assumptions, Theorem 4.1 asserts that the token selection mechanism of attention architecture enables benign overfitting by selecting class-relevant tokens and noisy tokens required for label noise memorization.

*Proof of Proposition H.1.* We use the results of (Cao et al. (2021), Theorem 3.2) for the averaged inputs $\{\bar{\mathbf{x}}^{(i)}\}_i$. We note that due to the token averaging, the class signal is divided

by $T$, and the noise components of $\bar{\mathbf{x}}$ follow the distribution $N(\mathbf{0}, \boldsymbol{\Sigma}/T)$. Under the assumptions (A1) and (A2), the noise covariance $\boldsymbol{\Sigma}/T$ and signal strength $\|\boldsymbol{\mu}\|_2/T$ satisfies $\mathrm{Tr}(\boldsymbol{\Sigma}/T) \geq c \max\{n^{3/2}\|\boldsymbol{\Sigma}\|_2 T^{-1}, n\|\boldsymbol{\Sigma}\|_F T^{-1}, n\sqrt{\log n}\|\boldsymbol{\mu}\|_{\boldsymbol{\Sigma}} T^{-3/2}\}$, and $\|\boldsymbol{\mu}\|_2^2 T^{-2} \geq c\|\boldsymbol{\mu}\|_{\boldsymbol{\Sigma}} T^{-3/2}$ for some constant $c$. Here, we also used the assumption in Definition 3.1 that the diagonal elements of $\boldsymbol{\Sigma}$ are of constant order. Therefore, the conditions of the theorem 3.2 (Cao et al., 2021) are satisfied. Furthermore, since we have

$$n\left(\|\boldsymbol{\mu}/T\|_{\boldsymbol{\Sigma}/T}^2 + \|\boldsymbol{\Sigma}/T\|_2^2\right) \lesssim n\left(T^{-3}\|\boldsymbol{\mu}\|_{\boldsymbol{\Sigma}}^2 + T^{-2}\right) \tag{278}$$

$$\lesssim n\left(T^{-3}\mathrm{Tr}(\boldsymbol{\Sigma})^{1/2}\log(Tn/\delta) + T^{-2}\right) \tag{279}$$

$$\lesssim T^{-2}\mathrm{Tr}(\boldsymbol{\Sigma}) = \Theta(\|\boldsymbol{\Sigma}/T\|_F^2), \tag{280}$$

the signal strength $\|\boldsymbol{\mu}\|_2$ given in the statement corresponds to case 2 of theorem 3.2 (Cao et al., 2021), where noise dominates compared to the signal. We also have

$$\frac{\|\boldsymbol{\mu}/T\|_2^4}{\|\boldsymbol{\Sigma}/T\|_F^2} = \frac{\Theta(\mathrm{Tr}(\boldsymbol{\Sigma})\log^2(Tn/\delta))}{T^2\mathrm{Tr}(\boldsymbol{\Sigma}^2)} = \Theta\left(T^{-2}\log^2(Tn/\delta)\right). \tag{281}$$

Substituting this to the lower bound of generalization error provides

$$C'\exp\left(-Cn\|\boldsymbol{\mu}/T\|_2^4/\|\boldsymbol{\Sigma}/T\|_F^2\right) \geq C'\exp\left(-C\frac{n\log^2(Tn/\delta)}{T^2}\right), \tag{282}$$

which concludes the proof. □

**Scenario 2: One Relevant Token and Confusing Weakly Relevant Tokens.** This scenario is a more extreme case compared to the first scenario. We consider the setting where the inputs in the training data $\mathbf{X}^{(i)}$ have one relevant token, and all other tokens are the weakly relevant tokens that have the confusing class signals, i.e., belong to $\mathcal{W}_{-Y^{*(i)}}$. When the scale of the weakly relevant token is constant, i.e. $\rho = \Theta(1) < 1$, then the averaged input $\bar{\mathbf{x}}^{(i)}$ could align with confusing class signal $\boldsymbol{\mu}_{-Y^{*(i)}}$ more than the true class signals $\boldsymbol{\mu}_{Y^{*(i)}}$, depending on the sequence length $T$. The linear classifier $\boldsymbol{\nu}$ trained on such data fails completely to classify unseen samples $\mathbf{X}$. In this case, classification of the input sequence through token averaging fails, and selecting the token that contains the most class-relevant token (relevant token) becomes essentially significant.

## H.3 Additional Information for Section 4.3

We provide the supplementary notes for Section 4.3, specifically focusing on the two reasons mentioned for the difficulties in the approach from the max-margin problem.

The first reason is that the optimization of the empirical loss function is a non-convex optimization due to the softmax function, which does not guarantee global convergence. Without strong assumptions on the initial values or the nature of the data, there is no guarantee that the weights will converge to a max-margin solution (Tarzanagh et al., 2023b; Vasudeva et al., 2024).

The second reason is concerned with the formulation of the following token separation problem:

$$\underset{\mathbf{p} \in \mathbb{R}^d}{\arg\min} \|\mathbf{p}\|_2 \quad \text{subject to} \quad \min_{t \neq t_i^*} \mathbf{p}^\top \mathbf{W}\left(\mathbf{x}_{t_i^*}^{(i)} - \mathbf{x}_t^{(i)}\right) \geq 1, \ \forall i \in [n],$$

where $t_i^* \in [T]$ is the token index to pick up for the input $\mathbf{X}^{(i)}$. For the local solution of the empirical loss function to match the max-margin solution of this problem, simplifying assumptions are necessary, such as identical token scores $\gamma$ of unpicked tokens. In our analysis, the noise in the token scores is significant for fitting the noisy data, so the difference in token scores among noise vectors cannot be ignored. In addition, we consider a more general setting where there are multiple relevant tokens, which have large token scores.

## I Additional Experiments

### I.1 Synthetic Experiments

In this section, we provide the additional results of the synthetic experiments in the main text.

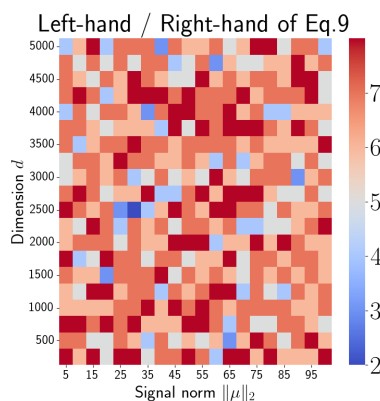
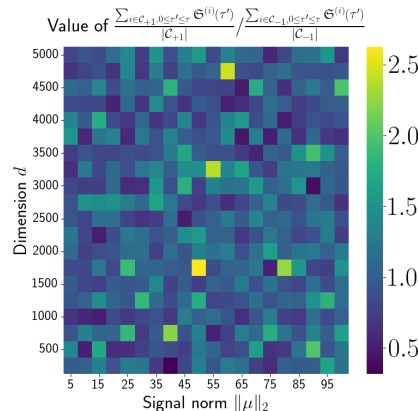

Figure 4: Heat-map of the left-hand side divided by the right-hand side of Eq.9.

Figure 5: Heat-map of the scaled ratio of each class term in Eq.10.

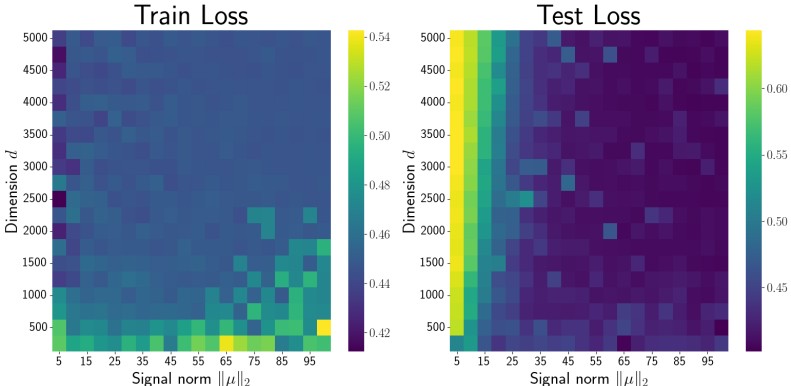

Figure 6: Heat-map of train loss and test loss under different signal norm $\|\boldsymbol{\mu}\|_2$ and the dimension $d$, after 20000 iterations. The yellow color indicates a higher loss value.

**Validity of Conditions in Theorem 4.2.** We first verify the conditions in Theorem 4.2. Figure 4 shows the ratio of the left-hand side to the right-hand side of Eq. 9 when varying $d$ and $\|\boldsymbol{\mu}\|_2$. The key point here is that the ratio exceeds 1 in all settings and training runs. The current setup is $n = 20$ and $\eta = 0.2$, with a small number of clean data in the same class, but the condition Eq.9 is expected to hold more easily when $n$ is larger. Figure 5 shows the ratio of each class term in Eq.10, scaled by the number of clean data with the same label. In other words, it shows the following value:

$$\frac{\sum_{i \in \mathcal{C}_{+1}} \sum_{0 \leq \tau' \leq \tau} \mathfrak{S}^{(i)}(\tau')}{|\mathcal{C}_{+1}|} \cdot \frac{|\mathcal{C}_{-1}|}{\sum_{i \in \mathcal{C}_{-1}} \sum_{0 \leq \tau' \leq \tau} \mathfrak{S}^{(i)}(\tau')} \tag{283}$$

The ratio was approximately in the range of $0.5$ to $2.5$ in the trials conducted under the different values of $d$ and $\|\boldsymbol{\mu}\|_2$,

**More Detailed Experiments with varying $d$ and $\|\boldsymbol{\mu}\|_2$.** Figure 6 shows the train and test loss when varying the dimension $d$ and the signal norm $\|\boldsymbol{\mu}\|_2$ under the same setting as in Section 5. This figure shows that the balance between the dimension and the signal norm is significant for achieving low train and test loss. In our paper, assuming this balance is satisfied (see assumptions in Section 3.5), we characterize the benign and harmful overfitting under two scenarios regarding label noise.

Table 3: Details of image datasets.

| Dataset | Number of Class | Input Size | Size of Step 1 | Size of Step 2 |
|---------|-----------------|------------|----------------|----------------|
| MNIST | 10 | $28 \times 28 \times 1$ | 5000 | 500 |
| CIFAR-10 | 10 | $32 \times 32 \times 3$ | 5000 | 500 |
| STL-10 | 10 | $96 \times 96 \times 3$ | 3000 | 500 |
| PneumoniaMNIST | 2 | $28 \times 28 \times 1$ | 3000 | 500 |
| BreastMNIST | 2 | $28 \times 28 \times 1$ | 300 | 100 |

Table 4: Details of natural language datasets.

| Dataset | Number of Class | Average Word Count | Size of Step 1 | Size of Step 2 |
|---------|-----------------|--------------------|----------------|----------------|
| SST-2 | 2 | 38 | 5000 | 500 |
| AG-news | 4 | 10 | 5000 | 500 |
| TREC | 6 | 10 | 3000 | 500 |

## I.2 REAL-WORLD EXPERIMENTS

We further conducted real-world experiments on image and natural language datasets for classification. For each task, we prepared the pre-trained ViT (Dosovitskiy et al., 2021) and BERT (Devlin et al., 2018) models using huggingface transformer library (Wolf et al., 2020). These models use the default configuration; specifically, they consist of 12 attention layers, the embedding dimension is set to 768, and the hidden dimension of the feed-forward layers is set to 3072. Since this is an experiment on overfitted models, dropout was not performed in any layers during the following training. We fine-tuned the models to the downstream tasks in the following two steps.

**Step 1:** Train only the classifier head on a subset of the training data **without** label noise.

**Step 2:** Train only the last attention weights on another subset of the training data **with** label noise.

In the first step, the classifier head is trained to adapt to the downstream task from the random initial values in the pre-trained model. At this time, only the last classifier is trained and label noise is not included. This step reflects our analysis setting in Section 3 that the linear head $\nu$ is fixed in the class signal direction $\mu$.

In the second step, only the last attention query-key weights are trained while keeping the classifier head fixed. We trained models on a sub-dataset, including label noise, that does not overlap with that in the first step until sufficient overfitting is achieved. This step can be corresponded to the settings of our theoretical analysis in Section 3 as follows. If the output of the backbone model up to the last attention layer is considered a fixed feature extractor, then the features of each data become the input for the final attention layer, which is the trainable one-layer model. Since the output of the CLS token position is passed to the classifier head, this setting corresponds to the scenario where the attention weight $\mathbf{W}$ is trained for a CLS token's feature $\mathbf{p}$. We confirmed in Lemma A.6 that the same results follow when optimizing $\mathbf{p}$ for a fixed $\mathbf{W}$, which is the setting discussed in this paper. However, strictly speaking, please note that there is a difference from the theoretical setting in that the feature of the CLS token is not fixed for each data due to the use of an attention model as a feature extractor.

The used datasets are described as follows. Since this experiment focuses on analyzing the generalization ability under overfitting, the sizes of the sub-datasets in Step 1 and Step 2 are set to small, 5000 and 500, respectively. For datasets with small sizes of training datasets, the smaller sizes are used and explicitly stated in Tables 3 and 4.

**Image Dataset** We conducted experiments on image classification on the following five datasets. The **MNIST** (LeCun et al., 2010) dataset consists of gray-scale $28 \times 28$ images with 10 classes. Each image is copied to form a 3-channel input and fed to the common image processor for the pre-trained ViT model. The **CIFAR-10** (Krizhevsky, 2009) is the dataset composed of $32 \times 32$ color images in 10 classes. These classes are mainly made up of vehicles and animals. The **STL-10** dataset (Coates

Table 5: Training loss and test accuracy when only the final attention query-key weights are trained on a sub-dataset containing label noise (after Step 2). The results show the average over three different runs with the standard deviation. The test accuracies are highlighted in bold.

| Dataset | Metric | Label Noise Ratio | | | | |
|---|---|---|---|---|---|---|
| | | 0.0 (Ideal) | 0.1 | 0.2 | 0.3 | 0.4 |
| MNIST | Loss (train) | $0.01_{\pm 0.00}$ | $0.10_{\pm 0.03}$ | $0.26_{\pm 0.10}$ | $0.42_{\pm 0.16}$ | $0.60_{\pm 0.23}$ |
| | Acc (test) | $\mathbf{93.7}_{\pm 0.4}$ | $\mathbf{91.4}_{\pm 0.6}$ | $\mathbf{89.0}_{\pm 0.6}$ | $\mathbf{87.2}_{\pm 0.8}$ | $\mathbf{85.9}_{\pm 0.9}$ |
| CIFAR-10 | Loss (train) | $0.00_{\pm 0.00}$ | $0.24_{\pm 0.03}$ | $0.48_{\pm 0.07}$ | $0.70_{\pm 0.11}$ | $0.92_{\pm 0.14}$ |
| | Acc (test) | $\mathbf{96.5}_{\pm 0.1}$ | $\mathbf{95.7}_{\pm 0.1}$ | $\mathbf{95.0}_{\pm 0.2}$ | $\mathbf{94.2}_{\pm 0.2}$ | $\mathbf{93.5}_{\pm 0.4}$ |
| STL-10 | Loss (train) | $0.00_{\pm 0.00}$ | $0.32_{\pm 0.04}$ | $0.64_{\pm 0.14}$ | $0.90_{\pm 0.23}$ | $1.14_{\pm 0.29}$ |
| | Acc (test) | $\mathbf{99.5}_{\pm 0.0}$ | $\mathbf{99.2}_{\pm 0.1}$ | $\mathbf{99.0}_{\pm 0.1}$ | $\mathbf{98.8}_{\pm 0.2}$ | $\mathbf{98.4}_{\pm 0.2}$ |
| Pneumonia MNIST | Loss (train) | $0.00_{\pm 0.00}$ | $0.04_{\pm 0.01}$ | $0.08_{\pm 0.04}$ | $0.12_{\pm 0.06}$ | $0.14_{\pm 0.07}$ |
| | Acc (test) | $\mathbf{82.8}_{\pm 0.8}$ | $\mathbf{84.0}_{\pm 0.3}$ | $\mathbf{83.3}_{\pm 1.5}$ | $\mathbf{80.2}_{\pm 1.5}$ | $\mathbf{80.2}_{\pm 1.7}$ |
| Breast MNIST | Loss (train) | $0.19_{\pm 0.02}$ | $0.21_{\pm 0.01}$ | $0.20_{\pm 0.01}$ | $0.23_{\pm 0.01}$ | $0.22_{\pm 0.01}$ |
| | Acc (test) | $\mathbf{81.4}_{\pm 1.8}$ | $\mathbf{80.8}_{\pm 2.4}$ | $\mathbf{78.0}_{\pm 1.3}$ | $\mathbf{78.0}_{\pm 1.6}$ | $\mathbf{78.6}_{\pm 1.3}$ |
| SST-2 | Loss (train) | $0.06_{\pm 0.01}$ | $0.09_{\pm 0.02}$ | $0.10_{\pm 0.03}$ | $0.09_{\pm 0.01}$ | $0.10_{\pm 0.01}$ |
| | Acc (test) | $\mathbf{73.7}_{\pm 0.9}$ | $\mathbf{72.5}_{\pm 1.8}$ | $\mathbf{69.9}_{\pm 1.4}$ | $\mathbf{68.6}_{\pm 1.5}$ | $\mathbf{66.5}_{\pm 2.1}$ |
| AG-news | Loss (train) | $0.00_{\pm 0.00}$ | $0.00_{\pm 0.00}$ | $0.00_{\pm 0.00}$ | $0.00_{\pm 0.00}$ | $0.00_{\pm 0.00}$ |
| | Acc (test) | $\mathbf{83.9}_{\pm 0.5}$ | $\mathbf{79.6}_{\pm 0.8}$ | $\mathbf{74.3}_{\pm 1.4}$ | $\mathbf{69.9}_{\pm 1.8}$ | $\mathbf{64.8}_{\pm 1.5}$ |
| TREC | Loss (train) | $0.00_{\pm 0.00}$ | $0.04_{\pm 0.00}$ | $0.09_{\pm 0.02}$ | $0.15_{\pm 0.03}$ | $0.20_{\pm 0.03}$ |
| | Acc (test) | $\mathbf{84.3}_{\pm 0.4}$ | $\mathbf{77.9}_{\pm 1.0}$ | $\mathbf{73.9}_{\pm 0.6}$ | $\mathbf{71.5}_{\pm 1.6}$ | $\mathbf{68.2}_{\pm 1.3}$ |

et al., 2011) is also a 10-class classification dataset inspired by the CIFAR-10, but each image has a size of $96 \times 96$ pixels. Finally, we focused on the MedMNIST (Yang et al., 2023) dataset, specifically using **PneumoniaMNIST** and **BreastMNIST**, which are tasks for disease detection. Both datasets consist of $28 \times 28$ gray-scale images similar to MNIST, and they are binary classification settings based on the presence or absence of disease. Table 3 summarizes the details of these datasets, as well as the sizes of the sub-datasets used in Step 1 and Step 2.

Furthermore, we provide comments on how our data model, based on relevant, weakly relevant, and irrelevant tokens defined in Definition 3.1, corresponds to the input images. For example, consider the case of cancer detection using medical images such as MedMNIST. In this case, the patches directly indicating cancer, such as tumors or lesions, can considered relevant tokens. The patches showing enlarged lymph nodes or inflammatory signs, which often co-occur with cancer but are not definite, can be categorized as weakly relevant tokens. Meanwhile, the patches showing normal tissue or background anatomy, which are irrelevant to the task, correspond to irrelevant tokens.

**Natural Language Dataset**  We conducted experiments on sentence classification in natural language in addition to image data. The **SST-2** (Socher et al., 2013) from GLUE benchmark (Wang et al., 2019) is the dataset for the sentiment analysis, i.e., binary classification with positive and negative. The **AG-news** (Zhang et al., 2015) is the dataset for the topic classification of news articles. It has four largest classes: "world", "sports", "business", and "science/technology". The **TREC** (Li & Roth, 2002) dataset is the dataset for question classification in 6 classes. Each question is labeled based on the content and the question type. The details of these datasets are summarized in Table 4. Note that the text data can also correspond to the data model in the paper based on the relevance of each word to the class, similar to the case of images.

The experimental results are provided in Table 5. During the training, the AdamW optimizer (Loshchilov & Hutter, 2019) was used with a learning rate of $5\mathrm{e}{-5}$, along with linear warmup and learning rate decay. For all datasets, we trained the classification head for 250 epochs in Step 1, and we trained only the last attention query-key weights for 200 epochs in Step 2. In Step 2, we increased 100 epochs for every 0.1 increase in label noise ratio because the convergence speed slows down with data containing label noise. Table 5 presents the test accuracy after Step 2, where the model is sufficiently trained on the noisy sub-datasets. The training loss is included in the table to see the degree of fit to the training set. Please note that in this setup, since only the query-key

weights are trained while value matrices and feed-forward layers are fixed, the scale of outputs does not change, and it is not possible to reduce the training loss to 0 depending on the dataset.

We first discuss the degree of overfitting observed in our experiments. For tasks such as MedMNIST (pneumonia, breast) and text classification, the model shows relatively higher overfitting on the noisy dataset. In contrast, for datasets such as MNIST, CIFAR10, and STL-10, we can see that fitting data with label noise is difficult when only the query-key weights of the final layer are trained. The potential reasons for these differences are the following.

1. Dataset: As shown in Thoerem 4.1, fitting label noise requires the presence of tokens that can explain the label noise. This scenario becomes difficult with random label noise as in this experiment setup. The model may overfit more extensively with label noise arising from the presence of confusing tokens, such as annotation errors.

2. Feature extractor: In this experiment, we treat the backbone model up to the last layer as a fixed feature extractor. By the same reason in the first point, if the feature extractor effectively suppresses the influence of noisy tokens, overfitting becomes inherently difficult.

3. Parameter size: Overparameterization is essential to select noisy tokens over relevant tokens, leading to overfitting. Achieving more overfitting requires increasing the parameter size from the attention dimension $768$ in this experiment. Please also refer to our theoretical assumptions, such as assumption (A'1), in the main text.

We also discuss the generalization ability in this result. Compared to the ideal setting where there is no label noise, datasets such as PneumoniaMNIST, CIFAR10, STL-10, BreastMNIST, and SST-2 exhibit minimal changes of generalization error in $0.1$ noise ratio. On the other hand, datasets like AG-news and TREC show more significant decreases in generalization ability. In the following, we provide the potential factors contributing to these differences.

1. Presence of tokens fitting label noise: When no tokens exist to fit the label noise, the model not only fails to overfit but also avoids selecting the desirable tokens to reduce the training error. This worsens the generalization of the model.

2. Signal strength: If the class signal is insufficiently captured, the noisy components learned to select noisy tokens to fit label noise may interfere with the selection of truly relevant tokens, leading to a decline in generalization ability. Please also see assumption (A'2) in the main text.

3. Parameter size: As mentioned in the overfitting part, further overparameterization increases the signal strength and noisy components relative to the noisy dataset size $n$, thereby bridging the gap between the setting of this experiment and the theory in the main text.

Additionally, the table shows that the generalization error increases with the label noise. In our Theorem 4.2, the condition Eq. 11 addresses the balance of contributions between clean and noisy samples, but the possible constant $C_3$ becomes smaller as label noise increases, under the same setting of model and datasets.

