# OpenReview forum: "Benign or Not-Benign Overfitting in Token Selection of Attention Mechanism"
_ICLR.cc/2025/Conference — Submitted to ICLR 2025_

### Official Review · Reviewer_vumC · 2024-10-31

**Soundness:** 3
**Presentation:** 3
**Contribution:** 2
**Rating:** 5
**Confidence:** 3

**Summary:**

This paper provides a theoretical understanding of the benign overfitting phenomenon within the framework of training Transformers. It demonstrates the existence of benign overfitting in the Transformer architecture and characterizes the conditions under which a gradient descent-based approach can converge to this benign overfitting regime, supported by theoretical guarantees.

**Strengths:**

Overall, this paper is well-written. (1) The motivation and problem formulations are clearly articulated. (2) The discussion of related works is thorough and supports the claim that this paper is the first to study benign overfitting in the context of the attention mechanism. (3) The theoretical results are reasonable and intuitive, with sufficient explanations and a detailed description of the assumptions.

**Weaknesses:**

The primary weakness lies in the assumption of a favorable initialization of \( v \) (as indicated in Lemma 3.1). I assume that the scenario in Lemma 3.1 simplifies to a fully connected neural network when \( p=0 \). Consequently, the results of Lemma 3.1 could be seen as a straightforward extension of existing findings. However, it is essential to clarify the conditions under which Lemma 3.1 holds.

Additionally, if we assume such an effective initialization, it appears that this model performs sufficiently well even without the attention mechanism, which raises questions about the necessity of analyzing the attention mechanism.

Furthermore, I did not find any results concerning \( W \), representing \( W_Q W_k \) in the attention layer.

Finally, the training approach differs from standard Transformer training practices, as this paper requires \( W_v \) to be fixed from the start. The author should verify whether this approach is effective in real applications or if the insights are transferrable to practical scenarios.

**Questions:**

N/A

---

> ### Author Response · Authors · 2024-11-19
> **Response to Reviewer vumC (1/2)**
>
> Thank you for your helpful question. We hope that our answer has addressed all your concerns. We pray this reply will change your decision.
>
> We would like to start by commenting on the assumption that the linear classifier $\boldsymbol{\mathbf{\nu}}$, where the model is defined by $f(\boldsymbol{\mathbf{X}}; \boldsymbol{\mathbf{p}}) = \boldsymbol{\nu}^\top \boldsymbol{\mathbf{X}}^\top \mathbb{S}(\boldsymbol{\mathbf{X}} \boldsymbol{\mathbf{W}}^\top\boldsymbol{\mathbf{p}})$, is fixed during the training.  Since the benign overfitting in linear classifiers is already exhibited in the existing work [1, 2], if we also optimize the linear classifier head $\boldsymbol{\mathbf{\nu}}$, then it would be non-trivial to determine whether benign overfitting is primarily driven by the attention mechanism or the linear classifier. **Our work demonstrates that benign overfitting can be achieved solely through the token selection mechanism within the softmax, which is a contribution rather than a limitation.**
> Additionally, fixing the final layer is a commonly used setting in the existing analysis of benign overfitting in neural networks [3, 4, 5, 6].
>
> ---
> > Q1: The primary weakness lies in the assumption of a favorable initialization of ( v ) (as indicated in Lemma 3.1). I assume that the scenario in Lemma 3.1 simplifies to a fully connected neural network when ( p=0 ). Consequently, the results of Lemma 3.1 could be seen as a straightforward extension of existing findings. However, it is essential to clarify the conditions under which Lemma 3.1 holds.
>
> A1: Yes, the result in Lemma 3.1 is NOT one of our contributions but rather a straightforward extension of existing results. It was introduced only to validate the assumption that the linear classifier can be set as $\boldsymbol{\mathbf{\nu}}  \propto \boldsymbol{\mathbf{\mu_{+1}}} - \boldsymbol{\mathbf{\mu_{-1}}}$.
>
> As you pointed out, this lemma is the result of a linear classifier on averaged tokens and is unrelated to the attention mechanism. For the main theorems of our paper, it is necessary for the linear head to align with the class signal $\boldsymbol{\mathbf{\mu}}$ of the data model. The Lemma G.1, which is a formal version of Lemma 3.1, states that the gradient of the expected loss function aligns with class signals, indicating that sufficient pretraining allows the linear head to learn these signals. Alternatively, the main results also hold in scenarios where the linear head is randomly fixed, and the embedding layer learns the data embeddings aligned with these linear classifier's directions.
>
> Finally, we apologize for any confusion by the typos in Lemma 3.1. The correct term is "**expected risk"** as stated in the formal version Lemma G.1, not "empirical risk" as previously written. This has now been corrected.
>
> ---
> > Q2: if we assume such an effective initialization, it appears that this model performs sufficiently well even without the attention mechanism, which raises questions about the necessity of analyzing the attention mechanism.
>
> A2: This is a very good point. However, there are two key reasons why such a trained head is insufficient:
> - (Problem 1) The model does not overfit the training set, and thus the benign overfitting is not realized (as shown in Figure 1, right).
> - (Problem 2) The linear classifier head takes the affine combination (mean in the case of uniform attention) of the token sequence $\boldsymbol{\mathbf{X}} = (\boldsymbol{\mathbf{x_1}}, \ldots \boldsymbol{\mathbf{x_T}})$.
> Depending on the sequence length $T$, the influence of the noise parts $( \boldsymbol{\mathbf{\epsilon_t}} )_{t \in [T]}$ relative to the signal $\boldsymbol{\mathbf{\mu}}$ can become significant, resulting in a challenging data model with a low signal-noise ratio.
>
> This paper demonstrates that the only token selection within the attention mechanism allows the model to overfit the label noise in the training set and still generalize well to unseen data (Problem 1). In this case, the input to the linear classifier is not the average of the input sequence but rather, a single selected token with desirable properties. This leads to a simpler data model for the classifier head (Problem 2).
>
> ---
> > Q3: I did not find any results concerning ( W ), representing ( W_Q W_k ) in the attention layer.
>
> A3: As we have already discussed in Section 3.4 and Lemma A.6 of our paper, the training of the key-query matrix $\boldsymbol{\mathbf{W}}$ plays an essentially equivalent role to that of $\boldsymbol{\mathbf{p}}$ in the token selection mechanism, and analyzing the dynamics of either $\boldsymbol{\mathbf{W}}$ or $\boldsymbol{\mathbf{p}}$ is sufficient. As a result, optimizing $\boldsymbol{\mathbf{W}}$ with fixed $\boldsymbol{\mathbf{p}}$ would yield exactly the same results as presented in our paper. This argument is consistent with the existing studies on the training dynamics of the attention mechanism [7, 8].

---

> > ### Comment · Reviewer_vumC · 2024-11-25
> >
> > Thank the author for the rebuttal. However, there are some additional questions.
> >
> > Firstly, I do not believe that "fixing the final layer is a commonly used setting in the existing analysis of benign overfitting in neural networks" is a valid reason to overlook the weaknesses of this paper. To me, the current setup does not introduce any significant challenges. Compared to a two-layer neural network, the primary difference appears to be the shift in non-linear activation from sigmoid or ReLU to softmax. However, I do not find this change to be a significant contribution.
> > To make the results more compelling, the authors should at least consider a setup where, on top of the attention layer, there is a non-linear MLP layer.
> >
> > Secondly, is it possible to provide a rigorous proof or any corollary derived from the paper to support the statement: "resulting in a challenging data model with a low signal-to-noise ratio"?
> >
> > Thirdly, I understand that $W$ represents $W_Q W_K$. My question is: what are the theoretical results regarding the training dynamics of $W$, and where can I find them?

---

> > > ### Author Response · Authors · 2024-11-26
> > > **Respond to Reviewer vumC (1/2)**
> > >
> > > Thank you for your time and insightful questions. We appreciate the opportunity to have more discussion.
> > >
> > > ---
> > > > Q'1: I do not believe that "fixing the final layer is a commonly used setting in the existing analysis of benign overfitting in neural networks" is a valid reason to overlook the weaknesses of this paper. To me, the current setup does not introduce any significant challenges. Compared to a two-layer neural network, the primary difference appears to be the shift in non-linear activation from sigmoid or ReLU to softmax. However, I do not find this change to be a significant contribution. To make the results more compelling, the authors should at least consider a setup where, on top of the attention layer, there is a non-linear MLP layer.
> > >
> > > A'1: Fixing the linear head $\boldsymbol{\mathbf{\nu}}$ is NOT a weakness of this paper, and the common setting that fixing the final layer is not the main reason justifies fixing $\boldsymbol{\mathbf{\nu}}$.
> > >
> > > Our research question is **whether benign overfitting can be achieved solely through the training of token selection within the attention, without relying on training a classifier head**. This question is non-trivial, and **it is not an artificial problem setting but rather one that aligns with real-world scenarios that appeared in fine-tuning**.  In this paper, we optimize only the weights within the attention mechanism to address this question, which is the main reason for fixing $\boldsymbol{\mathbf{\nu}}$.
> > >
> > > In addition, the differences from existing analyses on two-layer neural networks (NN) go beyond simply replacing the non-linear activation from sigmoid or ReLU to softmax. It differs fundamentally in the following points.
> > >
> > > |  | Two-layer NN | Attention (Ours) |
> > > | - | - | - |
> > > | Model（without last layer) | $\sigma(\boldsymbol{\mathbf{W}} \boldsymbol{\mathbf{x}}) \in \mathbb{R}^m$ | $\boldsymbol{\mathbf{X}}^\top \sigma(\boldsymbol{\mathbf{X}} \boldsymbol{\mathbf{W}} \boldsymbol{\mathbf{p}}) \in \mathbb{R}^d$ |
> > > | (i) Trainable Paramter | $\boldsymbol{\mathbf{W}} = (\boldsymbol{\mathbf{w}}_1, \ldots, \boldsymbol{\mathbf{w}}_m)^\top \in \mathbb{R}^{m\times d}$ | $\boldsymbol{\mathbf{p}} \in \mathbb{R}^d$ or $\boldsymbol{\mathbf{W}} \in \mathbb{R}^{d\times d}$ |
> > > | (ii) Input | $\boldsymbol{\mathbf{x}} \in \mathbb{R}^d$ | $\boldsymbol{\mathbf{X}} = (\boldsymbol{\mathbf{x}}_1, \ldots, \boldsymbol{\mathbf{x}}_T)^\top \in \mathbb{R}^{T\times d}$ |
> > > | (iii) Activation $\sigma$ | ReLU, sigmoid | softmax |
> > >
> > > (i) **Trainable Parameter:** In a two-layer NN, **the parameters $\boldsymbol{\mathbf{w}}_i \in \mathbb{R}^d$ prepared for each neuron $i \in [m]$ can be optimized independently**. In contrast, the attention mechanism handles inputs with variable-length $T$, and instead of optimizing weights prepared for each token $\boldsymbol{\mathbf{x}}_t, t \in [T]$, only shared parameter $\boldsymbol{\mathbf{p}}$ or $\boldsymbol{\mathbf{W}}$ can be optimized. This is challenging because it requires considering the competitive relationships among the class signals, as well as noise memorizing.
> > > (When training $\boldsymbol{\mathbf{W}}$ only receives gradient updates in the direction of the fixed $\boldsymbol{\mathbf{p}}$, meaning that $d\times d$ parameters cannot be freely optimized. Please also refer to our response to Q'3.)
> > >
> > > (ii) **Input**: Unlike two-layer NNs, the attention mechanism handles sequential inputs. Furthermore, not all tokens $\boldsymbol{\mathbf{x}}_t$ are helpful for classification. Some tokens may be purely noise, while others are confusing (weakly relevant tokens).
> > >
> > > (iii) **Activation**: The softmax function introduces additional challenges because **the intermediate outputs cannot be computed independently and depend on one another**, which is fundamentally different from sigmoid or ReLU. Furthermore, learning with softmax cannot change the scale of the output. This differs from ReLU or smooth LeakyReLU, which is commonly considered in the existing benign overfitting work for two-layer NNs.
> > >
> > > We believe that these differences are significant and go far beyond simply replacing the non-linear activations.

---

> > > > ### Comment · Reviewer_vumC · 2024-11-26
> > > >
> > > > Thank you for the detailed rebuttal. However, I still have some concerns that need further clarification:
> > > >
> > > > 1. Based on my understanding, only $p$ is trainable in your analysis. Why are the trainable parameters $W$ and $p$ in your case? Please correct me if my understanding is incorrect.
> > > >
> > > > 2. I do not believe the input structure represents a significant contribution. For example, the following paper also considers a CNN model with a similar structure:
> > > >    *Benign Overfitting in Two-Layer ReLU Convolutional Neural Networks for XOR Data.*
> > > >
> > > > 3. I understand the differences you highlight. However, you are only updating $p$. Why would this lead to significant challenges? Could you point out the specific parts of your proof that address handling the softmax? Please elaborate on these challenges and how you resolved them. I will review this in more detail later this week.
> > > >
> > > > 4. Regarding Q3, I was wondering if you had proved anything specific about the training dynamics of $W$ in case I missed them. Lemma A.6 appears to be a restatement of existing work. I assume that you do not have theoretical results related to the training dynamics of $W$, as you repeatedly mention this is not the focus of your paper.
> > > >
> > > > To summarize my main points: the technical contribution of your work is unclear. The model you consider in this paper is far from a practical setup, though I am not suggesting it must match a practical scenario exactly. My concern is that the model is overly simplistic (a single Transformer layer with fixed $W_QW_K$, and only $p$ is trainable) and does not sufficiently differentiate itself from existing works (even if they consider different problems). This gives the impression that your framework could be a direct extension of existing works, with only incremental improvements.

---

> > > > > ### Author Response · Authors · 2024-12-03
> > > > > **Additional Response to Reviewer vumC (1/3)**
> > > > >
> > > > > ---
> > > > > > This gives the impression that your framework could be a direct extension of existing works, with only incremental improvements.
> > > > >
> > > > > Regarding this point, we have explained to another reviewer, citing specific papers and focusing on the analysis method rather than results, that our work is neither a mere extension nor a simple combination of existing studies on attention dynamics and benign overfitting.
> > > > > The following is a similar explanation, but we add it to address your concerns more definitely.
> > > > >
> > > > > ---
> > > > > First, we refer to two papers studying the training dynamics of a tunable token $\mathbf{p}$ during token selection.
> > > > >
> > > > > **Comparison with [Max-Margin Token Selection in Attention Mechanism, (Tarzanagh+, NeurIPS2023)]**
> > > > > This paper investigates implicit bias in token selection learning. There are two major differences from us:
> > > > > - (a) **Single optimal token**, with **all other tokens having identical token scores**. (Token score is given by $\gamma_t = \mathbf{\nu}^\top \mathbf{x}_t$)
> > > > > - (b) Analysis of a fixed training set $S$ without considering the underlying data distribution.
> > > > >
> > > > > In our setting, (a) corresponds to considering a single relevant token $\mathbf{x}\_1 \in \mathcal{R}$ and irrelevant tokens $\mathbf{x}\_t \sim N(0, \mathbf{\Sigma} - \mathbf{\mu}\_{+1}\mathbf{\mu}\_{+1}^\top / \\|\mathbf{\mu}\\|\_2^2 - \mathbf{\mu}\_{-1}\mathbf{\mu}\_{-1}^\top / \\|\mathbf{\mu}\\|\_2^2)$, which are orthogonal to the class signals and have an identical token score $\gamma_t = (\mathbf{\mu}\_{+1} - \mathbf{\mu}\_{-1})^\top \mathbf{x}\_t = 0$.
> > > > >
> > > > > However, this assumption is oversimplification from realistic data models, and we consider a more general data model in the following sense:
> > > > > - Multiple class-relevant tokens $|\mathcal{R}| \geq 1$.
> > > > > - Intermediate state termed *weakly relevant token*
> > > > >   - This allows input tokens to possess weak class information or even confusing information from other classes.
> > > > > - Non-orthogonal Noise and Signal
> > > > >   - In our model, noise comes from $N(0, \mathbf{\Sigma})$.
> > > > > While this generalization does not directly appear in the results, it poses substantial difficulties in the analysis.
> > > > > If we assume orthogonality, we can completely ignore the effect of the signal components in the learned parameters on the selection of noisy tokens (and vice versa, the effect of noise memorization on the selection of signal tokens.)
> > > > > This is also a simplification assumption often used in the existing analysis of CNN benign overfitting [Cao+, NeurIPS2022, Meng+, ICML2024].
> > > > >
> > > > > Another important reason we do not rely on assumption (a) is that it is fundamentally incompatible with studying benign overfitting of the token selection mechanism.
> > > > > For clean data, it suffices to select a single relevant token. However, for data with label noise, selecting any irrelevant token results in the model with output $0$. The model cannot fit the training data. Therefore, **this data model is too simple to analyze benign overfitting in the token selection, which is the focus of our study.**
> > > > >
> > > > > Since our data model does not satisfy (a), we cannot rely on the results concerning the max-margin solutions of this paper. This is discussed in the third paragraph of Section 4.3 and Section H.3.
> > > > > **While we follow the model setting and properties of the loss function (e.g., the proof of smoothness), our analytical methods differ entirely.**

---

> > > > > ### Author Response · Authors · 2024-12-03
> > > > > **Additional Response to Reviewer vumC (3/3)**
> > > > >
> > > > > ---
> > > > > **Challenges in Adapting Existing Benign Overfitting Analyses**
> > > > > 1. Comparison with studies using the implicit bias to max-margin solutions [Wang+, NeurIPS2021, Frei+, COLT2023].
> > > > > - As discussed in our previous reply, since our data model does not satisfy the assumptions on token scores in [Tarzanagh+, 2023], we cannot apply the convergence results of max-margin solutions to our analysis.
> > > > >
> > > > > 2. Comparison with studies tracking the training dynamics of two-layer NN and CNN [Frei+, COLT2022, Xu+, AISTATS2023, Cao+, NeurIPS2022, Kou+, ICML2023].
> > > > > - In the following, we explain the difference from two main perspectives.
> > > > >
> > > > > **Training Loss Analysis**
> > > > > We consider the parameter update of gradient descent at each step
> > > > > $$
> > > > > \mathbf{W}^{(\tau+1)} - \mathbf{W}^{(\tau)} = - \alpha \nabla_\mathbf{W} \mathcal{L}(\mathbf{W}^{(\tau)}),
> > > > > $$
> > > > > and assume that this update converges. Then, the question is whether this convergence results from:
> > > > > 1. The convergence of the binary cross-entropy $\ell_i$ for each training example $x_i$ (NN, CNN case), or
> > > > > 2. The convergence of the gradient of the model output $\nabla_\mathbf{W} f(x_i; \mathbf{W}^{(\tau)})$ (attention case).
> > > > >
> > > > > For NN and CNN, which are the focus of existing research on benign overfitting, it is possible to demonstrate the convergence of binary cross-entropy loss $\ell$. Below are some specific examples.
> > > > > - (Lemma 4.12, "Benign Overfitting without Linearity: Neural Network Classifiers Trained by Gradient Descent for Noisy Linear Data" Frei+2022) shows that for smooth activation function, $\\|\nabla_\mathbf{W} \mathcal{L}(\mathbf{W}^{(\tau)})\\|\_F$ can be bounded from below using a surrogate loss involving $\ell^\prime$. The convergence of $\nabla_\mathbf{W}\mathcal{L}(\mathbf{W}^{(\tau)})$ implies the overfitting immediately.
> > > > > - (Lemma C.1 Q(t), "Benign overfitting of non-smooth neural networks beyond lazy training" Xu+2023) shows that the increase in margin $y_if(x_i; \mathbf{W}^{(\tau)})$ is bounded within bounds proportional to the surrogate loss. This ensures the convergence of the loss function.
> > > > > - (Lemma D.4, "Benign Overfitting in Two-layer Convolutional Neural Networks" Cao+2022) and (Lemma D.5, Benign Overfitting in Two-layer ReLU Convolutional Neural Networks Kou+2023) define the desirable direction $\mathbf{W}^*$ of the parameter and show that the update $\\|\mathbf{W}^{(\tau)} - \mathbf{W}^*\\|_F - \\|\mathbf{W}^{(\tau+1)} - \mathbf{W}^*\\|_F$ can be bounded from below using $\mathcal{L}(\mathbf{W}^{(\tau)})$. This demonstrates the convergence of loss function with sufficiently large $\tau$.
> > > > >
> > > > > In these studies, we can show the overfitting of the model by proving the convergence of training loss.
> > > > >
> > > > > In contrast, the analysis of attention mechanism with a softmax function is fundamentally different as follows:
> > > > > - The model output is merely an affine combination of the input sequence (weighted by the learned probabilities). Therefore, the binary cross-entropy $\ell$ cannot converge to $0$.
> > > > > - As softmax approaches hardmax, the gradient of the model output $\nabla_\mathbf{W} f(x_i; \mathbf{W})$ converges to $0$. Although we can show that $\nabla_\mathbf{W} \mathcal{L}(\mathbf{W}^{(\tau)})$ converges to $0$ with a sufficiently small learning rate using the fact that the objective function $\mathcal{L}$ is smooth, this does not imply $\ell$ approaches $0$. Instead, it reflects the convergence of the gradient of the model output $\nabla_\mathbf{W} f(x_i; \mathbf{W}^{(\tau)})$.
> > > > > **This introduces the challenge of tracking the paramter from time-step $0$ to ensure that the learned parameter fits training data in terms of 0-1 loss, by selecting the correct token.**
> > > > >
> > > > > **Generalization Analysis**
> > > > > Additionally, we also discuss the difference in the generalization error analysis.
> > > > > In NN and CNN papers, **the learning direction of the signal is determined by the ratio of clean to noisy samples in the training data**. For sufficiently small label noise, learning progresses positively in the signal direction. To demonstrate this, the loss gradient ratio of different samples $\ell^\prime(x_1) / \ell^\prime(x_2)$ is proved to be bounded at constant order across all time steps using induction arguments. This is a significant result of these studies.
> > > > > In contrast, for our setting, since the output scale is not learned, proving bounds on gradient ratios is unnecessary (as noted in Remark 4). Instead, **the difficulty lies in the interplay of the softmax probabilities, where the clean-to-noisy sample ratio alone cannot determine the learning direction of signals**. This challenge is discussed in the second paragraph of Section 4.3.
> > > > >
> > > > > While we follow the paper construction and the decomposition of the learned parameters into signal and noise components as used in existing CNN papers, the token selection setting in our work introduces fundamentally different difficulties. **We believe these challenges cannot be addressed by a straightforward combination of existing analyses.**

---

> > > ### Author Response · Authors · 2024-11-26
> > > **Respond to Reviewer vumC (2/2)**
> > >
> > > ---
> > > > Q'2:  is it possible to provide a rigorous proof or any corollary derived from the paper to support the statement: "resulting in a challenging data model with a low signal-to-noise ratio"?
> > >
> > > A'2: In response to your question, **we have newly added rigorous discussion to Section H.2.**
> > > Proposition H.1 states that the max-margin linear classifier $\hat{\boldsymbol{\mathbf{\nu}}}_{\text{SVM}}$ trained on top of uniform attention (without training of inside attention) **fails to generalize when the sequence length $T$ is large.**
> > >
> > > Specifically, it states that if the assumption (A1) and $\\|\boldsymbol{\mathbf{\mu}} \\|\_2 = \Theta(\mathrm{Tr}(\boldsymbol{\mathbf{\Sigma}})^{1/4} \sqrt{\log(Tn/\delta)} )$, which is a special case of assumption (A2) hold, then there exists absolute constants $C, C^\prime$, such that with probability at least $1 - n^{-1}$,
> > > $$
> > > \mathrm{P}_{(\boldsymbol{\mathbf{X}}, Y^*) \sim P^*}  \Bigg[ \text{sign}  (f( \boldsymbol{\mathbf{X}};  \hat{\boldsymbol{\mathbf{\nu}}}\_{\text{SVM}} )) \neq Y^*  \Bigg] \geq C^{\prime} \exp \left( - C \frac{n \log^2 (Tn/\delta)}{T^2} \right).
> > > $$
> > > It implies that if the sequence length satisfies $T \gtrsim n^{1/2} \log(Tn/\delta)$, then the generalization error is bounded from below by a constant, meaning harmful overfitting.
> > >
> > > This demonstrates that increasing the sequence length $T$ accumulates the effects of noise components in tokens (resulting in a low signal-to-noise ratio), making classification more difficult for the averaged tokens through uniform attention.
> > > **Our main results state that a benign overfitting solution of the attention mechanism exists under the entirely same condition, highlighting the importance of token selection through the training of attention mechanisms.**
> > >
> > > ---
> > > > Q'3: I understand that $W$ represents $W_QW_K$. My question is: what are the theoretical results regarding the training dynamics of $W$, and where can I find them?
> > >
> > > A'3: Thank you for your questions. Each point is clarified as follows.
> > > 1. Where can I find them?
> > > The results regarding the training dynamics of $\boldsymbol{\mathbf{W}}$ can be found in Section 3.4 in the main text and Lemma A.6.
> > >
> > > 2. What are the theoretical results regarding the training dynamics of W?
> > > Lemma A.6 states that the training $\boldsymbol{\mathbf{p}}$ from $\boldsymbol{\mathbf{p}}(0) = 0$ under a fixed $\boldsymbol{\mathbf{W}}\_0$ and training $\boldsymbol{\mathbf{W}}$ from $\boldsymbol{\mathbf{W}}(0) = 0$ under a fixed $\boldsymbol{\mathbf{p}}\_0$ yield the identical model output $f_\tau(\boldsymbol{\mathbf{X}}) = \boldsymbol{\mathbf{\nu}}^\top\boldsymbol{\mathbf{X}}^\top \mathbb{S}(\boldsymbol{\mathbf{X}}\boldsymbol{\mathbf{W}}\boldsymbol{\mathbf{p}}\_0) = \boldsymbol{\mathbf{\nu}}^\top\boldsymbol{\mathbf{X}}^\top \mathbb{S}(\boldsymbol{\mathbf{X}}\boldsymbol{\mathbf{W}}\_0\boldsymbol{\mathbf{p}})$ at time step $\tau$.
> > > Therefore, all discussions in the paper, including the results for overfitting and generalization, hold regardless of training $\boldsymbol{\mathbf{p}}$ or $\boldsymbol{\mathbf{W}}$.
> > >
> > > ---
> > > Finally, we hope that your satisfaction with this response, as well as the real-world experiments we have added in the rebuttal period, will be reflected in your evaluation score.

---

> ### Author Response · Authors · 2024-11-19
> **Response to Reviewer vumC (2/2)**
>
> ---
> > Q4: The training approach differs from standard Transformer training practices, as this paper requires ( W_v ) to be fixed from the start. The author should verify whether this approach is effective in real applications or if the insights are transferrable to practical scenarios.
>
> A4: Our results for the fixed classifier head have implications for fine-tuning settings in real-world applications. Additionally, the assumption that the classification layer leans the class signal, which was discussed in A1, is highly plausible for pre-trained models.
>
> To emphasize the practical takeaways, we have updated the last part of the Introduction as follows:
> **From a practical perspective, benign overfitting solely through the token selection mechanism has implications, particularly in parameter-efficient fine-tuning. For example, prompt-tuning [Li & Liang, 2021; Lester et al., 2021] trains only the tunable input token, and LoRA [Hu et al., 2022] focuses on training only the attention weights. The presence of benign overfitting suggests that adapting the model to low-quality downstream tasks with label noise would not be problematic in terms of generalization.**
>
> Furthermore, to strengthen the connection with real-world settings, **we have newly added the experiments on real-world datasets** in Section I of the appendix. Specifically, we conducted experiments using pre-trained ViT and BERT models, fine-tuning only the last attention layer on downstream tasks with label noise. These tasks include image classification on MNIST, CIFAR10, STL-10, and MedMNIST (pneumonia, breast) and text classification on SST-2, AG-news, and TREC, comprising a total of eight real datasets. We showed the generalization error of the model overfitted to noisy downstream tasks by varying the label noise. This experiment can be interpreted within our theorem's framework by treating the layers up to the final attention layer as fixed feature extractors and feeding their output feature vectors into the last trainable layer.
>
> ---
> Reference
> [1] Niladri S Chatterji and Philip M Long. Finite-sample analysis of interpolating linear classifiers in the overparameterized regime. Journal of Machine Learning Research, 22(129):1–30, 2021.
> [2] Yuan Cao, Quanquan Gu, and Mikhail Belkin. Risk bounds for over-parameterized maximum margin classification on sub-gaussian mixtures. Advances in Neural Information Processing Systems, 34:8407–8418, 2021.
> [3] Spencer Frei, Niladri S Chatterji, and Peter Bartlett. Benign overfitting without linearity: Neural network classifiers trained by gradient descent for noisy linear data. In Conference on Learning Theory, pp. 2668–2703. PMLR, 2022.
> [4] Yuan Cao, Zixiang Chen, Misha Belkin, and Quanquan Gu. Benign overfitting in two-layer convolutional neural networks. Advances in neural information processing systems, 35:25237–25250, 2022.
> [5] Xingyu Xu and Yuantao Gu. Benign overfitting of non-smooth neural networks beyond lazy training. In International Conference on Artificial Intelligence and Statistics, pp. 11094–11117. PMLR, 2023.
> [6] Yiwen Kou, Zixiang Chen, Yuanzhou Chen, and Quanquan Gu. Benign overfitting in two-layer relu convolutional neural networks. In International Conference on Machine Learning, pp. 17615–17659. PMLR, 2023.
> [7] Samet Oymak, Ankit Singh Rawat, Mahdi Soltanolkotabi, and Christos Thrampoulidis. On the role of attention in prompt-tuning. In International Conference on Machine Learning, pp. 26724–26768. PMLR, 2023.
> [8] Davoud Ataee Tarzanagh, Yingcong Li, Xuechen Zhang, and Samet Oymak. Max-margin token selection in attention mechanism. In Thirty-seventh Conference on Neural Information Processing Systems, 2023b

---

> ### Author Response · Authors · 2024-11-27
> **Response to Reviewer vumC (1/3)**
>
> Thank you for your precious time and detailed questions. We are really delighted to have the opportunity for further discussion.
>
> ---
> > Q''1: Based on my understanding, only p is trainable in your analysis. Why are the trainable parameters W and p in your case?
>
> A''1:
> In sequence classification using the attention mechanism, it is common to add a CLS token [Devlin+, 2018, Dosovitskiy+, ICLR2021], and the features at its position are used for classification. Additionally, adding tunable tokens for fine-tuning downstream tasks is standard practice in prompt tuning [Li&Liang, 2021, Lester+, 2021].
> By appending an additional token $\boldsymbol{\mathbf{p}}$ to the sequence input $\boldsymbol{\mathbf{X}}$ and passing the features at this token's position to the classifier head, the following our model setting is naturally obtained.
> $$
> f(\boldsymbol{\mathbf{X}}) = \mathbb{S}([\boldsymbol{\mathbf{p}}; \boldsymbol{\mathbf{X}}]^\top \boldsymbol{\mathbf{W}} \boldsymbol{\mathbf{X}}^\top)\_{1, :} \boldsymbol{\mathbf{X}} \boldsymbol{\mathbf{\nu}} = \boldsymbol{\mathbf{\nu}}^\top \boldsymbol{\mathbf{X}}^\top \mathbb{S}( \boldsymbol{\mathbf{X}}\boldsymbol{\mathbf{W}}^\top \boldsymbol{\mathbf{p}}),
> $$
> where $A\_{1,:}$ denotes the first row of the matrix $A$.
> The same model setting appears in recent theoretical analyses for the token selection [Oymak+, ICML2023, Tarzanagh+, NeuriPS2023], and we followed this for the first analysis of benign overfitting in token selection.
> Please also refer to our response to Q''3 to discuss why this setting is a sufficiently challenging problem.
> Additionally, we emphasize that our results also address the training dynamics of $\boldsymbol{\mathbf{W}}$ (see our response to Q''4).

---

> ### Author Response · Authors · 2024-11-27
> **Response to Reviewer vumC (2/3)**
>
> ---
> > Q''2: I do not believe the input structure represents a significant contribution. For example, the following paper also considers a CNN model with a similar structure.
>
> A''2:
> As you correctly pointed out, papers on benign overfitting in the CNN model, including "Benign Overfitting in Two-Layer ReLU Convolutional Neural Networks for XOR Data" you mentioned, also consider input models where a signal vector and a noise vector are concatenated.
> However, our data model significantly differs in the following ways, reflecting realistic settings of sequential inputs.
>
> |  | CNN work | Ours  |
> | - | - | - |
> | (i) Length (patch or token)  | 2 | $T$   |
> | (ii) Noise in relevant patch (or token) | No | Yes |
> | (iii) Intermediate state | No | Yes |
> | (iv) Noise covariance | $\Sigma - \mu\_{+1} \mu\_{+1}^\top / \|\mu\|\_2^2 - \mu\_{-1} \mu\_{-1}^\top / \|\mu\|\_2^2$ | $\Sigma$ |
>
> **(i): Arbitrary Sequence Length $T$**
> Our model allows for sequence inputs of any length $T$, whereas the CNN papers consider inputs with just two patches (a signal patch and a noise patch).
> Extending these studies to sequence inputs naturally results in a sequence length of $T=2$, but we avoid this because it is an unrealistic setting.
> - Note that if we can assume $T=2$, then our analysis would be very simplified. This is because proving that one token is not selected implies that another token is selected.
> - Furthermore, **multiple class-relevant tokens can exist in our analysis when extending to the general length of the sequence**. We do not assume only one class-relevant token with all other noisy tokens because it is an unrealistic simplification in practice.
>
> **(ii): Noise in Relevant Token**
> Our model allows that the class-relevant tokens include token noise ($\boldsymbol{\mathbf{x}}_t = \boldsymbol{\mathbf{\mu}} + \boldsymbol{\mathbf{\epsilon}}_t$). In contrast, CNN papers handle only pure class signals ($\boldsymbol{\mathbf{x}}_t = \boldsymbol{\mathbf{\mu}}$), meaning that every data in the same class contains an identical class-relevant patch (or token). This assumption differs from real-world data, where signal patches inherently contain some noise.
>
> **(iii): Intermediate state**
> Our model introduces the intermediate states termed "weakly relevant tokens". This allows input tokens to possess weak class information or even confusing information from other classes.
> In contrast, the existing CNN work simplifies that the patches are cleanly split into either pure class signal or pure noise. Our model more closely reflects real-world scenarios.
>
> **(iv): Non-Orthogonal Noise and Signal**
> In our model, the noise vectors are generated from $N(0, \boldsymbol{\mathbf{\Sigma}})$, while existing CNN studies assume noise vectors are drawn from $N(0, \boldsymbol{\mathbf{\Sigma}} - \boldsymbol{\mathbf{\mu}}\_{+1}\boldsymbol{\mathbf{\mu}}\_{+1}^\top / \|\boldsymbol{\mathbf{\mu}}\|\_2^2 - \boldsymbol{\mathbf{\mu}}\_{-1}\boldsymbol{\mathbf{\mu}}\_{-1}^\top / \|\boldsymbol{\mathbf{\mu}}\|\_2^2)$ for simplicity.
> Especially in multi-class settings discussed in Section F, such an assumption that all noise vectors are orthogonal to every class signal becomes implausible.
>
>
> In particular, we would like to emphasize (iv). While it does not directly appear in the results, it poses substantial difficulties in the analysis.
> If we assume orthogonality, we can completely ignore the effect of the signal components in the learned parameters on the selection of noisy tokens (and vice versa, the effect of noise memorization on the selection of signal tokens.)
> However, this assumption is unrealistic. In reality, our Lemma B.6 states that the interaction term $\langle \boldsymbol{\mathbf{\mu}}, \boldsymbol{\mathbf{\epsilon}}_t \rangle$ can be as large as $O(|\boldsymbol{\mathbf{\mu}}|_2 \sqrt{\log(Tn/\delta)})$.
> **We needed to carefully analyze the subtle interactions between noise and signal during token selection** (see also our response to the next question).

---

> ### Author Response · Authors · 2024-11-27
> **Response to Reviewer vumC (3/3)**
>
> ---
> > Q''3: However, you are only updating p. Why would this lead to significant challenges? Could you point out the specific parts of your proof that address handling the softmax? Please elaborate on these challenges and how you resolved them.
>
> A''3:
> Thank you for your question.
>
> Our analysis of benign overfitting in the attention framework is not a trivial extension of existing theoretical approaches for linear classifiers or two-layer neural networks. Section 4.3 discusses the unique difficulties raised by the softmax and how we address them. Specifically:
>
> (i) **Track attention probabilities dynamically due to local minima that do not fit the training data.**
> The existence of such local minima requires us to track how attention probabilities are allocated across token during training dynamics (Lemma A.2, A.3, A.4). In contrast, prior work only needed to show convergence of the loss gradient (Lemma A.7) to demonstrate convergence to solutions that overfit training dataset. This is not enough to analyze the attention mechanism because the loss gradient becomes zero no matter which token is selected with probability $1$.
>
> (ii) **Handle dynamic balance of gradient updates between clean and noisy data due to the properties of softmax.**
> In the prior settings, the gradient update direction was determined solely by the number of clean and noisy samples. It can be determined before the training. In our case, the contribution of clean and noisy data to gradients changes dynamically during training due to softmax. This required us to introduce a new metric and proceed with analysis based on it (Definition 4.1, Lemma E.1, Proof of Theorem 4.2).
>
> These two points are **challenges that are entirely absent from existing works on benign overfitting, and our paper tackles them newly.**
>
> Finally, although this is a citation from another paper, we would like to mention the statement at the top of p.4 in [Tarzanagh+, NeuriPS2023], which analyzes the same model setting:
> *"The highly nonlinear and nonconvex nature of the softmax operation makes the training problem in (ERM) a challenging nonconvex optimization problem for p, even with a fixed ν".*
> To prevent any misunderstanding, we clarify that this existing study focuses on implicit bias, which is entirely different from our study on benign overfitting that addresses generalization error. The aforementioned difficulties (i, ii) are novel challenges that our work resolved.
>
> ---
> > Q''4: Regarding Q3, I was wondering if you had proved anything specific about the training dynamics of W in case I missed them. Lemma A.6 appears to be a restatement of existing work. I assume that you do not have theoretical results related to the training dynamics of W, as you repeatedly mention this is not the focus of your paper.
>
> A''4: Thank you for your clarification.
> As you correctly point out, Lemma A.6 is an extension of a lemma of existing work, which considers the same model and trainable parameters. Therefore, it is not part of our contributions.
>
> However, **this does NOT mean that we provide no theoretical results related to the training dynamics of $\boldsymbol{\mathbf{W}}$**.
> Combined with Lemma A.6, which shows the one-to-one mapping between $\boldsymbol{\mathbf{W}}(\tau)$ and $\boldsymbol{\mathbf{p}}(\tau)$, Theorem 4.2 describing the training dynamics of $\boldsymbol{\mathbf{p}}$, also provides the results regarding the training dynamics of $\boldsymbol{\mathbf{W}}$.
> This is the result of benign and harmful overfitting when $\boldsymbol{\mathbf{W}}$ is trained under a fixed $\boldsymbol{\mathbf{p}}$.

---

> ### Author Response · Authors · 2024-12-03
> **Additional Response to Reviewer vumC (2/3)**
>
> ---
> **Comparison with [On the Role of Attention in Prompt-tuning, (Oymak+, ICML2023)]**
> This paper focuses on prompt tuning, comparing the expressivity of linear attention and self-attention, and analyzing generalization errors during the early training phase. The key differences from our study are as follows:
> - (a) Without label noise
> - (b) Not focusing on overfitting (related to point (a))
> - (c) Three optimization steps $(\mathbf{\nu}(1), \mathbf{p}(1), \mathbf{\nu}(2))$. In particular, the learning within the softmax is a single-step optimization of $\mathbf{p}$.
> - (d) Minor differences in input distribution
> 	- Without noise in relevant tokens, i.e., $\mathbf{x}\_t = \mathbf{\mu}$. Our setting allows the relevant token to have noise $\mathbf{x}\_t = \mathbf{\mu} + \mathbf{\epsilon}\_t$.
> 		- Note that the context-relevant token $\mathbf{q}\_*$ in [Oymak+, ICML2023] is given by $(\mathbf{\mu}\_{+1} + \mathbf{\mu}\_{-1}) / 2$ in our setting.
> 	- Intermediate state termed *weakly relevant tokens*
>
> (a, b):
> Due to the presence of label noise, we must analyze the learning direction of the class signals $\mathbf{\mu}$. As noted in (weakness 2), this direction is not determined solely by the number of clean and noisy data in the training set due to the inherent difficulty of the softmax function.
>
> (c):
> **The analysis is in the very early stage of training, so it is difficult to fit noisy data containing label noise. Therefore, their setting cannot be applied in our study.**
>
> The training starts from $\mathbf{p}(0) = 0$, which corresponds to uniform attention. Only at the first step, it suffices to consider the gradient descent with respect to the average of the tokens, and the complex softmax term does not need to be considered in the gradient calculation.
> In contrast, our analysis is for all time steps $\tau$ after a certain step $T_0$. Optimizing token selection beyond a single step introduces the softmax probabilities into the gradient calculation, resulting in a complex dependency on the current parameters. This analytical challenge differs entirely from that in [Oymak+, ICML2023].
> To address this, we needed to demonstrate the relationships among attention probabilities across all training steps using an inductive approach (Lemma A.2, A.3, and A.4). These results are significant for analyzing the dynamics of token selection in our setting.

---

### Official Review · Reviewer_HY7u · 2024-11-01

**Soundness:** 3
**Presentation:** 3
**Contribution:** 3
**Rating:** 8
**Confidence:** 2

**Summary:**

This paper investigates the phenomenon of benign overfitting within the attention mechanism, marking the first study to do so. Specifically, the authors demonstrate the existence of benign overfitting solutions under certain conditions. They establish that both signal learning and noise memorization are essential for benign overfitting in token selection. Furthermore, they illustrate that benign overfitting can be achieved through gradient descent, given specific conditions. Theoretical findings are supported by experimental results.

**Strengths:**

1) The first work to perform theoretical analysis of benign overfitting in the attention mechanism.
2) The paper is well written.
3) Theoretical findings are supported empirically.

**Weaknesses:**

1) N/A

**Questions:**

1) Any practical takeaways from the analysis of the benign overfitting phenomenon for transformer-based models (especially about their training)?

---

> ### Author Response · Authors · 2024-11-19
> **Response to Reviewer HY7u**
>
> Thank you for your review. We highly appreciate that you positively evaluate our work!
>
> ---
> > Q1: Any practical takeaways from the analysis of the benign overfitting phenomenon for transformer-based models (especially about their training)?
>
> A1: The finding in our paper provides valuable takeaways in the context of fine-tuning.
> To emphasize the practical application, we have updated the last part of the Introduction as follows:
> **From a practical perspective, benign overfitting solely through the token selection mechanism has implications, particularly in parameter-efficient fine-tuning. For example, prompt-tuning [Li & Liang, 2021; Lester et al., 2021] trains only the tunable input token, and LoRA [Hu et al., 2022] focuses on training only the attention weights. The presence of benign overfitting suggests that adapting the model to low-quality downstream tasks with label noise would not be problematic in terms of generalization.**
>
> Furthermore, to strengthen the connection with real-world settings, **we have newly added the experiments on real-world datasets** in Section I of the appendix. Specifically, we conducted experiments using pre-trained ViT and BERT models, fine-tuning only the last attention layer on downstream tasks with label noise. These tasks include image classification on MNIST, CIFAR10, STL-10, and MedMNIST (pneumonia, breast) and text classification on SST-2, AG-news, and TREC, comprising a total of eight real datasets. We showed the generalization error of the model overfitted to noisy downstream tasks by varying the label noise.
>
> Once again, thank you for your positive evaluation.

---

### Official Review · Reviewer_wjUP · 2024-11-04

**Soundness:** 2
**Presentation:** 2
**Contribution:** 2
**Rating:** 5
**Confidence:** 4

**Summary:**

This work studies benign overfitting and harmful overfitting in the token selection mechanism of the attention architecture. The theoretical analysis first shows the existence of the attention mechanism and then provides the convergence analysis. Conditions on benign and harmful overfitting are characterized theoretically.

**Strengths:**

1. The problem to solve is interesting and significant.

2. The provided theoretical analysis makes sense and seems solid.

**Weaknesses:**

1. The presentation is not satisfactory.
i) In Introduction (line 52), it says Theorem 4.1 explains the mechanism in the token selection of attention architecture. However, I cannot find any related discussion around Theorem 4.1. ii) Assumptions A3-A6 can be presented as conditions needed in the theory instead of assumptions. iii) Section 4.3 mentions some challenges, but does not mention how this paper addresses them.

2. Equations 11 and 13 are indirect, which makes the results weak. Meanwhile, it seems it cannot be associated with Figures 3 (a) and (b) about $d$ and $||\mu||$. One way to improve is to further interpret Equations 11 and 13 in terms of $||\mu||$ and $d$.

3. No experiments on real-world datasets are conducted.

4. Only $p$ is optimized, while other parameters are not changed, which makes the problem to solve less challenging.

**Questions:**

The legend in Figure 3 is confusing. There are five different lines but with the same name "irrelevant". Figure 3(a) shows that $W\_{-Y^*}$ is the largest in the attention weights, which seems contradictory to the sentence in lines 483-484.

---

> ### Author Response · Authors · 2024-11-19
> **Response to Reviewer wjUP (1/3)**
>
> Thank you for your time and insightful question. We hope that our answer has addressed all your concerns and that this will be reflected in your rating.
>
> ---
> > Q1: The presentation is not satisfactory. i) In Introduction (line 52), it says Theorem 4.1 explains the mechanism in the token selection of attention architecture. However, I cannot find any related discussion around Theorem 4.1. ii) Assumptions A3-A6 can be presented as conditions needed in the theory instead of assumptions. iii) Section 4.3 mentions some challenges, but does not mention how this paper addresses them.
>
> A1: Thank you for your constructive comments. We have addressed each point as follows.
>
> i) We have added the following at the end of Section 4.1.
> **This theorem reflects the intuition that 1) selecting class-relevant tokens is desirable for generalization and 2) selecting alternative tokens that align with the label noise is necessary for fitting noisy data. The proof provided in Section C demonstrates such token selection is performed.**
>
> ii) In this paper, we followed the style of existing papers on benign overfitting [1, 2, 3, 4], where assumptions on parameters such as training size and learning rate are presented before the main results.  Why do you believe that assumptions A3-A6 would be more appropriately referred to as conditions rather than assumptions? We are happy to revise this part if necessary.
>
> iii) We have appended explanations of our approaches for each difficulty at the end of the corresponding paragraph, as follows:
>
> - At the end of the paragraph “Presence of local minima”:
> **In the appendix, we not only analyzed the convergence of the training loss gradient but also tracked the probabilities assigned to each token, identifying which token converges to 1 and is thus selected by the model.**
>
> - At the end of the paragraph “Diminishing parameter updates due to softmax”:
> **To address this, we introduced a metric in Definition 4.1, which allows us to present scenarios that lead to either benign or harmful overfitting, as demonstrated in Eqs. (11) and (13).**
>
> ---
> > Q2: Equations 11 and 13 are indirect, which makes the results weak. Meanwhile, it seems it cannot be associated with Figures 3 (a) and (b) about d and ||μ||. One way to improve is to further interpret Equations 11 and 13 in terms of ||μ|| and d.
>
> A2: It is a very good point, and we are sorry for the confusion.
>
> Equations (11) and (13) are the conditions on the **noise in the labels $y$**, whereas $d$ and $||\boldsymbol{\mathbf{\mu}}||$ are the conditions on the **noise in the input sequence $\boldsymbol{\mathbf{X}}$**. Both of them are significant factors characterizing benign overfitting but are entirely different aspects of the problem. The conditions (11) and (13) in our Theorem 4.2 suggest that even when the label noise satisfies assumption (A6): $\eta < 1/C$, the training dynamics of attention probabilities can lead to either benign or harmful overfitting.
> In Figure 3, only the balanced setting (a) satisfies the assumptions of our main theorem. This synthetic experiment is designed to confirm the realization of benign overfitting under this condition. In contrast, Figure 3 (b) and (c) exhibit two distinct scenarios that violate the main theorem's assumptions and demonstrate the behavior differing from benign overfitting.
>
> We newly conducted a synthetic experiment under a balanced setting similar to (a) with $d = 2000$ and $||\boldsymbol{\mathbf{\mu}}||_2 = 20$, but increasing the label noise from $0.2$ to $0.8$. We observed that while the model still perfectly fits the training data, the test accuracy no longer achieves $1.0$ and instead drops to $0.993$. This scenario represents the same harmful overfitting as in Figure 3(b); however, as described above, the underlying cause is entirely different. Furthermore, this deterioration of generalization ability with increased label noise is also supported by the newly added real-world experiments discussed in the response to Question 3.
>
> To clarify the above points, we have updated the following parts in Sections 3 and 5.
>
> - Append before the sentence “The reason for requiring such conditions on $\mathfrak{S}$ will be discussed later in Section 4.3.” in Section 4.2:
> **Eqs. 11 and 13 are regarded as conditions on the balance between the contributions of clean and noisy data.**
>
> - Update the captions of Figure 3:
> Balanced setting → Balanced setting **(satisfying assumption)**
> Large noise setting → Large noise setting **(not satisfying assumption)**
> Large signal setting → Large signal setting **(not satisfying assumption)**
>
> - Update the explanation for Figure 3 (a) in Section 5:
> (Before)  The left figure shows the case where benign overfitting is achieved,
> (After) **The left figure shows the case where the balance between signal and noise satisfies the theorem assumptions (see assumptions (A1) and (A2)), and benign overfitting is achieved.**

---

> ### Author Response · Authors · 2024-11-19
> **Response to Reviewer wjUP (2/3)**
>
> ---
> > Q3: No experiments on real-world datasets are conducted.
>
> A3: **We have newly added the experiments on real-world datasets** in Section I of the appendix. Specifically, we conducted experiments using pre-trained ViT and BERT models, fine-tuning only the last attention layer on downstream tasks with label noise. These tasks include image classification on MNIST, CIFAR10, STL-10, and MedMNIST (pneumonia, breast) and text classification on SST-2, AG-news, and TREC, comprising a total of eight real datasets. We showed the generalization error of the model overfitted to noisy downstream tasks by varying the label noise. This experiment can be interpreted within our theorem's framework by treating the layers up to the final attention layer as fixed feature extractors and feeding their output feature vectors into the last trainable layer. For the rationale for considering this experimental setup, please also refer to the end of our response to Question 4.
>
> ---
> > Q4: Only p is optimized, while other parameters are not changed, which makes the problem to solve less challenging.
>
> A4: The problem setting optimizing only $\boldsymbol{\mathbf{p}}$ follows the standard setup in existing studies on the training dynamics of attention mechanisms [5, 6]. We consider this setting to be a good first step for analyzing benign overfitting in the attention because, as our paper demonstrates, even with such a model, there are scenarios where harmful overfitting occurs due to incorrect token selection. The theoretical setup is sufficiently challenging and non-trivial.
>
> In the following, we provide comments on individual components.
> 1. Fix the key-query matrix $\boldsymbol{\mathbf{W}}$
> As we discussed in Section 3.4 and Lemma A.6 of our paper, the training of the key-query matrix $\boldsymbol{\mathbf{W}}$ plays an essentially equivalent role to that of $\boldsymbol{\mathbf{p}}$ in the token selection mechanism, and analyzing the dynamics of either $\boldsymbol{\mathbf{W}}$ or $\boldsymbol{\mathbf{p}}$ is sufficient. As a result, optimizing $\boldsymbol{\mathbf{W}}$ with fixed $\boldsymbol{\mathbf{p}}$ would yield exactly the same results as presented in our paper. This argument is consistent with the existing studies on the training dynamics of the attention mechanism [5, 6].
>
> 2. Fix the linear head $\boldsymbol{\mathbf{\nu}}$
> Since the benign overfitting in linear classifiers is already shown in the existing work [7, 8], if we also optimize the linear classifier head $\boldsymbol{\mathbf{\nu}}$, then it would be non-trivial to determine whether benign overfitting is primarily driven by the attention mechanism or the linear classifier. **Our work demonstrates that benign overfitting can be achieved solely through the token selection mechanism within the softmax, which is a contribution rather than a limitation.** Additionally, fixing the final layer is a commonly used setting in the existing analysis of benign overfitting in neural networks [1, 2, 3, 4, 9, 10].
>
> Finally, the setting with a fixed classifier head offers valuable insights, particularly for the field of fine-tuning in real-world applications. To emphasize this point, we have updated the last part of the Introduction as follows:
> **From a practical perspective, benign overfitting solely through the token selection mechanism has implications, particularly in parameter-efficient fine-tuning. For example, prompt-tuning [Li & Liang, 2021; Lester et al., 2021] trains only the tunable input token, and LoRA [Hu et al., 2022] focuses on training only the attention weights. The presence of benign overfitting suggests that adapting the model to low-quality downstream tasks with label noise would not be problematic in terms of generalization.**

---

> ### Author Response · Authors · 2024-11-19
> **Response to Reviewer wjUP (3/3)**
>
> ---
> > Q5: The legend in Figure 3 is confusing. There are five different lines but with the same name "irrelevant". Figure 3(a) shows that W−Y∗ is the largest in the attention weights, which seems contradictory to the sentence in lines 483-484.
>
> A5: Thank you for pointing this out. Figure 3 is correct, but there was an error in lines 483-484.
> We have already updated it to the following:
> **selecting the weakly relevant token $u \in \mathcal{W}_{-Y^{*(j)}}$ for the $j \in \mathcal{N}$ aligns with our analysis.**
>
> For the noisy data $j \in \mathcal{N}$ in the training set, the model should select a weakly relevant token that aligns with the opposite direction of the true label $Y^{\*(j)}$, i.e., $Y^{(j)} = -Y^{\*(j)}$ to adapt to the label noise.
>
> Additionally, the five tokens labeled as "irrelevant" in Figure 3 correspond to the irrelevant tokens $\boldsymbol{\mathbf{x_4}}, \ldots, \boldsymbol{\mathbf{x_8}}$ as defined in our data model (Definition 3.1). To make this clearer, we have updated the Figure 3 legend as follows:
> (Before)
> Relevant: $\mathcal{R}$
> Weakly relevant: $\mathcal{W}_{Y^*}$
> $\vdots$
> Irrelevant: $\mathcal{I}$
>
> (After)
> $\boldsymbol{\mathbf{x_1}}$, Relevant: $\mathcal{R}$
> $\boldsymbol{\mathbf{x_2}}$, Weakly relevant: $\mathcal{W}_{Y^*}$
> $\vdots$
> $\boldsymbol{\mathbf{x_8}}$, Irrelevant: $\mathcal{I}$
>
> ---
> References
> [1] Spencer Frei, Niladri S Chatterji, and Peter Bartlett. Benign overfitting without linearity: Neural network classifiers trained by gradient descent for noisy linear data. In Conference on Learning Theory, pp. 2668–2703. PMLR, 2022.
> [2] Xingyu Xu and Yuantao Gu. Benign overfitting of non-smooth neural networks beyond lazy training. In International Conference on Artificial Intelligence and Statistics, pp. 11094–11117. PMLR, 2023.
> [3] Zhiwei Xu, Yutong Wang, Spencer Frei, Gal Vardi, and Wei Hu. Benign overfitting and grokking in reLU networks for XOR cluster data. In The Twelfth International Conference on Learning Representations, 2024
> [4] Erin George, Michael Murray, William Joseph Swartworth, and Deanna Needell. Training shallow reLU networks on noisy data using hinge loss: when do we overfit and is it benign? In Thirty-seventh Conference on Neural Information Processing Systems, 2023.
> [5] Samet Oymak, Ankit Singh Rawat, Mahdi Soltanolkotabi, and Christos Thrampoulidis. On the role of attention in prompt-tuning. In International Conference on Machine Learning, pp. 26724–26768. PMLR, 2023.
> [6] Davoud Ataee Tarzanagh, Yingcong Li, Xuechen Zhang, and Samet Oymak. Max-margin token selection in attention mechanism. In Thirty-seventh Conference on Neural Information Processing Systems, 2023b
> [7] Niladri S Chatterji and Philip M Long. Finite-sample analysis of interpolating linear classifiers in the overparameterized regime. Journal of Machine Learning Research, 22(129):1–30, 2021.
> [8] Yuan Cao, Quanquan Gu, and Mikhail Belkin. Risk bounds for over-parameterized maximum margin classification on sub-gaussian mixtures. Advances in Neural Information Processing Systems, 34:8407–8418, 2021.
> [9] Yuan Cao, Zixiang Chen, Misha Belkin, and Quanquan Gu. Benign overfitting in two-layer convolutional neural networks. Advances in neural information processing systems, 35:25237–25250, 2022.
> [10] Yiwen Kou, Zixiang Chen, Yuanzhou Chen, and Quanquan Gu. Benign overfitting in two-layer relu convolutional neural networks. In International Conference on Machine Learning, pp. 17615–17659. PMLR, 2023.

---

> > ### Comment · Reviewer_wjUP · 2024-11-27
> >
> > I have increased the score to 5 since the rebuttal partially addressed my concerns. I am still not satisfied with Weaknesses 2 and 4. For Weakness 2, I would like a complete answer to how conditions in Equations 11 and 13 are affected by tuning parameters like $||\mu||$ or the label noise. A theoretical explanation of why different values of these parameters lead to Equation 11 or 13 is needed. If there are experiment results to directly show the boundary of Equation 11 and 13 with varying parameters, that will be better. For Weakness 4, I have this concern since I know some works on benign overfitting with similar data formulation and some other works on prompt tuning by theoretically optimizing $p$ only. I am concerned the contribution of this paper is only incremental by combining these two.

---

> > > ### Author Response · Authors · 2024-11-29
> > > **Response to Reviewer wjUP (2/4)**
> > >
> > > ---
> > > > For Weakness 4, I have this concern since I know some works on benign overfitting with similar data formulation and some other works on prompt tuning by theoretically optimizing p only. I am concerned the contribution of this paper is only incremental by combining these two.
> > >
> > > Thank you for your insightful question. We will provide a detailed explanation of the analytical challenges, emphasizing why our work is not just a combination of existing analysis of benign overfitting and training dynamics of attention.
> > >
> > > First, we refer to two papers studying the training dynamics of a tunable token $\boldsymbol{\mathbf{p}}$ during token selection.
> > >
> > > **Comparison with [Max-Margin Token Selection in Attention Mechanism, (Tarzanagh+, NeurIPS2023)]**
> > > This paper investigates implicit bias in token selection learning. There are two major differences from us:
> > > - (a) **Single optimal token**, with **all other tokens having identical token scores**. (Token score is given by $\gamma_t = \boldsymbol{\mathbf{\nu}}^\top \boldsymbol{\mathbf{x}}_t$)
> > > - (b) Analysis of a fixed training set $S$ without considering the underlying data distribution.
> > >
> > > In our setting, (a) corresponds to considering a single relevant token $\boldsymbol{\mathbf{x}}\_1 \in \mathcal{R}$ and irrelevant tokens $\boldsymbol{\mathbf{x}}\_t \sim N(0, \boldsymbol{\mathbf{\Sigma}} - \boldsymbol{\mathbf{\mu}}\_{+1}\boldsymbol{\mathbf{\mu}}\_{+1}^\top / \\|\boldsymbol{\mathbf{\mu}}\\|\_2^2 - \boldsymbol{\mathbf{\mu}}\_{-1}\boldsymbol{\mathbf{\mu}}\_{-1}^\top / \\|\boldsymbol{\mathbf{\mu}}\\|\_2^2)$, which are orthogonal to the class signals and have an identical token score $\gamma_t = (\boldsymbol{\mathbf{\mu}}\_{+1} - \boldsymbol{\mathbf{\mu}}\_{-1})^\top \boldsymbol{\mathbf{x}}\_t = 0$.
> > >
> > > However, this assumption is oversimplification from realistic data models, and we consider a more general data model in the following sense:
> > > - Multiple class-relevant tokens $|\mathcal{R}| \geq 1$.
> > > - Intermediate state termed *weakly relevant token*
> > >   - This allows input tokens to possess weak class information or even confusing information from other classes.
> > > - Non-orthogonal Noise and Signal
> > >   - In our model, noise comes from $N(0, \boldsymbol{\mathbf{\Sigma}})$.
> > > While this generalization does not directly appear in the results, it poses substantial difficulties in the analysis.
> > > If we assume orthogonality, we can completely ignore the effect of the signal components in the learned parameters on the selection of noisy tokens (and vice versa, the effect of noise memorization on the selection of signal tokens.)
> > > This is also a simplification assumption often used in the existing analysis of CNN benign overfitting [Cao+, NeurIPS2022, Meng+, ICML2024].
> > >
> > > Another important reason we do not rely on assumption (a) is that it is fundamentally incompatible with studying benign overfitting of the token selection mechanism.
> > > For clean data, it suffices to select a single relevant token. However, for data with label noise, selecting any irrelevant token results in the model with output $0$. The model cannot fit the training data. Therefore, **this data model is too simple to analyze benign overfitting in the token selection, which is the focus of our study.**
> > >
> > > Since our data model does not satisfy (a), we cannot rely on the results concerning the max-margin solutions of this paper. This is discussed in the third paragraph of Section 4.3 and Section H.3.
> > > **While we follow the model setting and properties of the loss function (e.g., the proof of smoothness), our analytical methods differ entirely.**

---

> ### Author Response · Authors · 2024-11-29
> **Response to Reviewer wjUP (1/4)**
>
> Thank you for your precious time and the specific concerns you raised. We are really delighted to have the opportunity for further discussion.
>
> ---
> > (Weakness 2) For Weakness 2, I would like a complete answer to how conditions in Equations 11 and 13 are affected by tuning parameters like ||μ|| or the label noise. A theoretical explanation of why different values of these parameters lead to Equation 11 or 13 is needed. If there are experiment results to directly show the boundary of Equation 11 and 13 with varying parameters, that will be better.
>
> **Theoretical explanation**
> At the initial stage of training (time step $\tau = \Theta(1)$), these conditions are determined solely by the ratio of clean data to noisy data samples in the training set. Therefore, under the assumption of label noise ratio $\eta < 1/C$, Eq.13 does not occur, and only condition Eq.11 holds.
> **We have newly added its rigorous discussion in the form of a proposition in Section H.4 in the appendix.** This scenario, where the direction of signal learning is determined by the number of clean and noisy data, is the same as the existing benign overfitting analysis of linear and two-layer neural networks (NN). (For a comparison with existing studies, please refer to our response to weakness 4).
>
> However, as the training proceeds, the softmax function makes it difficult to discuss Eqs. (11) and (13) based solely on the ratio of clean to noisy samples, which are the values determined before the training.
> Instead, the complex interactions of attention probabilities during training become critical, and parameters such as $\\|\boldsymbol{\mathbf{\mu}}\\|_2$, which influence the speed of token selection, also affect conditions (11) and (13). In this study, we reduced these conditions to the behavior of a metric $\mathfrak{S}$, defined in Definition 4.1, and discussed based on it. We validate how this separation of Eqs. (11) and (13) occur through the following synthetic experiments.
>
> **Experimental results**
> Thank you for your valuable feedback. We fully agree that it is important to demonstrate the boundaries of Eqs. (11) and (13) under varying parameters.
>
> To address this, **we are conducting additional synthetic experiments and creating new heat maps showing the ratio between both sides of (11) and (13) under 1) different signal strength $\\|\boldsymbol{\mathbf{\mu}}\\|_2$ and label noise $\eta$, and 2) different dimension $d$ and label noise $\eta$.**
> Although we cannot show the heat map because the deadline for updating the PDF has already passed when getting your reply, we promise to add these heat maps to Section I.1 in the appendix as soon as the experiments for all configurations are completed.
>
> The following is a part of the obtained results, showing the ratio of left-hand to right-hand of Eqs. (11) and (13) under $d = 6000, \\|\boldsymbol{\mathbf{\mu}}\\|_2=100, \rho=0.05$, where please recall that $\rho$ is the scale of weakly relevant tokens.
>
> | Label Noise Ratio $\eta$ | 0.0 | 0.05 | 0.1 | 0.15 | 0.2  | ... | 0.7  | 0.75 | 0.8  | 0.85 | 0.9  |
> | - | - | - | - | - | - | - | - | - | - | - | - |
> | $\min_{c=\pm 1} \sum_{i \in C_c, 0\leq\tau^\prime\leq\tau} \mathfrak{S}^{(i)}(\tau^\prime) / \sum_{j \in N_{-c}, 0\leq\tau^\prime\leq\tau} \mathfrak{S}^{(j)}(\tau^\prime)$ | inf | 20.3 | 8.2 | 1.67 | 1.63 | ... | 1.31 | 0.47 | 0.33 | 0.25 | 0.15 |
> | Test accuracy                                                                                                                                                                   | 1.0 | 1.0  | 1.0 | 0.99 | 0.99 | ... | 0.97 | 0.90 | 0.72 | 0.65 | 0.63 |
>
> The ratio does not change gradually with the noise ratio; instead, it changes abruptly at $0.7$ and $0.75$. This is the boundary where relevant tokens in the training data are either selected or suppressed.
> Note that unlike classifiers that learn the decision plane in the direction of $\boldsymbol{\mathbf{\mu}}\_{+1} - \boldsymbol{\mathbf{\mu}}\_{-1}$, token selection can achieve correct classification even with label noise greater than $0.5$, provided the signal direction $\boldsymbol{\mathbf{\mu}}_{+1} + \boldsymbol{\mathbf{\mu}}\_{-1}$ is learned. However, with high label noise, the model incorrectly learns the class signal in the negative direction, leading to failures in token selection and generalization. This boundary shifts further left as the $\rho$, the scale of the weakly relevant token, decreases from $0.05$ (and consequently, the dimension $d$ required for overfitting increases).
>
> Finally, we have updated L.429 as follows.
> (Before)
> We will provide the experiments that discuss the validity of Eqs.9 and 10 in Section I.
> (After)
> **We will provide the synthetic experiments in Section I to discuss the validity of Eqs.9 and 10, as well as show the boundaries of Eqs.11 and 13 under varying parameter settings.**

---

> ### Author Response · Authors · 2024-11-29
> **Response to Reviewer wjUP (3/4)**
>
> **Comparison with [On the Role of Attention in Prompt-tuning, (Oymak+, ICML2023)]**
> This paper focuses on prompt tuning, comparing the expressivity of linear attention and self-attention, and analyzing generalization errors during the early training phase. The key differences from our study are as follows:
> - (a) Without label noise
> - (b) Not focusing on overfitting (related to point (a))
> - (c) Three optimization steps $(\boldsymbol{\mathbf{\nu}}(1), \boldsymbol{\mathbf{p}}(1), \boldsymbol{\mathbf{\nu}}(2))$. In particular, the learning within the softmax is a single-step optimization of $\boldsymbol{\mathbf{p}}$.
> - (d) Minor differences in input distribution
> 	- Without noise in relevant tokens, i.e., $\boldsymbol{\mathbf{x}}\_t = \boldsymbol{\mathbf{\mu}}$. Our setting allows the relevant token to have noise $\boldsymbol{\mathbf{x}}\_t = \boldsymbol{\mathbf{\mu}} + \boldsymbol{\mathbf{\epsilon}}\_t$.
> 		- Note that the context-relevant token $\boldsymbol{\mathbf{q}}\_*$ in [Oymak+, ICML2023] is given by $(\boldsymbol{\mathbf{\mu}}\_{+1} + \boldsymbol{\mathbf{\mu}}\_{-1}) / 2$ in our setting.
> 	- Intermediate state termed *weakly relevant tokens*
>
> (a, b):
> Due to the presence of label noise, we must analyze the learning direction of the class signals $\boldsymbol{\mathbf{\mu}}$. As noted in (weakness 2), this direction is not determined solely by the number of clean and noisy data in the training set due to the inherent difficulty of the softmax function.
>
> (c):
> **The analysis is in the very early stage of training, so it is difficult to fit noisy data containing label noise. Therefore, their setting cannot be applied in our study.**
>
> The training starts from $\boldsymbol{\mathbf{p}}(0) = 0$, which corresponds to uniform attention. Only at the first step, it suffices to consider the gradient descent with respect to the average of the tokens, and the complex softmax term does not need to be considered in the gradient calculation.
> In contrast, our analysis is for all time steps $\tau$ after a certain step $T_0$. Optimizing token selection beyond a single step introduces the softmax probabilities into the gradient calculation, resulting in a complex dependency on the current parameters. This analytical challenge differs entirely from that in [Oymak+, ICML2023].
> To address this, we needed to demonstrate the relationships among attention probabilities across all training steps using an inductive approach (Lemma A.2, A.3, and A.4). These results are significant for analyzing the dynamics of token selection in our setting.

---

> ### Author Response · Authors · 2024-11-29
> **Response to Reviewer wjUP (4/4)**
>
> **Challenges in Adapting Existing Benign Overfitting Analyses**
> From the perspective of existing benign overfitting analyses, we will show that the analytical difficulties of benign overfitting in token selection are fundamentally different from prior works.
>
> 1. Comparison with studies using the implicit bias to max-margin solutions [Wang+, NeurIPS2021, Frei+, COLT2023].
>
> As discussed in our previous reply, since our data model does not satisfy the assumptions on token scores in [Tarzanagh+, 2023], we cannot apply the convergence results of max-margin solutions to our analysis.
>
> 2. Comparison with studies tracking the training dynamics of two-layer NN and CNN [Frei+, COLT2022, Xu+, AISTATS2023, Cao+, NeurIPS2022, Kou+, ICML2023].
>
> The most significant analytical difference between these studies and our studies on token selection lies in the gradient of the loss function.
> Specifically, the difference is whether convergence is driven by **the convergence of each example's loss $\ell^\prime$** or by **the convergence of the gradient of the model's output $\nabla\_{\boldsymbol{\mathbf{W}}} f(\boldsymbol{\mathbf{W}})$.**
> $$
> \nabla\_{\boldsymbol{\mathbf{W}}} \mathcal{L}\_S(\boldsymbol{\mathbf{W}}) = \frac{1}{n} \sum\limits_{i=1}^n \ell^\prime (f(\boldsymbol{\mathbf{x}}\_i, \boldsymbol{\mathbf{W}})) \nabla\_{\boldsymbol{\mathbf{W}}} f(\boldsymbol{\mathbf{W}}) \rightarrow 0.
> $$
> In the existing analysis of two-layer NN and CNN, the scale of model output $f$ can be learned in the desirable direction, making $\ell^\prime$ approach zero $\ell^\prime \rightarrow 0$.
> At this time, **we can immediately show the overfitting of the model from $\ell = 0$**.
> Then, generalization analysis is performed based on the strength of the signal components in the learned parameters.
> Here, the gradient of the model itself does not become zero, as the gradient of the activation function, e.g., ReLU, is always $1$ in the positive region.
> Therefore, the model parameters are consistently updated at every time step.
> Additionally, the homogeneous property $\langle \nabla\_{\boldsymbol{\mathbf{W}}} f(\boldsymbol{\mathbf{W}}), \boldsymbol{\mathbf{W}} \rangle = f(\boldsymbol{\mathbf{W}})$, which is often exploited in these analyses, cannot apply to attention models due to the non-homogeneous properties of softmax (the output shape changes with the softmax temperature).
>
> In the token selection of the attention mechanism, the output scale of $f$ does not change and remains $\ell^\prime = \Theta(1) > 0$.
> Once a token is selected, the model's gradient $\nabla\_{\boldsymbol{\mathbf{W}}} f(\boldsymbol{\mathbf{W}})$ becomes zero, and the gradient of the loss function converges to zero even if the selected token does not fit the training data. Consequently, **merely showing gradient convergence is insufficient to show overfitting.**
> Instead, our analysis must account for whether the solution converges to one that overfits, requiring tracking the behavior of the softmax probabilities during training. This challenge is already discussed in the first paragraph of Section 4.3.
>
> Additionally, we also discuss the difference in the generalization error analysis.
> In NN and CNN papers, **the learning direction of the signal is determined by the ratio of clean to noisy samples in the training data**. For sufficiently small label noise, learning progresses positively in the signal direction. To demonstrate this, the loss gradient ratio of different samples $\ell^\prime(x_1) / \ell^\prime(x_2)$ is proved to be bounded at constant order across all time steps using induction arguments. This is a significant result of these studies.
> In contrast, for our setting, since the output scale is not learned, proving bounds on gradient ratios is unnecessary (as noted in Remark 4). Instead, **the difficulty lies in the interplay of the softmax probabilities, where the clean-to-noisy sample ratio alone cannot determine the learning direction of signals**. This challenge is discussed in the second paragraph of Section 4.3.
>
> While we follow the paper construction and the decomposition of the learned parameters into signal and noise components as used in existing CNN papers, the token selection setting in our work introduces fundamentally different difficulties. **We believe these challenges cannot be addressed by a straightforward combination of existing analyses.** If necessary, we can add detailed comparisons of these analytical techniques in the appendix to further clarify these differences.
>
> Once again, thank you for your precious time.

---

> > ### Comment · Reviewer_wjUP · 2024-12-01
> >
> > I don't quite understand the point `` Specifically, the difference is whether convergence is driven by the convergence of each example's loss $\ell'$ or by the convergence of the gradient of the model's output $\nabla_W f(W)$''. The authors seem to say that existing works only characterize the convergence of the loss function instead of the model outputs. However, how can we achieve the convergence of the model outputs by using gradient descent? Why is it meaningful to achieve the convergence of the model outputs? I don't think this difference can be treated as a contribution to help solve this problem.
> >
> > Moreover, I noticed this concurrent work [Jiang et al., 2024], which could be discussed.
> >
> > Jiang et al., Neurips 2024. Unveil Benign Overfitting for Transformer in Vision: Training Dynamics, Convergence, and Generalization.

---

> > > ### Author Response · Authors · 2024-12-01
> > > **Response to Reviwer wjUP (2/2)**
> > >
> > > ---
> > > > Q2: Moreover, I noticed this concurrent work [Jiang et al., 2024], which could be discussed.
> > > Jiang et al., Neurips 2024. Unveil Benign Overfitting for Transformer in Vision: Training Dynamics, Convergence, and Generalization.
> > >
> > > Thank you for your comments. We are aware of the mentioned paper and would like to clarify the following, keeping the double-blind policy in mind:
> > > - Our paper was made publicly available on arXiv earlier.
> > > - [Jiang+, NeurIPS2024] was released just a few days before the ICLR submission deadline.
> > >
> > > While [Jiang+, 2024] also analyzes benign overfitting with attention mechanism, its analysis setup and motivation fundamentally differ from ours in the following points.
> > > 1. **Absence of Label Noise**
> > > 	- Their setup assumes no noise in the target $y$, meaning all training data is clean. They focus on benign overfitting concerning the strength of input noises $\\{\mathbf{\epsilon}_t\\}_t$.
> > > 	- In contrast, **we focus on benign overfitting concerning label noise, a different problem formulation**. The behavior of softmax probabilities for noisy data differs significantly from that of clean data. The ability to fit label noise has not been analyzed before, and even when possible, the direction of signal learning is not straightforwardly determined by the sample ratio of clean and noisy data due to the intrinsic softmax difficulties. These challenges are unique to settings with label noise, as addressed in our work.
> > > 2. **Difference in Input Distribution**
> > > 	- [Jiang+, 2024] assumes orthogonality between signal and noise to simplify the analysis. As noted in our earlier comparison with [Tarzanagh+, 2023], this assumption overlooks the interactions between learned parameters and training samples, allowing for simplifications that ignore signal-noise interplay.
> > > 	- [Jiang+, 2024] considers only one relevant token (a token with a class signal) and other irrelevant tokens.
> > > 3. **Focus on ViT model**
> > > 	- [Jiang+, 2024] explores the advantage of Vision Transformer (ViT) compared to linear models and CNNs from the perspective of benign overfitting. Their analysis includes the optimization of self-attention and value matrix, which involve challenging interactions during training.
> > > 	- In contrast, **our study is motivated by a different question: examining the possibility and characteristics of benign overfitting in token selection mechanisms.** To address this, we focus on optimization within attention. This focus also allows for a data model aligning real-world scenarios more than [Jiang+, 2024], such as the label noise setting and realistic data assumptions, as described in points 1 and 2 above.
> > > 		- In our first response (2/3): *Since the benign overfitting in linear classifiers is already shown in the existing work [7, 8], if we also optimize the linear classifier head ν, then it would be non-trivial to determine whether benign overfitting is primarily driven by the attention mechanism or the linear classifier. Our work demonstrates that benign overfitting can be achieved solely through the token selection mechanism within the softmax, which is a contribution rather than a limitation.*
> > > 	- Furthermore, our motivation is not an artificial problem just for an analytical extension; optimization within the attention mechanism is a common setting in parameter-efficient fine-tuning techniques such as prompt-tuning and LoRA. The generalization performance when overfitting to noisy downstream tasks is of sufficient practical interest.
> > >
> > > Even though both studies share the keywords of benign overfitting and attention mechanism, the focus of each paper differs. This leads to differences in the analytical setups, as highlighted above. We believe these differences constitute a meaningful and sufficient contribution to the field.
> > > In the final version of our PDF, we will cite this concurrent work and include a paragraph discussing the comparison outlined above.

---

> ### Author Response · Authors · 2024-12-01
> **Response to Reviwer wjUP (1/2)**
>
> ---
> > Q1: I don't quite understand the point "Specifically, the difference is whether convergence is driven by the convergence of each example's loss ℓ′ or by the convergence of the gradient of the model's output ∇Wf(W)". The authors seem to say that existing works only characterize the convergence of the loss function instead of the model outputs. However, how can we achieve the convergence of the model outputs by using gradient descent? Why is it meaningful to achieve the convergence of the model outputs? I don't think this difference can be treated as a contribution to help solve this problem.
>
> We apologize for the confusing explanation. First, we clarify that the answer to the next is **NO**.
> > The authors seem to say that existing works only characterize the convergence of the loss function instead of the model outputs.
>
> Of course, existing research handles the convergence of the model output $f(x; \mathbf{W}^{(\tau)})$.
> In the following explanation, we consider the binary cross-entropy loss $\ell$ and its gradient $\ell^\prime$, noting that $0 < -\ell^\prime(z) < \ell(z)$. This ensures the convergence of $\ell$ to $0$ can be equivalently analyzed through $\ell^\prime$.
>
> The point "Specifically, ..." in our original response meant the following:
> We consider the parameter update of gradient descent at each step
> $$
> \mathbf{W}^{(\tau+1)} - \mathbf{W}^{(\tau)} = - \alpha \nabla_\mathbf{W} \mathcal{L}(\mathbf{W}^{(\tau)}),
> $$
> and assume that this update converges. Then, the question is whether this convergence results from:
> 1. The convergence of the binary cross-entropy $\ell_i$ for each training example $x_i$ (NN, CNN case), or
> 2. The convergence of the gradient of the model output $\nabla_\mathbf{W} f(x_i; \mathbf{W}^{(\tau)})$ (attention case).
>
>
> For NN and CNN, which are the focus of existing research on benign overfitting, it is possible to demonstrate the convergence of binary cross-entropy loss $\ell$. Below are some specific examples.
> - (Lemma 4.12, "Benign Overfitting without Linearity: Neural Network Classifiers Trained by Gradient Descent for Noisy Linear Data" Frei+2022) shows that for smooth activation function, $\\|\nabla_\mathbf{W} \mathcal{L}(\mathbf{W}^{(\tau)})\\|\_F$ can be bounded from below using a surrogate loss involving $\ell^\prime$. The convergence of $\nabla_\mathbf{W}\mathcal{L}(\mathbf{W}^{(\tau)})$ implies the overfitting immediately.
> - (Lemma C.1 Q(t), "Benign overfitting of non-smooth neural networks beyond lazy training" Xu+2023) shows that the increase in margin $y_if(x_i; \mathbf{W}^{(\tau)})$ is bounded within bounds proportional to the surrogate loss. This ensures the convergence of the loss function.
> - (Lemma D.4, "Benign Overfitting in Two-layer Convolutional Neural Networks" Cao+2022) and (Lemma D.5, Benign Overfitting in Two-layer ReLU Convolutional Neural Networks Kou+2023) define the desirable direction $\mathbf{W}^*$ of the parameter and show that the update $\\|\mathbf{W}^{(\tau)} - \mathbf{W}^*\\|_F - \\|\mathbf{W}^{(\tau+1)} - \mathbf{W}^*\\|_F$ can be bounded from below using $\mathcal{L}(\mathbf{W}^{(\tau)})$. This demonstrates the convergence of loss function with sufficiently large $\tau$.
>
> In these studies, analyzing the convergence of model output $f(x; \mathbf{W}^{(\tau)})$ is crucial. Moreover, since the scale of the model output can be learned, $\nabla_\mathbf{W} f(x_i; \mathbf{W})$ does not typically converge to $0$.
>
>
> In contrast, the analysis of attention mechanism with a softmax function is fundamentally different as follows:
> - The model output is merely an affine combination of the input sequence (weighted by the learned probabilities). Therefore, the binary cross-entropy $\ell$ cannot converge to $0$.
> - As softmax approaches hardmax, the gradient of the model output $\nabla_\mathbf{W} f(x_i; \mathbf{W})$ converges to $0$. Although we can show that $\nabla_\mathbf{W} \mathcal{L}(\mathbf{W}^{(\tau)})$ converges to $0$ with a sufficiently small learning rate using the fact that the objective function $\mathcal{L}$ is smooth, this does not imply $\ell$ approaches $0$. Instead, it reflects the convergence of the gradient of the model output $\nabla_\mathbf{W} f(x_i; \mathbf{W}^{(\tau)})$.
> This introduces the challenge of tracking the paramter from time-step $0$ to ensure that the learned parameter fits training data in terms of 0-1 loss, by selecting the correct token.
>
> Our previous response emphasized the essential differences between existing research on benign overfitting in NN and CNN and our analysis of the attention model.
> This addressed your concerns that our work might merely adapt the existing analysis approach without novelty.
> Combined with the the differences in the training loss analysis detailed here and the generalization analysis explained earlier response, we believe **our study represents a novel and independent approach to analyzing benign overfitting in the unique setting of token selection.**

---

### Official Review · Reviewer_auqU · 2024-11-04

**Soundness:** 3
**Presentation:** 2
**Contribution:** 2
**Rating:** 5
**Confidence:** 3

**Summary:**

This paper presents theoretical analysis of benign overfitting in transformer attention mechanisms, examining how these models can perfectly fit noisy training data while maintaining generalization capability. Authors have maintained that it is the first work to analyze benign overfitting in the attention mechanism of transformer models. Based on a set of assumptions, the work provides theoretical guarantee for token selection behavior and characterizes conditions for benign versus harmful overfitting. While the paper makes a novel theoretical contribution to understanding attention mechanisms, several critical limitations make it difficult to comprehend the full significance of this work. The highly simplified setting, strong assumptions, and lack of real-world validation raise concerns about the practical relevance of the findings. Here’s what this reviewer feels about the proposed method

**Strengths:**

Originality:
First theoretical analysis of benign overfitting in attention mechanisms and unique analysis of token selection behavior and how attention handles noisy labels. Several setting build on existing benign overfitting analysis frameworks

Significance:
Good theoretical insights into transformer attention behavior but limited practical impact due to simplified setting and need for strong assumptions. Results may not translate to real-world applications and missing key transformer applications (e.g., next token prediction, multi-head attention)

Summary of Strengths:
1.	Novel theoretical contribution to understanding attention mechanisms
2.	Clear mathematical analysis and proofs
3.	Well-structured validation of theoretical findings
4.	Potential implications for prompt-tuning applications based classification

**Weaknesses:**

1.	The proposed method has been implemented in a oversimplified setting with single-layer attention and binary classification through single prompt-tuning. Hence, it is difficult to give verdict that the method can be a general tool for understanding benign overfitting in transformers’ attention mechanism setting.
2.	Some assumptions in the paper seems very strong or unrealistic, for example, in the data distribution, a constant ratio of relevant tokens and weakly-relevant tokens in an input sequence is assumed, with experiment done on single relevant and irrelevant token in a sequence assumed. This may certainly differ in a real data distribution
3.	Limited empirical validation on the proposed theories with only small synthetic data. The applicability of the proposed method is also unclear and there is not enough discussion on the scalability.
4.	This paper is a first step of understanding benign overfitting on attention mechanism, however, it is not clear from reading the paper the real motivation, applicability or significance of the analysis and understanding of benign overfitting in transformer attention mechanism. This should be addressed in the writing.

**Questions:**

Please refer to the weakness section.

---

> ### Author Response · Authors · 2024-11-19
> **Response to Reviewer auqU (1/3)**
>
> Thank you for your detailed review and insightful questions. We hope that our answer has addressed all your concerns. We pray this reply will change your decision.
>
> ---
> > Q1: The proposed method has been implemented in a oversimplified setting with single-layer attention and binary classification through single prompt-tuning. Hence, it is difficult to give verdict that the method can be a general tool for understanding benign overfitting in transformers’ attention mechanism setting.
>
> A1: The analyzed model $f(\boldsymbol{\mathbf{X}}; \boldsymbol{\mathbf{p}}) = \boldsymbol{\mathbf{\nu}}^\top\boldsymbol{\mathbf{X}}^\top \mathbb{S}(\boldsymbol{\mathbf{X}}\boldsymbol{\mathbf{W}}^\top\boldsymbol{\mathbf{p}})$ follows the standard setup in existing studies on the training dynamics of attention mechanisms [1, 2]. We consider this model a good first step for analyzing benign overfitting in the attention because, as our paper demonstrates, even with such a model, there are scenarios where harmful overfitting occurs due to incorrect token selection. The theoretical setup is sufficiently challenging and non-trivial. Furthermore, the model setup that only attention weights are trained aligns well with some aspects of fine-tuning in the real-world application, as will be discussed in our response to Q4.
>
> Below, we provide comments on individual components for clarification.
>
> 1. Fix the key-query matrix $\boldsymbol{\mathbf{W}}$
> As we discussed in Section 3.4 and Lemma A.6 of our paper, the training of the key-query matrix $\boldsymbol{\mathbf{W}}$ plays an essentially equivalent role to that of $\boldsymbol{\mathbf{p}}$ in the token selection mechanism, and analyzing the dynamics of either $\boldsymbol{\mathbf{W}}$ or $\boldsymbol{\mathbf{p}}$ is sufficient. As a result, optimizing $\boldsymbol{\mathbf{W}}$ with fixed $\boldsymbol{\mathbf{p}}$ would yield exactly the same results as presented in our paper. This argument is consistent with the existing studies on the training dynamics of the attention mechanism [1, 2].
>
> 2. Fix the linear head $\boldsymbol{\mathbf{\nu}}$
> Since the benign overfitting in linear classifiers is already shown in the existing work [3, 4], if we also optimize the linear classifier head $\boldsymbol{\mathbf{\nu}}$, then it would be non-trivial to determine whether benign overfitting is primarily driven by the attention mechanism or the linear classifier. **Our work demonstrates that benign overfitting can be achieved solely through the token selection mechanism within the softmax, which is a contribution rather than a limitation.** Additionally, fixing the final layer is a commonly used setting in the existing analysis of benign overfitting in neural networks [5, 6, 7, 8, 9].
>
> 3. Binary classification
> While we focused on the binary classification case in the main text, Section F in the appendix discusses how the analysis can be extended to the multi-class setting without fundamentally changing the arguments. This has already been mentioned at the end of the Introduction as follows:
> **In this paper, we consider binary classification for simplicity of discussion, but as shown in Section F, it can be extended to the multi-class setting without fundamentally modifying the argument.**

---

> ### Author Response · Authors · 2024-11-19
> **Response to Reviewer auqU (2/3)**
>
> ---
> > Q2: Some assumptions in the paper seems very strong or unrealistic, for example, in the data distribution, a constant ratio of relevant tokens and weakly-relevant tokens in an input sequence is assumed, with experiment done on single relevant and irrelevant token in a sequence assumed. This may certainly differ in a real data distribution
>
> A2: The constant ratio assumption is just a simplification introduced for the notation simplicity and can be removed. **We have newly added Section H.1** to explicitly state which parts would need to be revised and confirm that our analysis does not fundamentally rely on this assumption.
>
> Building on the previous research, we have carefully designed the analysis setting to better align with real-world scenarios. First, in the analysis of benign overfitting, it is necessary to explicitly model the input distribution for generalization analysis without relying on the classical generalization bound using a uniform convergence argument. The existing benign overfitting analysis in CNN models commonly adopts a simplified data model where input images are composed of only two separate parts: signal and noise vectors. [7, 8, 9]. Similarly, in existing studies on the training dynamics of attention mechanisms, it is often assumed that each example contains only a single optimal token [1, 10, 11] and a constant ratio of signal tokens [2, 12]. Our study introduces several realistic elements to the analysis, including the multiple relevant tokens, noises in relevant tokens, and intermediate states termed *weakly relevant tokens.*
> Furthermore, we have added new experiments on multiple real-world images and natural language datasets in Section I of the appendix, which will be discussed in more detail in the next Question 3. We also examine how our data model captures the aspects of these real-world datasets.
>
> ---
> > Q3: Limited empirical validation on the proposed theories with only small synthetic data. The applicability of the proposed method is also unclear and there is not enough discussion on the scalability.
>
> A3: **We have newly added the experiments on real-world datasets** in Section I in the appendix. Specifically, we conducted experiments using pre-trained ViT and BERT models, fine-tuning only the last attention layer on downstream tasks with label noise. These tasks include image classification on MNIST, CIFAR10, STL-10, and MedMNIST (pneumonia, breast) and text classification on SST-2, AG-news, and TREC, comprising a total of eight real datasets. We showed the generalization error of the model overfitted to noisy downstream tasks by varying the label noise. This experiment can be interpreted within our theorem's framework by treating the layers up to the final attention layer as fixed feature extractors and feeding their output feature vectors into the last trainable layer.
> In this fine-tuning experimental setup, as will be discussed in our response to the next Question 4, please note that the classifier head is pre-trained and fixed, aligning with the theoretical settings of our paper.
>
> ---
> > Q4: This paper is a first step of understanding benign overfitting on attention mechanism, however, it is not clear from reading the paper the real motivation, applicability or significance of the analysis and understanding of benign overfitting in transformer attention mechanism. This should be addressed in the writing.
>
> A4: Thank you for pointing this out. We have updated the last part of the Introduction to emphasize the applicability and significance of our analysis.
> **From a practical perspective, benign overfitting solely through the token selection mechanism has implications, particularly in parameter-efficient fine-tuning. For example, prompt-tuning [Li & Liang, 2021; Lester et al., 2021] trains only the tunable input token, and LoRA [Hu et al., 2022] focuses on training only the attention weights. The presence of benign overfitting suggests that adapting the model to low-quality downstream tasks with label noise would not be problematic in terms of generalization.**
>
> Additionally, we emphasize that we have appended new experiments on real-world datasets, as detailed in our response to Question 3.

---

> ### Author Response · Authors · 2024-11-19
> **Response to Reviewer auqU (3/3)**
>
> ---
> References
> [1] Davoud Ataee Tarzanagh, Yingcong Li, Xuechen Zhang, and Samet Oymak. Max-margin token selection in attention mechanism. In Thirty-seventh Conference on Neural Information Processing Systems, 2023b
> [2] Samet Oymak, Ankit Singh Rawat, Mahdi Soltanolkotabi, and Christos Thrampoulidis. On the role of attention in prompt-tuning. In International Conference on Machine Learning, pp. 26724–26768. PMLR, 2023.
> [3] Niladri S Chatterji and Philip M Long. Finite-sample analysis of interpolating linear classifiers in the overparameterized regime. Journal of Machine Learning Research, 22(129):1–30, 2021.
> [4] Yuan Cao, Quanquan Gu, and Mikhail Belkin. Risk bounds for over-parameterized maximum margin classification on sub-gaussian mixtures. Advances in Neural Information Processing Systems, 34:8407–8418, 2021.
> [5] Spencer Frei, Niladri S Chatterji, and Peter Bartlett. Benign overfitting without linearity: Neural network classifiers trained by gradient descent for noisy linear data. In Conference on Learning Theory, pp. 2668–2703. PMLR, 2022.
> [6] Xingyu Xu and Yuantao Gu. Benign overfitting of non-smooth neural networks beyond lazy training. In International Conference on Artificial Intelligence and Statistics, pp. 11094–11117. PMLR, 2023.
> [7] Yuan Cao, Zixiang Chen, Misha Belkin, and Quanquan Gu. Benign overfitting in two-layer convolutional neural networks. Advances in neural information processing systems, 35:25237–25250, 2022.
> [8] Yiwen Kou, Zixiang Chen, Yuanzhou Chen, and Quanquan Gu. Benign overfitting in two-layer relu convolutional neural networks. In International Conference on Machine Learning, pp. 17615–17659. PMLR, 2023.
> [9] Xuran Meng, Difan Zou, and Yuan Cao. Benign overfitting in two-layer reLU convolutional neural networks for XOR data. In Forty-first International Conference on Machine Learning, 2024.
> [10] Davoud Ataee Tarzanagh, Yingcong Li, Christos Thrampoulidis, and Samet Oymak. Transformers as support vector machines. arXiv preprint arXiv:2308.16898, 2023a.
> [11] Bhavya Vasudeva, Puneesh Deora, and Christos Thrampoulidis. Implicit bias and fast convergence rates for self-attention. arXiv preprint arXiv:2402.05738, 2024.
> [12] Deora, Puneesh, et al. "On the Optimization and Generalization of Multi-head Attention." Transactions on Machine Learning Research.

---

### Official Review · Reviewer_xrjE · 2024-11-04

**Soundness:** 3
**Presentation:** 2
**Contribution:** 2
**Rating:** 6
**Confidence:** 3

**Summary:**

Authors investigate "benign overfitting" (where model generalizes as well as overfit the training data) for attention architecture. Benign overfitting has not yet been defined for transformers, so the authors take a first step by studying it in the context of token selection. They theoretically prove its existence and studies different training trajectories when it can happen.

**Strengths:**

- Novelty: Authors are first to study it for transformer models.
- Contributions seems to be rigorously proved.

**Weaknesses:**

- Paper is challenging to follow.

Scope/Practicality of the paper seems limited as:
- Papers made various assumptions both while formulating the problem (like tokens being splitting nicely into three groups) and while defining the data.
- Only synthetic data has been used.

**Questions:**

1. Please justify the use of synthetic data.
2. Does real data also adhere to the various assumptions made?

**Details Of Ethics Concerns:**

No concern.

---

> ### Author Response · Authors · 2024-11-19
> **Response to Reviewer xrjE (1/2)**
>
> Thank you for your insightful question. We hope that our answer has addressed all your concerns. We would greatly appreciate it if our reply is reflected in your score.
>
> ---
> > Q1: Paper is challenging to follow.
>
> A1: The structure of our paper aligns with the flow of existing studies on benign overfitting [1, 2, 3, 4, 5, 6]. To enhance clarity, we have added the following statement in Section 2 to indicate a comparison table provided in the appendix:
> **Please also refer to Table 2 in the appendix for the comparison of existing work, including ours.**
>
> Additionally, although the proof sketch is provided in Appendix A.2, there is no reference to it in the main text. To address this, we have added the following sentence at the end of Section 4.2:
> **The proof sketch for the main theorems is provided in Section A.2, while the proof of Theorem 4.2 is detailed in Section D.**
>
> Furthermore, to emphasize the practical takeaways, we have updated the last part of the Introduction to provide a more detailed explanation, as follows.
> **From a practical perspective, benign overfitting solely through the token selection mechanism has implications, particularly in parameter-efficient fine-tuning. For example, prompt-tuning [Li & Liang, 2021; Lester et al., 2021] trains only the tunable input token, and LoRA [Hu et al., 2022] focuses on training only the attention weights. The presence of benign overfitting suggests that adapting the model to low-quality downstream tasks with label noise would not be problematic in terms of generalization.**
>
> ---
> > Q2: Papers made various assumptions both while formulating the problem (like tokens being splitting nicely into three groups) and while defining the data.
>
> A2: We have carefully designed the analysis setting to better align with real-world scenarios, building on previous research.
> We have also provided specific real-world examples for the three token groups in our response to Question 5. Please refer to them as well.
>
> The existing analysis of benign overfitting in CNN models commonly adopts a simplified data model where input images are composed of only two separate parts: signal and noise vectors. [5, 6, 7]. Similarly, in existing studies on the training dynamics of attention mechanisms, it is often assumed that each example contains only a single optimal token [8, 9]. Our study introduces several realistic elements to the analysis, including the multiple relevant tokens, noises in relevant tokens, and intermediate states termed *weakly relevant tokens*. Furthermore, even under this data model, analyzing benign overfitting in attention mechanism remains challenging and non-trivial. We highlight that there exists a scenario where incorrect token selection occurs, leading to harmful overfitting.
>
> ---
> > Q3: Only synthetic data has been used.
>
> A3: **We have newly added the experiments on real-world datasets** in Section I in the appendix. Specifically, we conducted experiments using pre-trained ViT and BERT models, fine-tuning only the last attention layer on downstream tasks with label noise. These tasks include image classification on MNIST, CIFAR10, STL-10, and MedMNIST (pneumonia, breast) and text classification on SST-2, AG-news, and TREC, comprising a total of eight real datasets. We showed the generalization error of the model overfitted to noisy downstream tasks by varying the label noise. This experiment can be interpreted within our theorem's framework by treating the layers up to the final attention layer as fixed feature extractors and feeding their output feature vectors into the last trainable layer.
>
> ---
> > Q4: Please justify the use of synthetic data.
>
> A4: The experiments on synthetic data presented in Section 5 are essential for empirically validating the claims of the main theorems. The experimental setups are totally aligned with the theoretical settings of our paper. Specifically, under the condition where the balance between signal and noise is satisfied, as assumed in the main theorems, we observed that the tokens predicted by our theoretical analysis are indeed selected for both clean and noisy data in the training set. This is shown in Figure 3 (a), where benign overfitting is successfully achieved.

---

> ### Author Response · Authors · 2024-11-19
> **Response to Reviewer xrjE (2/2)**
>
> ---
> > Q5: Does real data also adhere to the various assumptions made?
>
> A5: Our data model reflects the real-world setting well enough for the theoretical analysis.
>
> For example, in the case of image patch sequences, certain regions containing substantial class information and useful for classification, regions representing irrelevant background noise, and ambiguous regions can correspond to relevant, irrelevant, and weakly relevant tokens in our data model. Similarly, for word sequences, the analogies of relevant tokens can be considered: in topic classification, words strongly associated with the topic (e.g., baseball for sports class), and in sentiment analysis, words that express emotions well (e.g., attractive for positive sentiment). Below, we provide examples of the real-world experiment added in A3, explaining specific situations.
>
> Example 1: Lung cancer detection via images (e.g., MedMNIST)
> - Relevant token: A patch showing a tumor or lesion
> - Weakly relevant tokens: Patches showing enlarged lymph nodes or inflammatory signs are weakly relevant. These features often co-occur with lung cancer but are not definite indicators.
> - Irrelevant tokens: Patches showing normal lung tissue or background anatomy
>
> Example 2: Topic classification of news headlines (e.g., AG-news)
> “*Breakthrough Vaccine Brings Hope to Millions*”, categorized under the topic Health & Science.
> - Relevant token: “Vaccine” is crucial for identifying Health & Science.
> - Weakly relevant tokens: Words like “Breakthrough” and “Hope” are used in the positive sentiment, and they often co-occur with technology, business, and health-related terms.
> - Irrelevant tokens: Words like “Brings”, “to”, “Millions” are general terms.
>
> Finally, as discussed in our response to Q2, we emphasize that employing such data models is a common approach in the existing theoretical analysis.
>
> ---
> References
> [1] Niladri S Chatterji and Philip M Long. Finite-sample analysis of interpolating linear classifiers in the overparameterized regime. Journal of Machine Learning Research, 22(129):1–30, 2021.
> [2] Yuan Cao, Quanquan Gu, and Mikhail Belkin. Risk bounds for over-parameterized maximum margin classification on sub-gaussian mixtures. Advances in Neural Information Processing Systems, 34:8407–8418, 2021.
> [3] Spencer Frei, Niladri S Chatterji, and Peter Bartlett. Benign overfitting without linearity: Neural network classifiers trained by gradient descent for noisy linear data. In Conference on Learning Theory, pp. 2668–2703. PMLR, 2022.
> [4] Xingyu Xu and Yuantao Gu. Benign overfitting of non-smooth neural networks beyond lazy training. In International Conference on Artificial Intelligence and Statistics, pp. 11094–11117. PMLR, 2023.
> [5] Yuan Cao, Zixiang Chen, Misha Belkin, and Quanquan Gu. Benign overfitting in two-layer convolutional neural networks. Advances in neural information processing systems, 35:25237–25250, 2022.
> [6] Yiwen Kou, Zixiang Chen, Yuanzhou Chen, and Quanquan Gu. Benign overfitting in two-layer relu convolutional neural networks. In International Conference on Machine Learning, pp. 17615–17659. PMLR, 2023.
> [7] Xuran Meng, Difan Zou, and Yuan Cao. Benign overfitting in two-layer reLU convolutional neural networks for XOR data. In Forty-first International Conference on Machine Learning, 2024.
> [8] Davoud Ataee Tarzanagh, Yingcong Li, Xuechen Zhang, and Samet Oymak. Max-margin token selection in attention mechanism. In Thirty-seventh Conference on Neural Information Processing Systems, 2023b
> [9] Bhavya Vasudeva, Puneesh Deora, and Christos Thrampoulidis. Implicit bias and fast convergence rates for self-attention. arXiv preprint arXiv:2402.05738, 2024.

---

### Meta-Review · Area_Chair_RfhW · 2024-12-18

**Metareview:**

This submission received mixed reviews. During the discussion phase, Reviewer HY7u, who gave the highest rating (8), did not participate in the discussion, and the associated comments were quite brief, providing insufficient grounds to champion the paper for acceptance.

Meanwhile, Reviewers vumC and wjUP raised key concerns about the paper’s contributions, summarized as follows:

(1) The model is perceived as overly simplistic and does not sufficiently differentiate itself from existing works, even if it addresses a different problem.

(2) The real-world experiments presented at the end of the Appendix focus primarily on noisy training and partially verify the theory. However, more practical insights that could guide real-world improvements are preferred.

Given these concerns and the current state of the submission, I regretfully recommend rejection.

**Additional Comments On Reviewer Discussion:**

Reviewers acknowledged the authors' efforts in their rebuttal. However, major concerns remain: (1)The model is viewed as overly simplistic and does not sufficiently differentiate itself from existing works, even if addressing a different problem. (2) The real-world experiments in the Appendix primarily focus on noisy training and only partially verify the theory. More practical insights to guide real-world improvements are needed. Additionally, Reviewer HY7u, who provided the highest rating (8), did not participate in the discussion, and the brief comments offer insufficient grounds to advocate for acceptance.

---

### Decision · Program_Chairs · 2025-01-22

Reject